Resource

# Multi-omics identify hallmark protein and lipid features of small extracellular vesicles circulating in human plasma

**Alin Rai** [1,2,3] ✉, **Kevin Huynh**[1,2,3], **Jonathon Cross** [1], **Qi Hui Poh**[1], **Haoyun Fang**[1,2], **Bethany Claridge**[1], **Thy Duong** [1], **Carla Duarte**[1], **Jonathan E. Shaw**[1,3,4], **Thomas H. Marwick**[1,2,3], **Peter Meikle** [1,2,3] & **David W. Greening** [1,2,3] ✉

Extracellular vesicles (EVs) are an essential signalling entity in human plasma implicated in health and disease. Still, their core protein and lipid componentry, which lie at the centre of EV form and function, remain poorly defined. Here we performed high-resolution density gradient fractionation of over 140 human plasma samples to isolate circulating EVs, and systematically constructed their quantitative proteome (4,500 proteins) and lipidome (829 lipids) landscapes using mass spectrometry. We identified a highly conserved panel of 182 proteins (including ADAM10, STEAP23 and STX7) and 52 lipids (including PS, PIPs, Hex2Cer and PAs), providing a deep survey of hallmark molecular features and biological pathways characteristic to circulating EVs. We also mapped the surfaceome diversity, identifying 151 proteins on the EV surface. We further established a set of 42 proteins and 114 lipids features that served as hallmark features of non-EV particles in plasma. We submit ADAM10 and PS(36:1) as conserved EV biological markers that precisely differentiate between EV and non-EV particles. Our findings, which can be explored via an open-source Shiny web tool (evmap.shinyapps.io/evmap/), will serve as a valuable repository to the research community for a clearer understanding of circulating EV biology.

Extracellular vesicles (EVs) are membrane-enclosed nanoscale particles (30–1,000 nm in diameter) released by cells into their extracellular space[1,2]. By transferring bioactive cargo such as proteins, lipids, nucleic acids and metabolites between cells[3–6], EVs execute diverse biological functions in various physiological and pathological processes. EVs are also found ubiquitously in the human circulatory system[7–15], with an estimated $50 \times 10^6$ particles per millilitre of plasma[9], and are now recognized as an essential signalling entity in human plasma. While their precise functions in humans remain mostly elusive, circulating EVs have been implicated in essential life processes, including immune

regulation[16], interorgan crosstalk[17,18], tissue homeostasis and regeneration[19] and coordinating physiological responses[20–22]. Dysregulation of their cargo composition is also implicated in various human diseases such as cancer[23], cardiovascular disease[24], artery calcification[20] and coronavirus disease 2019 pathogenesis[25]. As minimally invasive liquid biopsies, they have also garnered interest for their potential diagnostic value[9] and real-time monitoring of therapeutic response[26,27].

Despite their wide biological and biomedical implications, the study of circulating EVs remains exceptionally challenging[28], with their conserved protein and lipid composition still poorly

[1]Baker Heart and Diabetes Institute, Melbourne, Victoria, Australia. [2]Baker Department of Cardiometabolic Health, University of Melbourne, Melbourne, Victoria, Australia. [3]Baker Department of Cardiovascular Research Translation and Implementation, La Trobe University, Melbourne, Victoria, Australia. [4]School of Public Health and Preventive Medicine, Monash University, Melbourne, Victoria, Australia. ✉e-mail: alin.rai@baker.edu.au; david.greening@baker.edu.au

characterized—representing an unmet milestone in the field. Given that conserved molecular blueprint epicentres form and function of a biological system—sharing unique proximity to phenotype and pathophysiology[29]—closing this knowledge gap is critical for advancing our fundamental understanding of circulating EVs and harnessing their clinical potential. Recently, while several seminal studies[30,31] have begun to define the precise core of EVs using in vitro culture systems, making such discoveries for EVs circulating in human plasma remains a formidable challenge. This is mainly due to the presence of large abundance of non-EV components (such as lipoprotein particles, soluble proteins, complement proteins and immunoglobulins) in plasma that outnumber EVs by six to seven orders of magnitude[32]. Such non-EV particles copurify with EVs, which invariably limits mass spectrometry (MS)-based quantifications primarily to high-abundant plasma proteins or lipoproteins particle-associated lipids, resulting in incomplete and low-coverage data[14,23,25]. Thus, there is an ongoing international effort within the EV community[9,11,31,33–35] to develop and refine plasma EV (pEV) isolation methods, with the aim of precisely defining their molecular landscape[10,12,14,36,37].

A systematic construction of the conserved protein and lipid componentry of circulating EVs in humans has several implications. It will inform us on fundamental building blocks of circulating EVs, providing high-confidence molecular maps and associated biological pathways such as biogenesis (including membrane curvature, stability and cargo recruitment), release, environmental interactions and uptake mechanisms in humans. Given that existing EV markers obtained from cell culture systems have limited conservation in humans[23]—potentially due to unique architecture and context of human tissues[38,39]—these conserved features will also serve as robust EV markers applicable to human plasma that can be rigorously implemented in large-scale population studies, advance EV purification and characterization techniques, and extend international EV guidelines[28] for standardized circulation EV research. In addition, these conserved features will bridge the knowledge gap between humans and cultures/animal models, enhancing knowledge transferability and translatability. Other implications include informing on strategies for engineering and functionalizing EV membranes for improved drug delivery vehicles (for example, extended half-life in circulation), and developing safer EV-based therapies.

In this study, we used high-resolution density gradient separation (DGS) to isolate a major EV subtype, known as small EVs, from human plasma. We verify the enrichment strategy and EV identity using various biochemical and biophysical characterization, ensuring a high degree of separation of EVs and non-EV particles in plasma. We then construct their detailed proteome and lipidome maps, identifying 182 proteins and 52 lipids as core protein and lipid componentry of EVs, which we refer to as EV hallmark features. We also identify 29 proteins and 114

lipids that are defining features of non-EV particles. These markers, in particular ADAM10 and PS(36:1), enable precise differentiation between EV and non-EV particles using machine learning. To enhance data accessibility, we developed an open-source R/Shiny web tool (evmap.shinyapps.io/evmap/).

## Results

### Isolation of circulating EVs from human plasma

While there is no method that can isolate EVs to absolute purity, high-resolution iodixanol-based DGS[40–42] remains a powerful strategy for enriching EVs from complex biofluids such as plasma. To this end, we subjected plasma to top-down DGS (Fig. 1a,b), which resolved the majority of signals for abundant plasma components such as albumin, apolipoproteins and argonaut 2 (AGO2, known to associate with non-EV RNA extracellularly[43]) to DGS fractions 1–5. By contrast, CD63 (tetraspanins that are found in small EVs) resolved in DGS fractions 6–8 (corresponding to flotation buoyancy of ~1.09 g ml⁻¹, typical of EVs) (Fig. 1a,b). This level of resolution was not achievable with ultracentrifugation alone (100,000$g$, referred to as p100K), which co-isolated large amounts of abundant plasma components (Fig. 1c and Supplementary Fig. 1a).

EV particles in DGS 6–8 fractions (pEVs) were morphologically intact, membrane-limited spherical vesicles, consistent with previous reports[44–47] (Fig. 1d), with a mean diameter of 220.4 nm (Supplementary Fig. 1b). By contrast, the p100K fraction contained abundant proteinaceous material that formed aggregates, whereas DGS 1–3, referred to as pDGS.LD (plasma DGS light-density particles), contained spherical structures lacking an apparent lipid membrane—resembling the size and morphology of lipoprotein particles[8].

Moreover, nanoparticle tracking analysis (Fig. 1e) revealed that pEVs ranged from 30 to 300 nm in size and displayed a net negative charge (Fig. 1f), consistent with a previous report[48]. Thus, our data show that we can successfully enrich for small EVs from human plasma with minimal contamination from non-EV plasma components. We further demonstrate EV enrichment from multiple plasma samples (Supplementary Fig. 1c,d). From 1 ml of plasma, we obtained ~8.7 μg of pEVs (a striking >24,000 fold reduction compared with total plasma protein) (Supplementary Fig. 1e), corresponding to ~4.2 × 10⁹ particles (Supplementary Fig. 1f,g). To assess separation efficiency, we also performed bottom-loaded DGS by carefully loading plasma at the bottom of the gradient before ultracentrifugation (Supplementary Fig. 1h). While EVs were recovered in similar fractions as in the top-loaded approach (cluster of differentiation (CD)63 signal in fraction 6), bottom-loaded DGS resulted in greater cofractionation of non-EV components, including ALB and APOA1, across multiple fractions. This suggests that, compared with top-loaded DGS, bottom-loaded DGS does not achieve the same resolution in plasma samples.

**Fig. 1 | Mapping the core proteome of circulating EVs in human plasmas.**
**a**, Workflow for DGS of human plasma (0.5 ml) for EV isolation and characterization. **b**, Western blot analysis of 12 DGS fractions (vol:vol matched) of human plasma with antibodies against indicated proteins. **c**, Western blot analysis of unprocessed plasma, p100K and DGS fractions 6–8 (pooled). **d**, Cryogenic electron microscopic images of p100K, DGS fractions 1–3 and 6–8. Scale bar, 500 nm ($n$ = 3, independent plasma samples). **e**, Size distribution (particle diameter (nm)) of plasma DGS fractions 1–3 and 6–8 ($n$ = 7, independent plasma samples) based on nanoparticle tracking analysis. Small EVs released by SW620 cells (in vitro EVs ($n$ = 5, biological replicates)) were also analysed. **f**, The zeta potential of DGS fractions 1–3 and 6–8 at 11 positions throughout the sample cell (top), or stationary layers in the sample cell (bottom) ($n$ = 8 for pDGS.LD and $n$ = 7 for pEVs, independent samples). The boxplot show the median (centre line), 25th–75th percentiles (box), minima and maxima within 1.5× interquartile range (whiskers) and outliers beyond. **g**, Proteome landscape construction of pEVs ($n$ = 38 independent plasma samples, 5 independent isolation sets; Supplementary Fig. 2) and NonEVs ($n$ = 42, independent plasma

samples). **h**, PCA of quantified proteins in pEVs and NonEVs. **i**, Occurrence analysis of proteins in 38 pEV proteomes where category 1 proteins represent ubiquitously quantified proteins. GO pathways enriched (Benjamini–Hochberg-corrected adjusted $P$ value <0.05) in each category of proteins are indicated. **j**, Scatter plot representing differential abundance ($P$ < 0.05) of category 1–3 proteins in pEVs compared with NonEVs. Two-sided $t$-tests were used to compare protein abundances between pEVs and non-EVs. Proteins with $P$ < 0.05 are shown. Multiple-comparison adjustment (Benjamini–Hochberg FDR) was applied, and adjusted values are reported in Supplementary Table 7. **k**, KEGG pathways enriched in pEV or NonEV protein features. Molecular Function GO terms enriched for each protein list were computed using a two-sided hypergeometric test. $P$ values were adjusted for multiple testing with Benjamini–Hochberg (reported as adjusted $P$ values; cut-off 0.05). **l**, Surface-accessible category 1 proteins of pEVs categorized based on their molecular/functional annotation. C/R/T represents clusters of differentiation (CDs), receptors and transporters. The heatmap shows the abundance of the indicated proteins in pEVs ($n$ = 9, independent plasma samples) and in vitro EVs ($n$ = 3, biological replicates).

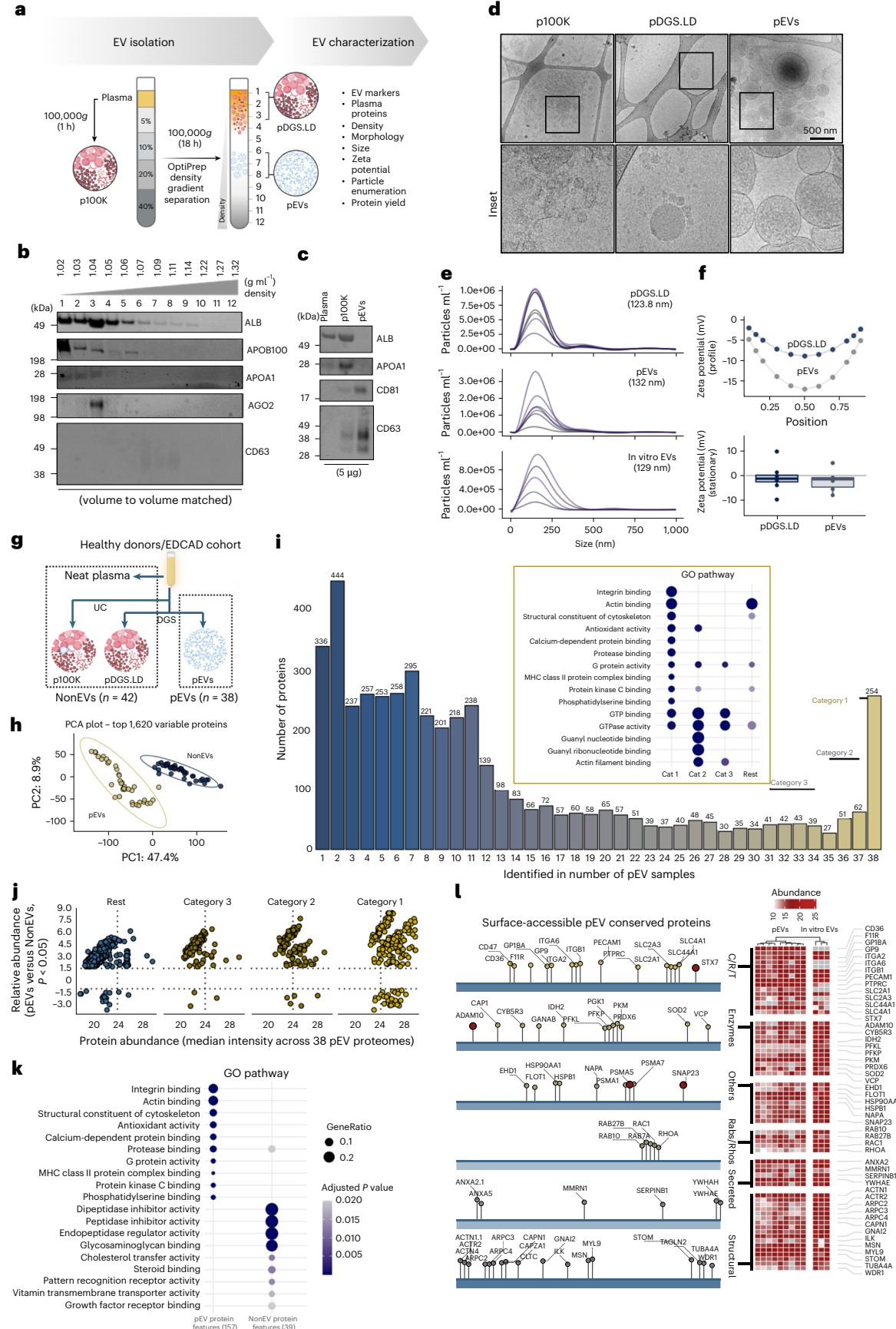

## Constructing the proteome draft of circulating EVs in humans

We next performed MS-based proteomics analysis of pEVs from 38 human plasma samples from multiple sources (Supplementary Fig. 2a). To identify EV-specific proteins, we compared the pEV proteomes with those of pDGS.LD, p100K and unprocessed plasma, collectively termed as NonEVs (n = 42) (Fig. 1g, Supplementary Fig. 2b and Supplementary Tables 1–6). Using stringent peptide and protein identification criterion (1% false discovery rate, FDR), we quantified 4,631 proteins in pEVs and 1,678 in NonEVs (Supplementary Fig. 2c and Supplementary Table 6). The size of pEV proteome dataset was comparable to those obtained for in vitro EV proteome data from four different cell lines (4,492 proteins), ensuring sufficient proteome coverage.

The pEV proteome displayed a remarkable dynamic range (Supplementary Fig. 2d), including extensive coverage of low-abundant circulating proteins[49] (Supplementary Fig. 2e), and was distinct to the NonEV proteome (Fig. 1h and Supplementary Fig. 2f). Moreover, differentially abundant proteins in pEVs were enriched for terms/pathways associated with small EVs (regulation of actin cytoskeleton and endocytosis) (Supplementary Fig. 3a–c and Supplementary Tables 7 and 8) and EV biogenesis proteins[30,45,47,50,51] (Supplementary Fig. 3d). By contrast, NonEVs contained components related to complement and coagulation pathways, along with abundant plasma proteins[52] (Supplementary Fig. 3a–d), further supporting our EV enrichment pipeline at omics level. Importantly, although activated platelet EV proteins such as CLEC1B, PF4 and PPBP (which we previously reported[53]) contribute towards the pEV proteome landscape (Supplementary Fig. 4 and Supplementary Table 9), we observed an abundance of non-platelet EV proteins (Extended Data Fig. 1a–c), suggesting that the pEV proteome represents a diverse cellular source[15,54,55] (Extended Data Fig. 1d,e and Supplementary Table 10). Indeed, we quantified diverse cell-, tissue- and organ-associated proteins in pEVs[56,57], suggesting that our dataset provides a snapshot of the diverse vesicular population in circulation (Extended Data Fig. 2a–d and Supplementary Table 11): many of these cell signatures were also enriched in pEVs versus NonEVs (Extended Data Fig. 2e,f and Supplementary Table 12). Furthermore, our high-resolution MS quantified low-abundant but biologically functional molecules in pEVs, including signal transduction proteins, cytokines and chemokines, kinases[58], cell-surface receptors, transporters and CD proteins[59], RNA-binding proteins[60], and transcription factors[61] (Extended Data Fig. 3a–b, Supplementary Table 11). These include RNA binding proteins such as HNRNPK[62] and PCBP2[63], TFs such as NME2[64], kinases such as SRC[65], chemokines such as TGFB1[66], and receptors such as integrins[67], previously reported to be functionally active cargo of EVs. These molecules (for example, cytokines) exhibit abundances comparable to canonical CD proteins typical of EVs, strongly supporting their active incorporation (Extended Data Fig. 3c,d).

## Defining the core protein features of circulating EVs

Echoing a recent report[23], although several EV proteins (namely 22 core proteins reported in in vitro EVs[30] and 94 EV marker proteins

recommended by the minimal information for studies of extracellular vesicles (MISEV) guidelines[28,36]) were detected in pEVs (Supplementary Fig. 5a and Supplementary Table 6), many were not universally quantified (Supplementary Fig. 5b–d).

Meanwhile, occurrence analysis identified 259 proteins ubiquitously quantified in all 38 pEV samples (termed category 1 proteins; Fig. 1i and Supplementary Table 7); these proteins were enriched for EV-related terms and processes (Extended Data Fig. 4a and Supplementary Table 13) and encompass proteins associated with endosomal trafficking network (Extended Data Fig. 4b) such as flotillins, integrins and CD proteins, whose close interconnectedness is highlighted by their protein–protein interaction network (Extended Data Fig. 4c).

For pEV protein feature selection, we first selected proteins that were 100% conserved in pEV proteomes (that is, present in all 38 proteome datasets, category 1 proteins), which resulted in 259 proteins. Next, of these 259 proteins, we selected proteins with fold change >1.5 and P value <0.0001 in pEV versus NonEVs, resulting in 182 proteins, which we refer to as pEV protein features (Fig. 1j, Extended Data Fig. 5a and Supplementary Table 7; FDR <0.05, as reported in Supplementary Table 20).

For NonEV protein feature selection, we first selected proteins that were 100% conserved in NonEV proteomes (that is, present in all 42 proteome datasets), which resulted in 114 proteins. Next, of these 114 proteins, we selected proteins with fold change <−1.5 and P value < 0.0001 in pEV versus NonEVs, resulting in 42 proteins, which we refer to as NonEV protein features (Extended Data Fig. 5b,c and Supplementary Table 7; FDR <0.05, as reported in Supplementary Table 20).

Emphasizing their significance as essential EV components, the pEV protein features were enriched for classical EV pathways (Fig. 1k and Supplementary Table 14). These pEV protein features showed coordinated molecular patterns across functional groups associated with 'vesicular transport', 'actin cytoskeleton regulation' and 'membrane-raft assembly' (see Extended Data Fig. 6a for enrichment-based associations). Upon closer inspection, these correlated molecular patterns were conserved across individual EV proteomes at the cellular level or within specific EV subpopulations (that is, CD9/CD81/CD63-positive EVs) (Extended Data Fig. 6b and Supplementary Table 15). Moreover, these enrichment trends were observed in EVs from primary human fibroblasts (Extended Data Fig. 6c) and were enriched in EVs relative to parental cell proteomes[30] (Extended Data Fig. 6d). By contrast, NonEV protein features were enriched for 'endopeptidase inhibitor activity', 'complement and coagulation cascades' and 'cholesterol metabolism' (Fig. 1k and Supplementary Table 14).

We next investigated whether the pEV protein features also include the EV surfaceome, the essential interactive platform on EV surfaces[68]. To explore this, surface-accessible proteins on pEV were labelled with membrane-impermeant Sulfo-NHS-SS-Biotin, captured with avidin-coated beads and analysed through MS-based proteome profiling (n = 9) (Supplementary Fig. 6a and Supplementary Table 16). Our data revealed that 151 pEV protein features were surface-accessible, including CD proteins (CD44 and CD47), integrins (ITGA2, ITGA6 and

**Fig. 2 | Conservation of circulating EV protein features. a,** A heatmap depicting the quantification of EV protein features in our pEV and NonEV proteome datasets (only protein features quantified in <30% of NonEV datasets were used). **b,** A heatmap depicting the quantification of EV protein features in previous reported proteomes for circulating EVs: A–C, EV preparations from plasma or serum sourced from healthy individuals or those with known pathology[15,69]; D–F, EV subtypes (CD81+, CD63+ and CD9+ EVs) prevalent in the circulatory system[70]. As an orthogonal validation, the heatmap also depicts conservation of EV protein features using TMT-based isobaric multiplexing of pEVs (n = 4, independent plasma samples) and in vitro EV proteomes (n = 6) (G–J), and using label-free quantitative proteomics for in vitro EVs (n = 3 per cell line) (I–L). **c,** A scatter plot of fold change (FDR <0.05) EV protein features and NonEV protein features in the discovery set and validation set (AusDiab set, independent plasma samples). The rest of the proteins are indicated with grey points. Pearson r = 0.851, P < 2 × 10⁻¹⁶.

The dotted line represents the linear regression fit (the shaded band indicates the 95% confidence interval, CI). Fold-change correlation between discovery and validation set. Features were first screened by one-way ANOVA (two-sided) on vsn-normalized intensities with Benjamini–Hochberg FDR across features; significant features (FDR <0.05) were analysed using Tukey's honestly significant difference (HSD) test with Benjamini–Hochberg adjustment applied within each feature. Grey line: least-squares fit with 95% CI band. **d,** Density plot of proteins in pEV proteomes from individuals with either positive CAC score (CAC group) or zero CAC score (Healthy group). CAC_DOWN: proteins significantly (FDR <0.05) downregulated in pEVs from CAC versus Healthy. CAC_UP: proteins significantly (FDR <0.05) upregulated in pEVs from CAC versus Healthy. EV feature: 182 pEV protein features. NS_proteins: proteins with similar abundance between pEVs from CAC versus Healthy.

ITGB1) and annexins (ANXA2 and ANXA5) (Fig. 1l and Supplementary Fig. 6b,c). We further confirmed their surface accessibility in in vitro EVs (Fig. 1l and Supplementary Table 16). By contrast, the conserved SDCBP protein remained inaccessible to biotin capture, which is in direct agreement with its luminal localization[30].

By leveraging data from published proteomes[15,69,70], we demonstrated a remarkable conservation of pEV protein features in multiple plasma or serum EVs (Fig. 2a,b, A–C, and Supplementary Table 7). These features were also conserved in different EV subpopulations (CD81+, CD63+ and CD9+ EVs) prevalent in the circulatory system (Fig. 2b, D–F),

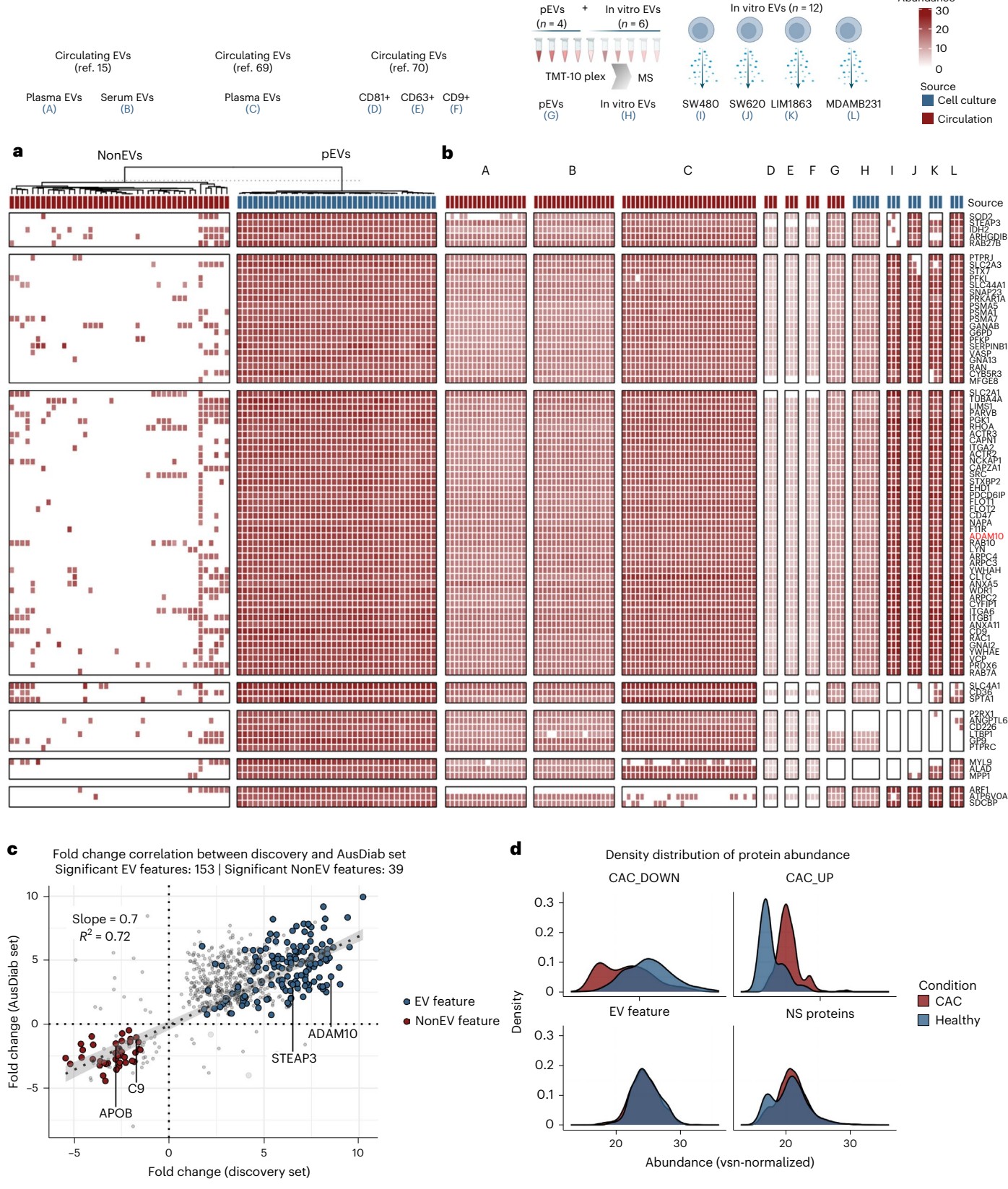

with 100% quantification of at least 74 proteins. Moreover, interrogating previous published sEV proteomes from 14 cell lines[30], 81 EV features were also conserved, displaying greater abundance in EVs compared with cells (Supplementary Fig. 7a,b). In addition, we experimentally verified their conservation in in vitro EVs ($n = 12$) release by four cell lines (Fig. 2b, I–L). Finally, we validated their conservation in pEVs ($n = 4$) and in vitro EVs ($n = 6$) using orthogonal tandem mass tag (TMT)-based isobaric multiplexed proteomics (Fig. 2b, G and H, Supplementary Table 26). Overall, 78 pEV proteins displayed 100% conservation, of which 63 were surface accessible (Supplementary Table 7).

To validate our identified pEV markers, we analysed plasma from an independent external cohort (AusDiab, $n = 12$ pEVs, $n = 12$ NonEVs) using the same EV isolation and proteomic pipeline (Supplementary Table 17). Comparative proteomic analysis revealed that 177 out of 182 pEV protein features and all 42 NonEV protein features were detected, with 87.4% (156/182) of pEV proteins and 92.9% (39/42) of NonEV proteins significantly enriched in their respective fractions (FDR <0.05) (Supplementary Table 18 and Supplementary Fig. 8), with similar enrichment in Gene Ontology (GO) pathways in our discovery cohort (Supplementary Table 19).

Comparative analysis of fold change of pEV and NonEV protein features between the discovery and validation datasets (Supplementary Table 20) demonstrated a strong positive correlation (Pearson $r = 0.851$, $P < 2 \times 10^{-16}$), confirming the reproducibility of pEV protein features across independent populations (Fig. 2c).

Importantly, we show that pEV protein cargo differs between individuals with and without coronary artery calcium (CAC) deposits in our early detection of coronary artery disease (EDCAD) cohort (Extended Data Fig. 7a–c, Supplementary Tables 21 and 22). The occurrence and development of calcification is a complex biological process that is regulated by multiple factors, including EVs that are regarded as the nidus for calcification by providing mineral nucleation sites[71]. CAC scoring, a predictor of future cardiovascular events, is determined by computed tomography. We identified 76 upregulated and 127 downregulated proteins (FDR <0.05) in pEVs from individuals with positive CAC scores (Extended Data Fig. 7d,e and Supplementary Tables 21 and 22), with the upregulated proteins significantly enriched in the GO pathway 'Abnormal cardiovascular system physiology' (Extended Data Fig. 7d and Supplementary Table 23). Notably, this includes Cystatin C (CST3)[72], TPM2[73], TPM1[74] and TXNRD2[75], all of which have well-established roles in cardiovascular disease and vascular calcification. Despite these proteomic differences between CAC and non-CAC pEV proteomes, the core set of 182 pEV marker proteins remained highly conserved across both groups (Extended Data Fig. 7f,g and Supplementary Table 24), indicating that the fundamental pEV molecular signature is stable across individuals, potentially even in the context of disease (Fig. 2d). This preservation of core molecular identity in pEVs across CAC and non-CAC individuals, while simultaneously capturing disease-specific molecular shifts in the EDCAD study, reinforces their potential as a well-defined reference for pEV research including disease biomarker discovery. By contrast, NonEV particles and neat plasma proteomes were unable to distinguish CAC from non-CAC individuals, underscoring the power of highly purified pEVs (separated from NonEV particles) in revealing disease-associated changes in the plasma proteome (Supplementary Fig. 9).

To assess the broader biological relevance of pEV protein features beyond plasma-derived EVs, we analysed EV proteomes from non-transformed human fibroblasts and endothelial cells (Extended Data Fig. 8a–d and Supplementary Table 25). Among the 182 pEV protein features, 135 were detected in this dataset, with 43 proteins showing 100% conservation across fibroblast- and endothelial-derived EVs. Conserved proteins included ADAM10, integrins, Rabs, Annexin A5, CD markers (CD9 and CD44) and SDCBP, reinforcing their role as core EV components across different biological sources. Proteins absent from non-transformed cell-derived EVs primarily included immunoglobulins

and complement proteins, supporting their plasma-specific nature or association with the EV protein corona.

Thus, our study defines highly conserved protein features of circulating EVs in human plasma (Table 1 highlights the top 25 EV or NonEV protein features; asterisks indicate surface-accessible proteins).

### The lipidome draft of circulating EVs in humans

Next, we defined the lipidome landscape of circulating EVs. The majority of membrane lipids fall under glycosphingolipids, sphingolipids and sterols (predominantly cholesterol (COH) in mammals). Therefore, to capture this lipid diversity, we used a high-throughput targeted-lipidomics platform interrogating 829 lipids representing 40 lipid classes within these three major groups (Fig. 3a, Supplementary Fig. 10 and Supplementary Table 27). Because the lipidomes of small EVs from cells are not as well defined as their proteome counterpart, we also performed lipidome analysis of in vitro EVs and parental cells using the same platform (Supplementary Fig. 10a,b and Supplementary Table 27). We reasoned that circulating EV lipid features should ideally be enriched in in vitro EVs compared with cells.

Principal component analysis (PCA) revealed that pEVs and NonEVs particles (p100K/pDGS.LD) lipidomes were distinct (Fig. 3b). Highly abundant lipids in both pEVs and in vitro EVs included COH, phosphatidylcholine (PC), phosphatidylethanolamine (PE) and sphingomyelin (SM) (Fig. 3c), lipids that are well-recognized major structural components in eukaryotic cell membranes[76] with housekeeping structural functions. For example, PC forms a planar bilayer with COH, ensuring membrane stability and integrity, while the incorporation of conical PE and SM imposes a curvature stress crucial for membrane budding, fission and fusion. These findings align with previous reports highlighting COH as one of the most abundant EV lipids (40–60% of total lipids)[77].

Over 30% of measured lipid species displayed differential abundance (adjusted $P$ values <0.05, fold change >1.5) between pEVs and NonEVs (p100K/pDGS.LD) (Fig. 3d, Supplementary Fig. 11a,b and Supplementary Table 28). Lipids enriched in pEVs are linked to EV biogenesis[77] and include dihydroceramides (dhCer)[78], trihexosylceramides (Hex3Cer) and dihexosylceramides (Hex2Cer) (Supplementary Fig. 11c). These lipids are similarly enriched in in vitro EVs versus their parental cells (Fig. 3d, Supplementary Fig. 11d,e and Supplementary Table 28). By contrast, lipids enriched in NonEVs include the lipid classes COH ester (CE), triacylglycerol (TG) and coenzyme Q10 (CoQ10)[79], major components of lipoprotein particles. This further supports our EV enrichment pipeline at the lipidomics level.

### The core lipidome of circulating EVs

To identify pEV lipid features, we performed $K$-means clustering of differentially abundant lipids between pEVs versus NonEVs and in vitro EVs versus cells (Fig. 4a and Supplementary Tables 29 and 30), which resulted in four major clusters. Representing pEV lipid features were cluster c4 lipids (52 lipids) enriched in pEVs (compared with NonEVs) and in vitro EVs (compared with cells). By contrast, representing NonEV lipid features were cluster c1 lipids (114 lipids) enriched in NonEVs and cells.

Notably, we observed a class-level coregulation of pEV lipid features (Fig. 4b), supporting previous reports that lipid species from the same pathway correlated/coregulated[80]. For example, lipids from PS and PE class were co-enriched in EVs, which is indicative of their colocalization to external leaflets of EV membranes and coregulating similar cellular processes[81,82]. Another example includes SM and ceramides co-enriched in EVs, which is indicative of SM–ceramide axis-driven EV biogenesis[78,83].

Emphasizing their significance as essential EV lipid components, cluster c4 lipid features (Extended Data Fig. 9) were linked to lipid ontology enrichment analysis (LION) terms such as bilayer membrane and plasma membrane (Fig. 4c), but more specifically with EV features such as glycerophosphatidylserine (known to decorate the outer

**Table 1 | List of EV and NonEV protein features**

| | Gene symbol | Protein name | GO (CC\|BP) | Reference |
|---|---|---|---|---|
| **EV protein features** | ADAM10* | Disintegrin and metalloproteinase domain-containing protein 10 | Cell membrane, endomembrane \| Adherens junction organization | 112 |
| | ARPC4* | Actin-related protein 2/3 complex subunit 4 | Cell projection, cytoskeleton \| Actin filament polymerization | 113 |
| | CD47* | Leukocyte surface antigen CD47 | Cell membrane \| Integrin-mediated signalling pathway | 114 |
| | F11R* | Junctional adhesion molecule A | Cell membrane \| Actomyosin structure organization | 115 |
| | VCP* | Transitional endoplasmic reticulum ATPase | Cytoplasm, ER \| ER-to-Golgi vesicle-mediated transport | 116 |
| | STXBP2 | Syntaxin-binding protein 2 | Cytosol \| Vesicle docking involved in exocytosis | 117 |
| | RAC1* | Ras-related C3 botulinum toxin substrate 1 | Cell membrane (lipid anchor) \| Actin cytoskeleton organization | 118 |
| | ACTR2* | Actin-related protein 2 | Cell projection, cytoskeleton \| Arp2/3 complex-mediated actin nucleation | 119 |
| | EHD1* | EH domain-containing protein 1 | Early/Recycling endosome \| Endocytic recycling | 120 |
| | FLOT1* | Flotillin-1 | Cell membrane, endosome \| Plasma membrane raft assembly | 121 |
| | CAPN1* | Calpain-1 catalytic subunit | Cell membrane, cytoplasm \| Regulation of catalytic activity | 122 |
| | ITGA2* | Integrin alpha-2 | Membrane \| Cell adhesion | 123 |
| | NAPA* | Alpha-soluble NSF attachment protein | Peripheral membrane protein \| Apical protein localization | 124 |
| | RAB7A* | Ras-related protein Rab-7a | Lipid anchor, late endosome \| Endosome to plasma membrane transport | 125 |
| | FLOT2 | Flotillin-2 | Cell membrane, endosome \| Plasma membrane raft | 126 |
| | ARPC3* | Actin-related protein 2/3 complex subunit 3 | Cell projection, cytoskeleton \| Actin polymerization | 127 |
| | ARPC2* | Actin-related protein 2/3 complex subunit 2 | Cell junction, cell projection \| Actin filament polymerization | 128 |
| | GNA13 | Guanine nucleotide-binding protein subunit alpha-13 | Membrane (lipid anchor) \| Rho protein signal transduction | 129 |
| | LYN | Tyrosine-protein kinase Lyn | Cell membrane \| Cell adhesion mediated by integrin | 130 |
| | CYFIP1 | Cytoplasmic FMR1-interacting protein 1 | Cell projection \| Actin filament polymerization | 131 |
| | ACTR3 | Actin-related protein 3 | Cell projection, cytoskeleton \| Cell motility | 119 |
| | PRKAR1A | cAMP-dependent protein kinase type I-alpha regulatory subunit | Cell membrane \| G protein-coupled receptor signalling | 132 |
| | SNAP23* | Synaptosomal-associated protein 23 | Cell membrane (lipid-anchor) \| Membrane fusion | 133 |
| | PDCD6IP | Programmed cell death 6-interacting protein | Cytoplasm, cytoskeleton \| Multivesicular body assembly | 134 |
| | ANXA11 | Annexin A11 | Cytoplasm, cytoskeleton \| Phagocytosis | 135 |
| **NonEV protein features** | C6 | Complement component C6 | Secreted \| Complement activation | 136 |
| | F10 | Coagulation factor X | Secreted \| Blood coagulation | 137 |
| | C7 | Complement component C7 | Secreted \| Complement activation | 138 |
| | CPB2 | Carboxypeptidase B2 | Secreted \| Blood coagulation | 136 |
| | SERPING1 | Plasma protease C1 inhibitor | Secreted \| Complement activation | 139 |
| | F13B | Coagulation factor XIII B chain | Secreted \| Fibrin clot formation | 140 |
| | APOA4 | Apolipoprotein A-IV | Secreted \| Lipid transport | 141 |
| | F2 | Prothrombin | Secreted \| Blood coagulation | 136 |
| | TF* | Serotransferrin | Secreted \| Iron ion transport | 142 |
| | ORM1* | Alpha-1-acid glycoprotein 1 | Secreted \| Acute-phase response | 136 |
| | APOB* | Apolipoprotein B-100 | Secreted \| Lipoprotein transport | 136 |
| | SERPINC1 | Antithrombin-III | Secreted \| Blood coagulation | 143 |
| | CFI | Complement factor I | Secreted \| Complement activation | 144 |
| | CLEC3B | Tetranectin | Secreted \| Bone mineralization | 145 |
| | AFM | Afamin | Secreted \| Vitamin transport | 146 |
| | AZGP1* | Zinc-alpha-2-glycoprotein | Secreted \| Immune response | 147 |
| | GC* | Vitamin D-binding protein | Secreted \| Vitamin transport | 148 |
| | TTR* | Transthyretin | Secreted \| Purine nucleobase metabolic process | 148 |
| | SERPINA3* | Alpha-1-antichymotrypsin | Secreted \| Lipid metabolic process | 149 |
| | AHSG* | Alpha-2-HS-glycoprotein | Secreted \| Bone mineralization | 145 |
| | HPX | Haemopexin | Secreted \| Haem metabolic process | 150 |
| | A1BG | Alpha-1B-glycoprotein | Secreted | 145 |
| | ITIH3 | Inter-alpha-trypsin inhibitor heavy chain H3 | Secreted \| Hyaluronan metabolic process | 148 |
| | RBP4 | Retinol-binding protein 4 | Secreted \| Retinol transport | 151 |
| | SERPINF1 | Pigment epithelium-derived factor | Secreted \| Cellular response to retinoic acid | 152 |

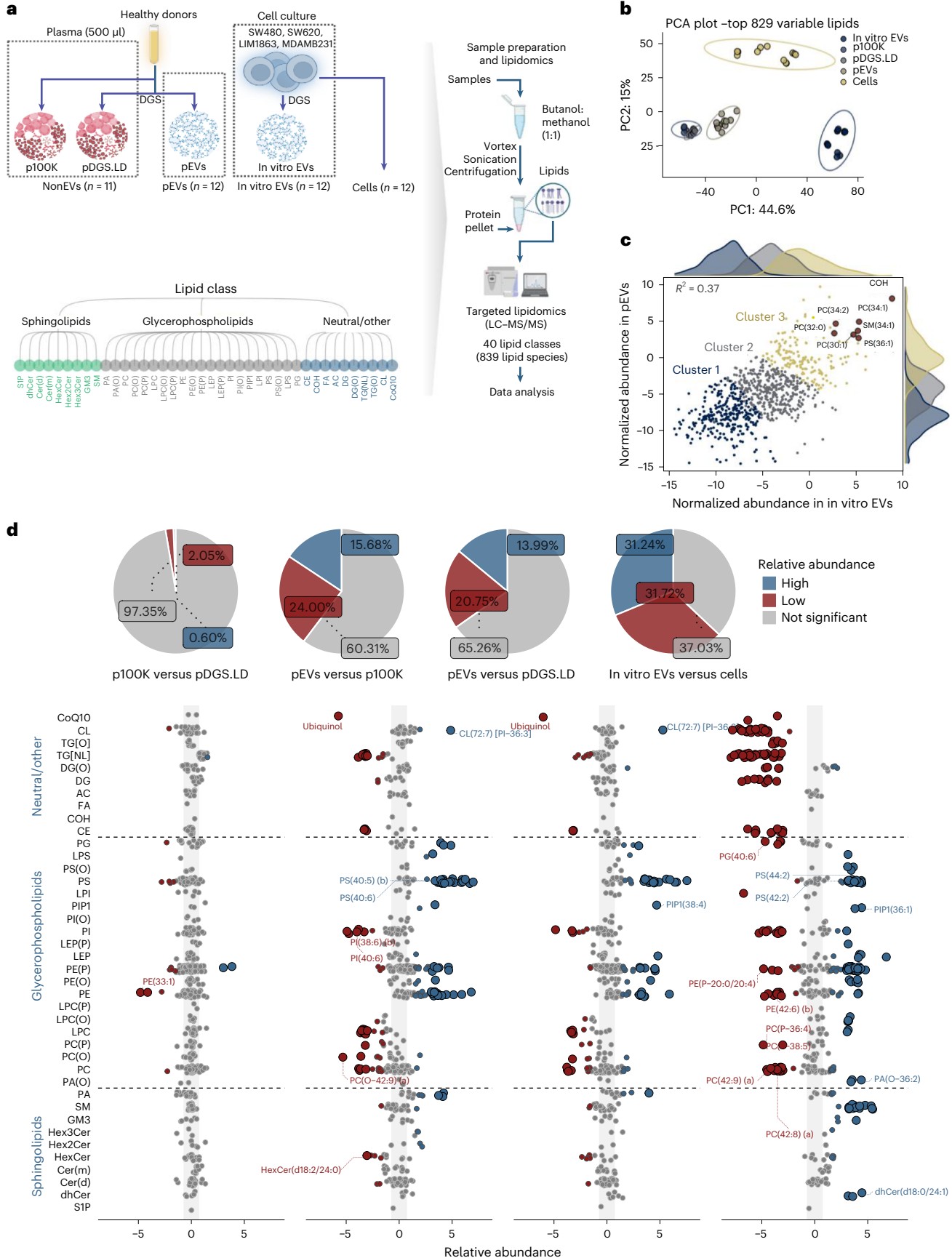

**Fig. 3 | Lipidome landscape of circulating EVs. a**, Workflow for lipidomic analysis, interrogating 829 lipids representing 40 lipid classes within three major groups. **b**, PCA of lipidome data showing group clustering and separation of lipidomes (pEVs; $n = 12$, p100K; $n = 6$, pDGS.LD; $n = 5$ (independent plasma samples), in vitro EVs; $n = 12$ (biological replicates), cells; $n = 12$ (3 biological replicates per cell line)). PC, principal component. **c**, Scatter plot depicting the abundance of 829 lipids in pEVs and in vitro EVs with lipids belonging to cluster 3 displaying high abundance in both pEVs and in vitro EVs. **d**, Scatter plot showing the relative abundance of lipids in pEVs versus p100K/pDGS.LD and in vitro EVs versus cell lipidome datasets. Blue circles (lipid markers) represent lipids with

significantly greater abundance (fold change >1.5 and $P < 0.05$). Red circles (exclusion lipids) represent lipids with significantly lower abundance (fold change <−1.5 and $P < 0.05$). Grey circles represent lipids that do not meet the above criteria. Pie chart depicts number of lipids (%) with differential abundance between the datasets. Differential lipid abundance was assessed using a limma-based linear-model framework with empirical-Bayes moderation applied to vsn-normalized intensities. Two-sided moderated $t$-tests were used for pairwise contrasts. Multiple-testing-adjusted $q$ values (Benjamini−Hochberg FDR) are reported in Supplementary Table 28.

---

membrane/leaflet of EVs) and low lateral diffusion (a distinct physical property of lipid rafts[84] that are hubs for EV formation). Conversely, cluster c1 NonEV lipid features (Extended Data Fig. 9) were associated with lipid storage and droplets, characteristic of lipoprotein particles.

To validate the association of cluster c4 lipids with circulating EVs, and cluster c1 with NonEV particles in plasma, we subjected 12 fractions of plasma DGS ($n = 3$ plasma samples) to the same targeted lipidomics platform (Fig. 4d,e, Supplementary Fig. 12a,b and Supplementary Tables 30 and 31). The normalized abundance of lipids across the fractions was plotted ($z$-scored within each fraction) (Fig. 4f). Indeed, cluster c4 lipids were enriched in DGS fractions 6 and 7, where pEVs resolve. By contrast, cluster c1 lipids were enriched in DGS fractions 1−4, where the majority of NonEV particles resolve, reinforcing the specificity of cluster c4 lipids for circulating EVs in plasma.

To further assess the conservation of pEV lipid features, we performed lipidomic profiling of pEVs and NonEVs isolated from two sets of plasma from AusDiab study ($n = 10$ plasma samples for validation set 1 and $n = 12$ plasma samples for validation set 2) and non-transformed (primary human fibroblasts and endothelial cells) cell-derived EVs (Extended Data Fig. 10a,b, Supplementary Table 32 and Supplementary Fig. 13). In this validation lipidomic workflow, we requantified 12 out of 52 pEV lipid features and 48 out of 114 NonEV lipid features and report their differential abundance analysis (Supplementary Tables 32 and 33 and Supplementary Fig. 14). Comparison of fold changes between these lipid features across the discovery and validation datasets showed a strong correlation (Fig. 4g and Supplementary Table 34, Pearson $r = 0.901$, $P < 2 \times 10^{-16}$ for validation set 1, and Pearson $r = 0.878$, $P < 2 \times 10^{-16}$ for validation set 2), confirming the reproducibility of pEV lipid features. The relative abundance of these features was also conserved in EVs from primary human fibroblasts and endothelial cells (Extended Data Fig. 10b and Supplementary Fig. 14). These findings highlight the robustness of our identified EV lipid markers across independent populations and multiple EV sources.

Thus, our study defines highly conserved lipid features of circulating EVs in human plasma (Table 2; asterisks indicate the top 25 enriched lipids in either pEVs or NonEVs, based on relative fold change in abundance).

### Biological protein and lipid markers for circulating EVs

We next investigated the ability of the EV and NonEV protein feature panel to distinguish between pEVs versus NonEV particles using

machine learning (naive Bayes algorithm). For this, pEV and NonEV proteomes were evenly partitioned on the basis of sample type into a training set (70% of the samples) and a validation set (remaining 30% of samples). By the bootstrapping resampling method using 25 resampling iterations, the model achieved absolute accuracy (Supplementary Fig. 15a). Our model also achieved excellent accuracy (97%) in EV particle identification in an independent test set comprising of pEVs and NonEVs proteomes from additional 16 plasma samples (Supplementary Fig. 15a). Moreover, a panel comprising 151 surface-accessible EV protein features was also able to distinguish between EV and NonEV particles with 100% accuracy (Supplementary Fig. 15b).

To facilitate routine implementation and translational feasibility, we applied recursive feature elimination (RFE) to systematically reduce dimensionality and identify a robust minimal signature with classification accuracy. Among the 182 EV protein features, the RFE-based random forest algorithm identified ADAM10 as a prominent feature contributing to model performance (Supplementary Table 35). Moreover, ADAM10 protein showed exclusive and absolute quantification in pEVs as well as in vitro EVs, compared with NonEVs (Fig. 5a). Moreover, within each EV proteome the abundance of ADAM10 was up to 7.5-fold higher compared with the median intensity (Fig. 5b and Supplementary Table 36); this relative abundance of ADAM10 within each proteome alone can distinguish between pEVs versus NonEV particles (Supplementary Fig. 15c).

Similarly, using a neural network algorithm ('nnet') for machine learning, a panel comprising EV and NonEV lipid features was also able to distinguish between EVs (pEVs or in vitro EV lipidomes) and NonEVs (p100K, pDGS.LD or cell lipidomes) (Supplementary Fig. 16a). Moreover, this model achieved 97% accuracy in classifying EV particles across DGS plasma fractions (Supplementary Fig. 16b).

To enhance translational feasibility, we applied RFE to reduce dimensionality and identify a minimal lipid signature with high classification accuracy. This approach identified PS(36:1) (EV lipid feature) and CE(18:0) as prominent lipid features (Supplementary Table 35), highlighting their marker potential.

Indeed, PS(36:1) lipid is one of the most abundant lipids present in both pEVs and in vitro EVs (Figs. 3c and 5c), making them amenable to robust and reliable measurements in diverse analytic tools. Moreover, PS(36:1) and CE(18:0) lipid abundances, and importantly their relative abundances, concur with EV protein CD63 and lipoprotein APOB100

---

**Fig. 4 | Conserved lipid features of circulating EVs in humans. a**, Heatmap depicting $K$-means clustering of differentially abundant lipids from **d. b**, Network map of lipids grouped based on lipid classes. Blue circles represent EV-associated lipids clusters c2, c3 and c4, whereas red circles represent NonEV-associated lipid cluster c1. Grey circles are lipids with similar abundance between EVs and NonEVs. **c**, LION Ontologies enriched in EV lipid features (cluster c4 lipids) or NonEV lipid features (cluster c1 lipids). The circle size indicates the number of lipids in EVs or NonEVs involved in each term. **d**, DGS of plasma. The 12 fractions were subjected to lipidomics analysis. **e**, PCA of lipidomes of 12 fractions ($n = 3$, independent plasma samples). **f**, Box plots depict abundance of cluster 1−4 lipids for DGS fractions ($n = 3$, DGS experiments using independent plasma samples). The $Y$ axis represents $Z$-scored abundance (MS-based abundance for each lipid · mean abundance)/standard deviation ($z$-score normalization). Grey lines mark either fractions 6−7 (corresponding to pEV fractions) or fractions 1−3

(corresponding to pDGS.LD fractions). Lipids from cluster c4 and cluster c1 are depicted. The boxplot displays the median (centre line), 25th−75th percentiles (box) and the minimum and maximum values within 1.5× the interquartile range (whiskers); individual data points are overlaid. **g**, Scatter plot of fold change (FDR <0.05) EV lipid features and NonEV lipid features in the discovery set and two validation sets (AusDiab set). The rest of the lipids are indicated with grey points. Fold-change correlation between discovery and validation sets. Features were first screened by one-way ANOVA (two-sided) on vsn-normalized intensities with Benjamini−Hochberg FDR across features; significant features (FDR <0.05) were subjected to Tukey's HSD test with Benjamini−Hochberg adjustment applied within each feature. Points highlight features significant as EV (blue) or NonEV (red) in both sets. Grey line: least-squares fit with 95% CI band. $P = 2.673044 \times 10^{-22}$ (validation set 1) and $P = 6.562873 \times 10^{-20}$ (validation set 2).

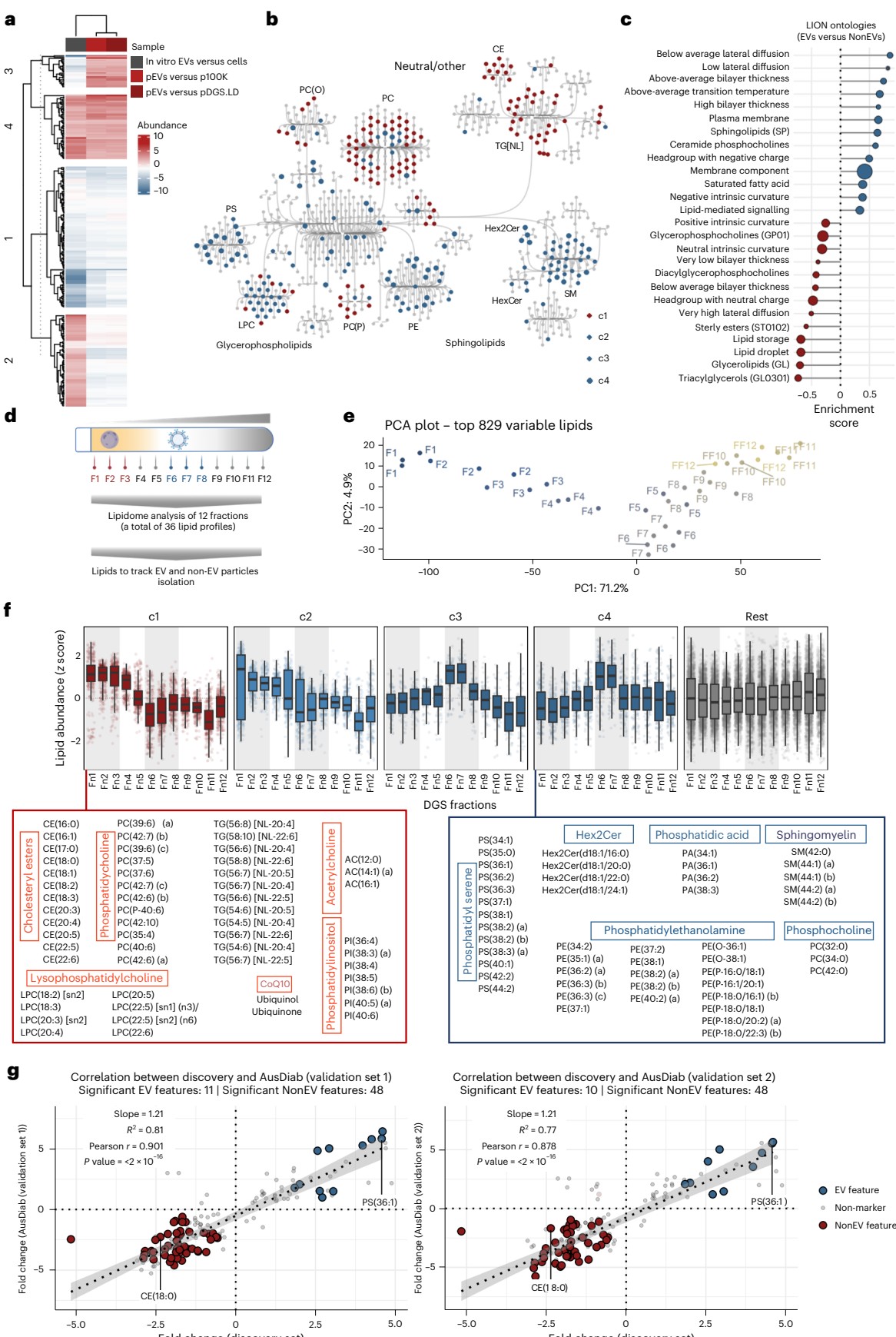

**Table 2 | List of EV and NonEV lipid features**

| | Class | Lipids | Number | Description | References |
|---|---|---|---|---|---|
| **EV lipid features** | PA | PA(34:1)*, PA(36:1)*, PA(36:2)*, PA(38:3)* | 4 | Phosphatidic acid (PA): involved in cell signalling, membrane trafficking and lipid metabolism. | 153,154 |
| | PC | PC(32:0), PC(34:0), PC(42:0) | 4 | PC: major component of cell membranes, critical for membrane structure and function. | 155,156 |
| | PE | PE(35:1) (a)*, PE(36:2) (a)*, PE(38:2) (a)*, PE(38:2) (b)*, PE(40:2) (a)* | 11 | PE: another major membrane phospholipid, essential for membrane integrity and cellular processes. | 157 |
| | PE(O) | PE(O-36:1), PE(O-38:1) | 2 | Alkylphosphatidylethanolamine (PE(O)): ether lipid class that is synthesized in a peroxisomal-dependent pathway. | 158 |
| | PE(P) | PE(P-18:0/16:1) (b)*, PE(P-18:0/20:2) (a)*, PE(P-18:0/22:3) (b)* | 6 | PE plasmalogen (PE(P)): also involved in membrane structure and antioxidant defence. | 157 |
| | PS | PS(34:1)*, PS(36:1)*, PS(36:2)*, PS(36:3)*, PS(37:1)*, PS(38:1)*, PS(38:2) (a)*, PS(38:2) (b)*, PS(38:3) (a)*, PS(40:1)* | 13 | Phosphatidylserine (PS): plays roles in cell signalling, apoptosis and blood clotting. Enriched in platelets and EVs. | 159,160 |
| | Hex2Cer | Hex2Cer(d18:1/20:0)*, Hex2Cer(d18:1/22:0)*, Hex2Cer(d18:1/24:1)* | 4 | Hex2Cer: a glycosphingolipid involved in cell signalling and membrane dynamics. | 161 |
| | SM | SM(42:0), SM(44:1) (a), SM(44:2) (a), SM(44:2) (b) | 5 | SM: another glycosphingolipid, important for membrane structure and stability. | 162,163 |
| **NonEV lipid features** | AC | AC(12:0), AC(14:1) (a)*, AC(16:1) | 3 | Acylcarnitine (AC): involved in fatty acid transport and mitochondrial energy metabolism. | 164 |
| | CE | CE(18:0)*, CE(18:2)*, CE(18:3)*, CE(20:3)*, CE(20:4)*, CE(20:5)*, CE(22:5)*, CE(22:6)* | 12 | Cholesteryl ester (CE): form of COH storage, found in lipid droplets and lipoprotein particles. | 165 |
| | LPC | LPC(18:2) [sn2]*, LPC(18:3)*, LPC(20:4)*, LPC(20:5)* | 7 | Lysophosphatidylcholine (LPC): implicated in inflammation, endothelial dysfunction and lipid metabolism. | 166 |
| | PC | PC(37:5)*, PC(37:6)*, PC(39:6) (a)*, PC(39:6) (c)*, PC(42:7) (b)*, PC(42:7) (c)* | 43 | PC: major component of cell membranes, essential for membrane structure and function. | 167 |
| | PC(O) | PC(O-36:4), PC(O-38:4) (b), PC(O-38:6) (b) | 6 | Alkylphosphatidylcholnie (PC(O)): ether lipid class that is synthesized in a peroxisomal-dependent pathway. | 168,169 |
| | PC(P) | PC(P-36:4), PC(P-38:3), PC(P-38:6), PC(P-42:5) | 9 | PC plasmalogen (PC(P)): also involved in membrane structure and antioxidant defence. | 170,171 |
| | PI | PI(38:3) (a)*, PI(38:6) (b)*, PI(40:5) (a), PI(40:6)* | 7 | Phosphatidylinositol (PI): important for cell signalling, membrane trafficking, and cytoskeletal dynamics. | 172 |
| | TG[NL] | TG(52:3) [NL-18:2], TG(56:7) [NL-20:4], TG(56:8) [NL-20:4]*, TG(58:10) [NL-22:6]* | 24 | Triglycerides (TG[NL]): main storage form of fats in the body, core component of lipid droplets, serving as an energy reservoir. | 173 |
| | CoQ10 | Ubiquinol*, Ubiquinone | 2 | CoQ10: essential for cellular energy production, antioxidant activity and mitochondrial function. | 174 |

resolution on plasma DGS fractions (Fig. 5d,e). Our machine learning ('nnet') model, based on PS(36:1) and CE(18:0) relative abundance within each lipidome serving as a feature (Supplementary Table 37), can distinguish EVs (pEVs or in vitro EV, DGS fraction 6–7 lipidomes) versus NonEVs (p100K, pDGS.LD or DGS fraction 1–5 lipidomes) with absolute accuracy (Fig. 5f). Because EV and NonEV enrichments along 12 fractions represent a continuum, which can be inferred from PS(36:1) and CE(18:0) relative abundances, our data suggest that these lipids can serve as a marker for assessment of EV purity. Importantly, quantitative measurement would be possible in subsequent studies owing to the availability of isotope-labelled internal standards for these two species, which assist in translation across different laboratories and instrumental setups.

We next investigated whether PS and ADAM10 constitute protein and lipid biological marker componentry on circulating EVs. For this, we used a previously reported[85] strategy of using an extracellular region of the T cell immunoglobulin domain and mucin domain-containing protein 4 (Tim4) immobilized on magnetic beads to directly capture (in the presence of calcium ions) PS+ EVs from plasma (n = 23), which was subsequently released (by adding EDTA) and subjected to MS proteomics (Fig. 5g and Supplementary Table 38). Compared with mock capture (beads alone) or unprocessed plasma, PS+ EV proteomes displayed 100% quantification of ADAM10 across all 23 samples (Fig. 5g).

Additional EV protein features (for example, FLOT2 and RAB7A) were also conserved in these proteomes, which support their EV identity.

Furthermore, single-vesicle analysis using Cytek Aurora flow cytometry confirmed the presence of ADAM10 on a subset of pEVs (Fig. 5h and Supplementary Fig. 17), with ~40% of pEVs exhibiting ADAM10+ signals, comparable to PS signal detected using Annexin V staining (Fig. 5h, Supplementary Fig. 17c and Supplementary Table 39). Sodium dodecyl sulfate (SDS) detergent treatment showed a strong reduction in fluorescence signal intensity and count, suggesting their EV origin[86] (Supplementary Figure 17b,c). These findings suggest that, while ADAM10 is a component of pEVs across individuals, it is not universally present across all vesicles, consistent with a previous report[87].

In our validation set, PS(36:1) lipid was significantly enriched in pEVs and in vitro EVs, reinforcing its role as a conserved pEV lipid marker (Extended Data Fig. 10c). By contrast, CE(18:0) lipid was enriched in NonEVs and neat plasma (Extended Data Fig. 10c). These findings confirm that pEV protein and lipid markers are conserved beyond plasma-derived EVs, reinforcing their biological significance and potential as robust EV classification markers.

Thus, our data show that ADAM10 and the ratio combination of PS(36:1) and CE(18:0) serve as highly conserved and reliable biological markers for EVs in human plasma.

### R/Shiny web tool for EV proteome and lipidome data

Lastly, to facilitate easy access to our data and enhance reuse, we have developed an open-source R/Shiny web tool (evmap.shinyapps.io/evmap/). This tool allows users to quickly interrogate our proteome (Fig. 6a,b and Supplementary Fig. 18) and lipidome datasets (Fig. 6a,b and Supplementary Fig. 19) for their molecule(s) of interest. This tool also allows the assessment of feature conservation in published studies, analysis of surface accessibility of EV proteins, construction of network analyses, and retrieval of GO and KEGG pathways for selected protein features. Lipid features can also be quickly interrogated for their abundance in EVs versus NonEV particles and their distribution in plasma DGS lipidome fractions. We anticipate that this tool will enable broad utilization of our data and will serve as a valuable repository for the broader EV community.

## Discussion

In our study, we integrated multi-omics investigation to systematically resolve the core protein and lipid componentry of circulating EVs in humans (Fig. 6a). Our discovery includes a conserved set of 182 proteins and 52 lipids intrinsic to circulating EVs, and a panel of 29 proteins and 114 lipids that are NonEV features in plasma, which together serve as biological markers for EV research applicable to human samples. As a resource, these extensive protein and lipid landscapes, which can be easily accessed with the Shiny web tool (Fig. 6b), will be instrumental to the EV community in advancing a clearer understanding of circulating EV biology. This includes the survey of circulating EV surfaceome, comprising 151 EV protein features, that could be exploited for antibody-based capture of circulating EVs based on their surface protein expression(s). In addition to this resource, we identified a minimal set of biomarkers—ADAM10 (protein), PS(36:1) and CE(18:0) (lipids)—with strong discriminatory power between EVs and NonEVs. This reduced marker set is compatible with targeted, scalable assays such as enzyme-linked immunosorbent assay or targeted MS, making it highly practical for clinical translation. Furthermore, the ranked feature list generated by our machine learning framework provides a valuable resource for the research community, enabling prioritization of alternative markers based on available reagents or disease-specific applications.

The highly conserved protein ADAM10 on circulating EVs is also included in this EV surfaceome. The role of ADAM10 as a conserved and robust EV marker is further supported by various studies reporting its expression in circulating EVs from human plasma, other biofluids and EVs released by diverse tissue and cell types, using a range of biochemical analyses and EV isolation strategies. Moreover, ADAM10 was also highly conserved in different circulating EV subpopulations (based on surface expression of tetraspanins, CD9, CD63 and CD81[70]). The surface localization of ADAM10 is also corroborated in previous reports[88]. Conversely, the EV lipid feature PS[15] has already been leveraged for capturing circulating EVs using TIM4-based magnetic beads. The selective presence of ADAM10 (and PS lipids) in only a subset of EVs highlights the non-random nature of EV cargo packaging, supporting models of regulated biogenesis pathways such as endosomal sorting complexes required for transport (ESCRT)-dependent sorting[89]. Evidence that ADAM10 retains proteolytic activity in EVs[90] suggests that these vesicles may serve as active mediators of extracellular remodelling and intercellular communication, expanding their functional relevance beyond passive biomarker carriers. The differential association of ADAM10 with specific EV subtypes, and its absence from some canonical small EV populations[87], underscores the existence of biophysically and functionally distinct EV in circulation. Together, these findings support a growing consensus that EV heterogeneity is biologically regulated rather than stochastic, with specific protein profiles reflecting distinct functional roles in physiological and pathological contexts[87,89,90].

Our data also provide 'high-confidence' molecular leads for studying EV biology in multimodal organisms, particularly targeting EV biogenesis to curb EV-driven pathogenesis in complex organisms such as the mouse[83], a model that remains largely unexplored. For instance, EV protein feature STEAP3 (also known as TSAP6) has been shown to regulate EV biogenesis in mice[91]. A holistic analysis of the core protein and lipid componentry can also provide insights into biological pathways co-involved in proteins and lipids; one such biology includes the potential involvement of lipid raft biology in human EVs. For instance, EV features such as flotillins[92], SM and PS, along with COH (the most abundant lipids in EVs), are core components of lipid raft domains[84]. Assembly of flotillins in SM/ceramide[84]-rich microdomains induces curvature stress[93] to regulate EV biogenesis and cargo selection[78,94] in ESCRT-[95] and syntenin/syndecan-independent mechanisms[96]. Moreover, proteins that constitute the cortical actin—identified as an enriched KEGG term among EV core proteins—help maintain and remodel these domains[84]. Accumulating evidence suggests that lipid rafts serve as a nidus for EV biogenesis and are present in EVs as functional units[94]. Moreover, several bioactive cargo quantified in pEVs—such as TGFB1[66], the RNA-binding protein HNRNPK[62], integrins[67] and SRC kinases[65]—are closely linked to lipid raft assembly and trafficking[97–100]. Importantly, classical lipid-raft-modulating proteins such as CAV1 and CAVIN1—known to regulate membrane microdomain composition and influence the EV proteome[101,102]—regulate microRNA cargo loading into EVs by modulating HNRNPK within the membrane raft[62]. Elevated levels of EV-associated HNRNPK have also been detected in body fluids from patients with metastatic prostate and colorectal cancers, underscoring its clinical relevance. Ubiquitous identification of these features in pEVs highlights similar EV biogenesis and cargo sorting mechanisms conserved for circulating EVs in humans.

A major caveat in our study is that it primarily focuses on bulk preparation of small EVs present in human plasma. Cells release

**Fig. 5 | ADAM10 protein and phosphatidylserine lipid are biological markers for circulating EVs in humans. a**, Heatmap depicting pEV protein features that are uniquely identified in pEVs compared with NonEVs and their conservation, in particular for ADAM10, in previously published circulating EV proteomes[15,69] (A–C) and EV subtypes (CD81+, CD63+ and CD9+ EVs) in plasma (D–F)[70], and in our TMT-based isobaric multiplexing of pEVs (n = 4, independent plasma samples) and in vitro EVs proteomes (n = 6, biological replicates) as well as label-free proteomes of small EVs from 4 cell lines (n = 3, biological replicates, per cell line) and pEV proteome (n = 38, independent plasma samples) (I–L). **b**, The ratio of ADAM10 intensity and corresponding sample median intensity for each proteome, depicted as a boxplot. The boxplot shows the median (centre line), 25th–75th percentiles (box) and minima and maxima within 1.5× interquartile range (whiskers); individual data points are overlaid. **c**, Box plot depicting normalized intensities (abundance log₂-tranformed) and raw intensities of PS(36:1) and CE(18:0) in indicated lipidome datasets (NonEV n = 11, pEV n = 12; independent plasma samples). Boxplot shows the median (centre line), 25th–75th percentiles (box) and minima and maxima within 1.5× interquartile range (whiskers); individual data points are overlaid. **d**, The abundance of PS(36:1) and CE(18:0) in lipidome datasets across 12 DGS fractions (n = 3, independent plasma samples). Grey points represent the rest of the quantified lipids. **e**, Top: the relative abundance of PS(36:1) and CE(18:0) in lipidome datasets across 12 DGS fractions (n = 3, independent plasma samples). Middle: the median of lipidome datasets across 12 DGS fractions. Bottom: western blot analysis of 12 fractions for indicated proteins. Boxplot shows the median (centre line), 25th–75th percentiles (box) and minima and maxima within 1.5× interquartile range (whiskers); individual data points are overlaid. **f**, Confusion matrix using the neural network algorithm ('nnet') classifier of the training set (70%) and validation set (30%) using the relative abundance of PS(36:1) versus CE(18:0). **g**, The workflow for capturing PS+ EVs from human plasma using TIM4–magnetic bead conjugate. Captured EVs (n = 23, independent plasma samples), along with mock capture using beads alone (n = 3, independent plasma samples) or unprocessed plasma (n = 3, independent plasma samples), were subjected to proteomics analysis. The heatmap depicts absolute and exclusive quantification of ADAM10 protein in captured EVs. **h**, Scatter plot showing the detection of ADAM10-positive and PS-positive pEVs, which was sensitive to SDS detergent (0.5%) solubilization.

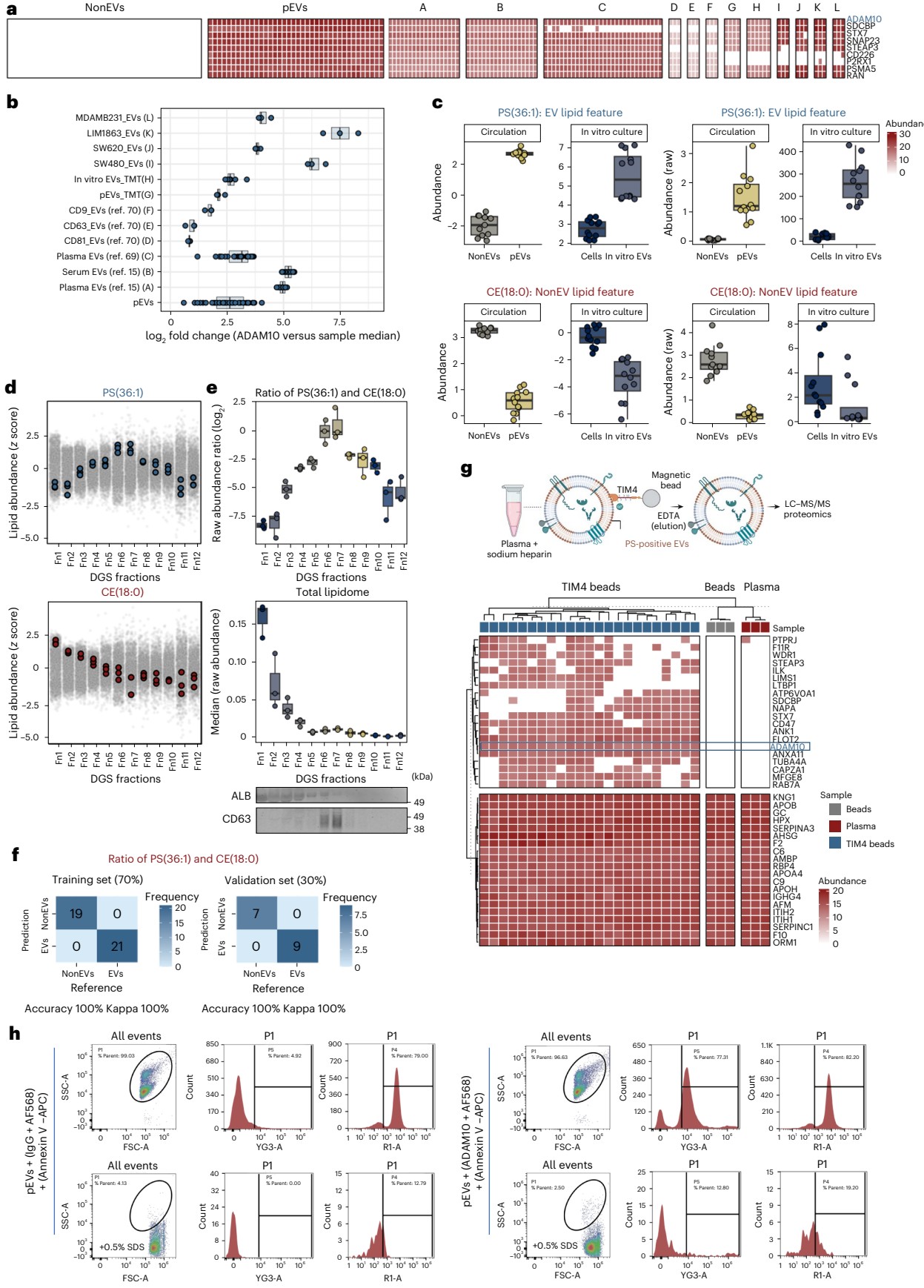

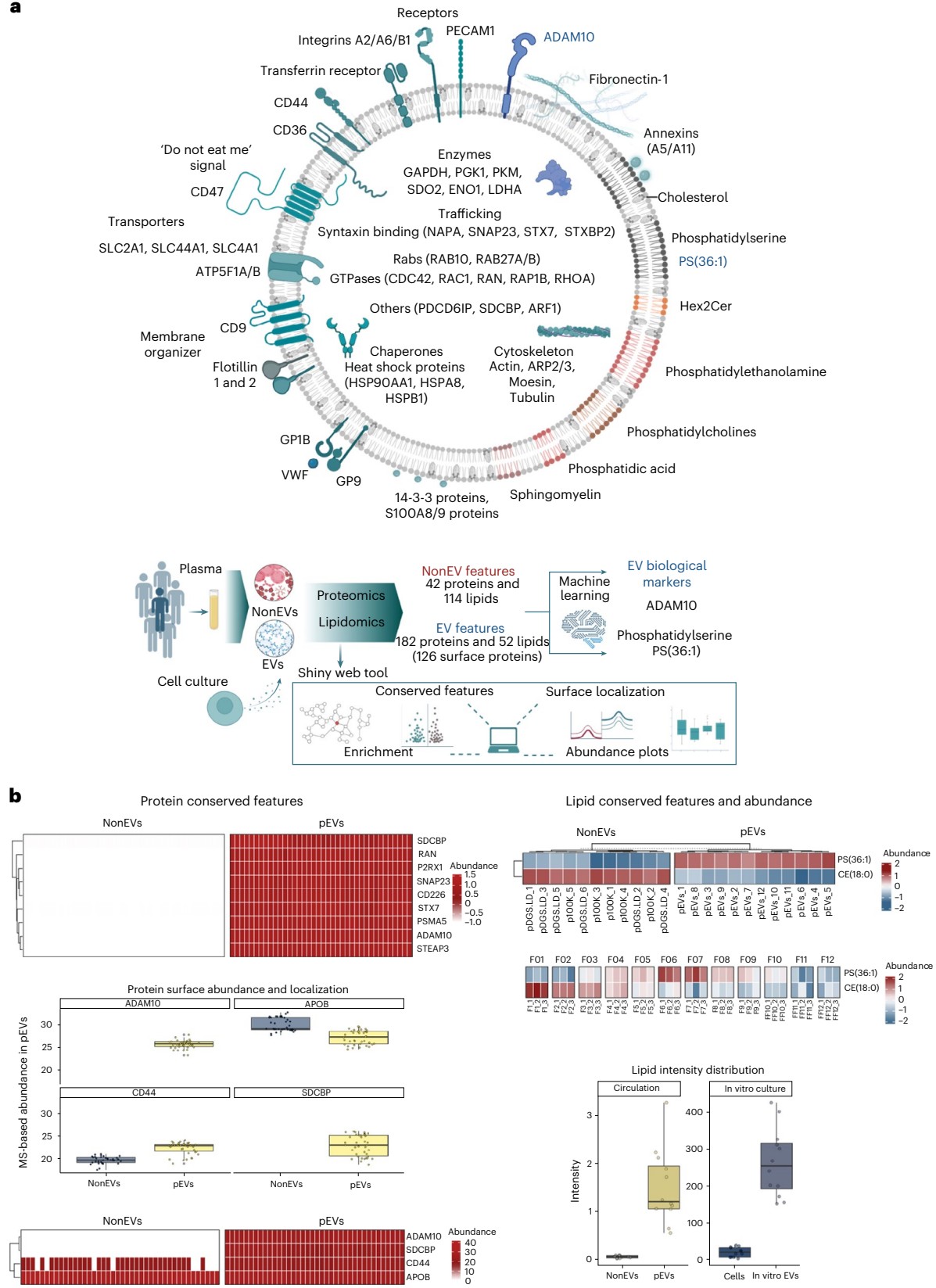

**Fig. 6 | Hallmark molecular features of circulating EVs. a**, Resolving the protein and lipid componentry intrinsic to circulating EVs. These features are also conserved in EVs released by cultured cells. We submit EV features, namely ADAM10 and phosphatidyl serine PS(36:1), as biological markers for confident EV identification and potentially purity assessment. **b**, The open-source Shiny web tool will facilitate interrogation of protein and lipid features in pEVs and NonEVs

and serve as a valuable repository to the research community for a clearer understanding of circulating EV biology. Boxplot show the median (centre line), 25th–75th percentiles (box) and minima and maxima within 1.5× interquartile range (whiskers); individual data points are overlaid. Panel **a** was created with BioRender.com.

heterogeneous EV populations[36,45,103,104] that display functional diversity[105]. They can be categorized on the basis of their origin[45,106,107], density[45,50,104], biochemical composition[31,50] and size[47,50]. Because current EV isolation strategies cannot purify a specific subset of EVs to homogeneity, operational terms such as small EVs (30–150 nm) and large EVs (100–1,000 nm) are encouraged where evidence for subcellular origin cannot be ascertained. In our study, small EVs potentially include both exosomes (endosomal-derived) and microvesicles (shed from plasma membrane) evident from our EV protein markers that include markers of both EV subtype (TSG101 and ANXA2). Thus, our EV markers are applicable in a broad sense to small EVs; their specificity in exosomes versus microvesicles can potentially provide greater resolution in human EV biology and thus warrants future investigation. Moreover, EV heterogeneity is reported in circulating EVs too based on surface expression of tetraspanins, CD9, CD63 and CD81[70]. Nonetheless, our EV protein features show high conservation in all three EV subpopulations[70]. While bulk EV study provides a broader understanding of the complex interactions and processes involving EVs within the body that specific EV subtypes might not fully capture, it is essential to study EV subpopulation-specific core features in humans, as this avenue holds great potential for advancing precise diagnostics and therapeutics. Nevertheless, bulk analyses can still offer insights into coordinated molecular patterns and biologically conserved pathways. For instance, enrichment patterns observed in our study (Extended Data Fig. 6) reflects functionally associated protein modules enriched in the pEV pool. While these associations do not imply direct molecular interactions or spatial colocalization within individual EVs, they highlight coordinated expression patterns that are recurrent across EV subtypes and cell types. Importantly, we validated that many of these molecular features are enriched in EVs derived from multiple cell types—including cancer cells and primary fibroblasts—as well as across CD9-, CD63- and CD81-positive pEVs, suggesting they are not artefacts of averaging but representative of fundamental EV-associated signatures. Our findings align with prior studies demonstrating that such molecules are actively enriched in EVs relative to their cells of origin and conserved across cell-derived EVs[30]. Thus, while single-vesicle resolution would offer even deeper insight, bulk EV analysis can still uncover biologically meaningful and source-representative features that contribute to systemic intercellular communication. In regard to source attribution, protein signatures associated with diverse cell types were represented in pEVs, including endothelial cells, fibroblasts, hepatocytes, cardiomyocytes, kidney cells and haematopoietic cells (such as platelets), corroborating previous reports[15,54,55,108,109]. In healthy individuals, circulating EVs reportedly arise from haematopoietic cells (such as platelets, erythrocytes and leukocytes) and endothelial cells[54,55]. While definitive source attribution of pEVs remains technically challenging due to the lack of universally exclusive surface markers, recent computational deconvolution algorithms offer promising solutions to estimate the relative contribution of different cell types[108,109]. Complementary proteomic studies have also identified tissue-specific EV-associated proteins, providing valuable leads for understanding EV origin and function in systemic circulation[15]. In parallel, high-sensitivity techniques such as single-vesicle flow cytometry and antibody-based surface profiling[110,111] offer additional means to enrich and classify EVs on the basis of cellular source, which may be used alongside our reference EV proteome dataset to further refine classification and improve translational application. Our dataset offers a high-confidence resource of tissue- and cell-enriched protein features in circulating EVs that can be harnessed to guide the development of affinity-based enrichment strategies, improve computational deconvolution models and support source-specific EV biomarker discovery. In future studies, combining these molecular leads with advanced single-EV technologies and disease-specific profiling could yield deeper insights into tissue-specific EV biogenesis, intercellular communication and their roles in health and disease. Importantly, despite disease-associated proteomic changes, the core set of 182 pEV protein features remained highly conserved across both CAC and non-CAC individuals in the EDCAD cohort, underscoring their robustness as molecular hallmarks of circulating EVs. This warrants future investigation on molecular stability across diverse disease states to establish the use of these core pEV features as a reliable foundation for human pEV studies in large population-based cohorts, enabling consistent characterization and cross-study comparability.

In summary, we identify core protein and lipid componentry of circulating EVs in humans. We propose these as 'hallmark molecular features' of circulating EVs, offering a valuable resource to the EV community and a tool to enhance the quality of human EV research.

## Online content

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

## Methods

### Generation of cell conditioned media

SW480 (CCL-288, ATCC), SW620 (CCL-227, ATCC) and LIM1863 cells[175] (Ludwig Institute for Cancer Research, Melbourne) cells were cultured in RPMI-1640 (Life Technologies). MDA MB 231 (HTB-26, ATCC) was cultured in Dulbecco's modified Eagle medium (Life Technologies). Primary human cell sources included the human neonatal foreskin fibroblast cell line (neoHFF) (kindly provided by P. Kaur, Australia) and adult human dermal fibroblasts (Gibco/Thermo Fisher Scientific, #C0135C). These cells were cultured in complete culture media supplemented with 5% (v/v) foetal bovine serum (FBS, Life Technologies) and 1% (v/v) penicillin–streptomycin (Life Technologies). Human atrial cardiac fibroblasts (Lonza, #CC-2903) and human ventricular cardiac fibroblasts (Lonza, #CC-2904) were cultured in FGM-3 Cardiac Fibroblast Growth Medium-3 BulletKit (Lonza, #CC-3131 and #CC-4525). Human umbilical vein endothelial cells (HUVEC, Lonza, #CC-2519; HUVEC sourced as gift (K. Peter, BHDI, Australia), HUVEC-RFP, Angio-Proteomie #cAP-0001) were cultured in EGM-2 Endothelial Cell Growth Medium-2 BulletKit (Lonza #CC-3156 and #CC-4176 supplements) prepared as detailed by the supplier. All cells were cultured at 37 °C with 10% $CO_2$ as described[45,68]. Cells were passaged with trypsin–EDTA (Gibco). Cells were cultured in CELLine AD-1000 Bioreactor classic flasks (Integra Biosciences) for conditioned media generation as described[68].

### Plasma preparation

Human blood plasma samples were obtained from Australian Red Cross Lifeblood, the EDCAD study or the AusDiab trial. For Red Cross, ethical permits were obtained from the Australian Red Cross Blood Service Human Research Ethics Committee, and La Trobe University Human Ethics Committee (HEC19485). Blood samples were collected via aseptic venipuncture into commercial EDTA-containing sampling containers at room temperature and centrifuged at 4,200g for 10 min. The top plasma supernatant was carefully collected and further centrifuged at 5,000g (15 min) at room temperature. The supernatant was immediately isolated and stored as 1-ml aliquots at −80 °C until further use. For EDCAD samples, ethics permit was approved through the Human Research Ethics Committee at Baker Heart and Diabetes Institute and by the Alfred Hospital Ethics Committee (EDCAD-PMS, #492/20). Blood was collected via venipuncture into a sterile EDTA tube and centrifuged at 2,700g at room temperature for 13 min. A small volume (1.6 ml) of the supernatant (plasma) was transferred into a clean sterile tube, and butylated hydroxytoluene solution (BHT) was added to a final concentration of 100 μM. The plasma/BHT sample was mixed and equally transferred into 3× 0.75 ml Fluid X cryotubes. Plasma/BHT samples were snap frozen on dry ice before being transferred and stored at −80 °C until use). For AusDiab trial samples, the ethics permit was approved through the Human Research Ethics Committee at Baker Heart and Diabetes Institute and by the Alfred Hospital Ethics Committee (#39/11). Blood was collected into commercial fluoride/oxalate tubes and centrifuged at 2,500 rpm for 10 min at room temperature. Plasma was isolated and snap frozen on dry ice before being transferred and stored at −80 °C until use. Clinical parameters for EDCAD plasma samples are provided in Supplementary Table 41.

For all plasma collections, only ~80% of top plasma supernatant was collected to avoid disturbing the buffy coat. Samples with potential haemolysis, cellular contamination and high fat/lipid content were not used for this study.

### Ultracentrifugation of plasma or conditioned media

To generate crude 100 K plasma pellet, plasma samples were thawed on ice and subjected to ultracentrifugation at 100,000g for 1 h or 18 hr at 4 °C (TLA-55 rotor; Optima MAX-MP Tabletop Ultracentrifuge, Beckman Coulter). Pellets were washed once in 1.0 ml phosphate-buffered saline (PBS), resuspended in 100 μl PBS and stored at −80 °C until further use.

Human fibroblasts and endothelial cells (80% confluent in T75 culture flasks) were cultured in Dulbecco's modified Eagle medium supplemented with 0.5% v/v insulin transferrin selenium (Invitrogen) and 1% penicillin–streptomycin) to generate conditioned media. To isolate EVs, we subjected conditioned media to differential centrifugation (500g for 5 min and 2,000g for 10 min at 4 °C), followed by concentration using Amicon Ultra Centrifugal Filter, 100 kDa molecular weight cut-off (low-speed centrifugation at 2,000g at 4 °C). The filter was washed five times with 1 ml ice-cold PBS. The concentrated retentate was retrieved and further centrifuged at 10,000g (30 min at 4 °C) to pellet crude large EVs (10 K EVs), while the resultant supernatant underwent ultracentrifugation at 100,000g (1 h at 4 °C) to pellet crude small EVs (100 K EVs). The isolated EVs were washed in PBS (1 ml, 10,000g for large EVs and 100,000g for small EVs) before resuspension in 50 μl PBS and stored at −80 °C until further use (subsequent proteomic and lipidomic analyses).

### Direct density-gradient separation

Plasma (500 μl) was directly overlaid on top of discontinuous gradient of OptiPrep (40% (3 ml), 20% (3 ml), 10% (3 ml) and 5% (2.5 ml) (diluent: PBS solution)) and ultracentrifuged at 100,000g for 18 h (4 °C, 41 Ti rotor; Optima XPN Ultracentrifuge). Twelve 1-ml fractions were carefully collected from top to bottom, diluted in PBS (2 ml) and ultracentrifuged at 100,000g (1 h, 4 °C, TLA-55 rotor; Optima MAX-MP Tabletop Ultracentrifuge). Fraction densities were determined as previously described[45]. Pellets were suspended in 100 μl PBS and stored at −80 °C until further use. For bottom-up DGS, 500 μl of plasma was carefully loaded at the bottom of a preformed discontinuous DGS (OptiPrep (40% (3 ml), 20% (3 ml), 10% (3 ml) and 5% (2.5 ml) (diluent: PBS solution)) using a long syringe to avoid disturbing the gradient. Ultracentrifugation was performed at 100,000g for 18 h (4 °C), as in the top-loaded DGS method. After centrifugation, 12 equal fractions (1 ml each) were collected, diluted with 2 ml PBS and subjected to another round of ultracentrifugation at 100,000g for 1 h (4 °C). The resulting pellets were resuspended in 50 μl PBS and analysed by western blotting.

For the isolation of cell culture-derived EVs, the conditioned medium was centrifuged at 500g (5 min, 4 °C), 2,000g (10 min, 4 °C) and 10,000g (30 min, 4 °C, SW28 rotor; Optima XPN Ultracentrifuge, Beckman Coulter) to pellet large EVs[68]. The supernatant (1 ml) was subjected to direct DGS for EV isolation as described for plasma above. EV pellets were reconstituted in 100 μl PBS and stored at −80 °C until further use.

### Immunoblotting

Samples were subjected to protein quantification (microBCA Protein Assay Kit (23235, Thermo Fisher Scientific)), and western blotting (iBlot 2 Dry Blotting System, Thermo Fisher Scientific) was performed as described[45,68]. Rabbit antibodies raised against albumin (ab207327, Abcam) and AGO2 (ab186733, Abcam) were used. Mouse antibodies CD63 (556019, BD Pharmingen), CD81 (555675, BD Pharmingen), APOB100 (3715-3-250, Mabtech) and APOA1 (3710-3-1000, Mabtech) were used (1:1,000). Secondary antibodies used were IRDye 800 goat anti-mouse IgG or IRDye 700 goat anti-rabbit IgG (1:15000, LI-COR Biosciences).

### Biophysical characterization of EVs

Cryo-electron microscopy (Tecnai G2 F30) on samples (1 μg) was performed as described[47]. In brief, Plasma 100 K, D-DGS fractions 1–3 (pooled) and pEVs (DGS fractions 6–8, pooled) (~1 μg) were loaded onto to glow-discharged C-flat holey carbon grids (ProSciTech Pty) that, after blotting away excess liquid, were plunge-frozen in liquid ethane and subsequently mounted in a Gatan cryoholder (Gatan) in liquid nitrogen. Images were then acquired at 300 kV using a Tecnai G2 F30 (FEI) in low-dose mode. EV samples (~1 μg; 1:1,000 dilution) were

prepared in PBS (#14190-144, Thermo Fisher Scientific)[176], and particle size distribution and zeta potential (surface charge) were determined by nanoparticle tracking analysis (ZetaView, Particle Metrix, PMX-120; 405-nm laser diode) according to manufacturers' instructions.

### Surface biotin labelling of EVs and proteomic sample preparation

EV surface proteins (from SW620 cells, $n = 3$ biological replicates) were captured using the Pierce Cell Surface Biotinylation and Isolation Kit (A44390, Thermo Fisher Scientific) and digested with trypsin (Promega, V5111), and the resulting peptides were analysed using the Q-Exactive HF-X, as previously described[68].

### Label-free proteomics sample preparation

Label-free proteomics sample preparation (~2–10 µg in 20–50 µl) was performed as described based on the single-pot solid-phase-enhanced sample preparation (SP3) workflow[68]. Protein sets were digested with trypsin and Lys-C (1:50 and 1:100 enzyme-to-protein ratio, respectively), and peptide digests were frozen at −80 °C and lyophilized by vacuum-based speedVac (Savant SPD121P, Thermo Fisher Scientific), reconstituted in 0.07% trifluoroacetic acid and quantified by fluorometric peptide assay (23290, Thermo Fisher Scientific).

### TMT-based proteomics sample preparation

TMT-based labelling was performed as described[177], using 10-plex TMT according to the manufacturer's instructions with modifications (Thermo Fisher Scientific, 90406, lot UG287488/278919). The 10-plex experiment included nine different chemical tags for EV peptide labelling, with the tenth tag used as a reference channel generated from a pooled peptide digest of all samples. A list of the sample labelling strategy is available in MassIVE proteomeXchange (MSV000094307). Peptide samples (15.5 µg) were labelled with TMT10plex Isobaric Label (90406, Thermo Fisher Scientific) at 4:1 label-to-peptide ratio for 2 h at room temperature and quenched with 0.5% (v/v) hydroxylamine for 30 min at room temperature. Labelled peptide samples were acidified with 4% (v/v) formic acid (FA) and pooled into a new microtube. Pooled samples were desalted using Sep-Pak tC18 96-well µElution plates (186002318, Waters), and the eluates were lyophilized using a SpeedVac. Peptides were fractionated into 20 fractions (with increasing concentration of acetonitrile from 2% to 50%) using high-pH reversed-phase chromatography with in-house SPE StageTips (SDB-RPS, Empore). Peptide eluates were lyophilized, reconstituted in 0.07% trifluoroacetic acid (TFA) and quantified using a colorimetric peptide assay. Peptide fractions with low yields were pooled.

### Nano liquid chromatography (LC)–MS/MS

Peptides (label-free and TMT-labelled) were analysed in randomized sequence batches on a Dionex UltiMate NCS-3500RS nanoUHPLC coupled to a Q-Exactive HF-X hybrid quadrupole-Orbitrap mass spectrometer equipped with nanospray ion source operating in positive mode as described[68,178].

For label-free proteomics, peptides were loaded (Acclaim Pep-Map100 C18 3-µm beads with 100 Å pore size, Thermo Fisher Scientific) and separated (1.9-µm particle size C18, 0.075 × 250 mm, Nikkyo Technos) with a gradient of 2–28% acetonitrile containing 0.1% FA over 49 or 95 min at 300 nl min$^{-1}$ at 55 °C (butterfly portfolio heater, Phoenix S&T).

For data-dependent acquisition (DDA), MS1 scan was acquired from 350 to 1,650 $m/z$ (60,000 resolution, $3 \times 10^6$ automatic gain control (AGC), 128 ms injection time) followed by MS/MS DDA (top 25) with collision-induced dissociation and detection in the ion trap (30,000 resolution, $1 \times 10^5$ AGC target, 27 ms injection time, 28% normalized collision energy, 1.3 $m/z$ quadrupole isolation width). Unassigned and precursor ions with charge states 1 and 6–8 were rejected, and peptide match was disabled. Selected sequenced ions were dynamically excluded for 30 s. Data were acquired using Xcalibur software v4.0.

For data independent aqusition (DIA)[179], full-scan MS was performed in the $m/z$ range of 350–1,100 $m/z$ with a 60,000 resolution, using an AGC of $3 \times 10^6$, maximum injection time of 50 ms and 1 microscan. MS2 was set to 15,000 resolution, with an AGC target of $1 \times 10^6$ and the first fixed mass set to 120 $m/z$. The default charge state was set to 2 and recorded in centroid mode. Total scan windows (38 windows; 49-min gradient, 63 windows, 95-min gradient) with a staggered isolation window from 350 to 1,100 $m/z$ were applied with 28% normalized collision energy.

TMT-labelled peptides were separated (1.9 µm particle size C18, 0.075 × 250 mm, Nikkyo Technos) with a gradient of 4–28% acetonitrile containing 0.1% FA over 110 min at 300 nl min$^{-1}$ at 55 °C (butterfly portfolio heater, Phoenix S&T). The MS1 scan was acquired from 300 to 1,650 $m/z$ (60,000 resolution, $3 \times 10^6$ AGC, 128 ms injection time) followed by MS/MS DDA of the top 15 ions with higher-energy collisional dissociation (HCD) (30,000 resolution, $1 \times 10^5$ automatic gain control (AGC), 60 ms injection time, 33 normalized collision energy (NCE), 0.8 $m/z$ isolation width). Unassigned precursor ions and those with charge states 1 and 6–8 were rejected, and peptide matching was disabled. Selected sequenced ions were dynamically excluded for 30 s.

MS-based proteomics data (including sample/label annotation) are deposited to the ProteomeXchange Consortium via the MassIVE partner repository and available via MassIVE with identifier MSV000094307.

### Proteomic data processing

For DDA and TMT proteomics, peptide identification and quantification were performed using MaxQuant (v1.6.6.0-v1.6.14) with its built-in search engine Andromeda as described[68] against *Homo sapiens* (UP000005640) including the contaminants database. For label-free quantification-based analyses (DDA), cysteine carbamidomethylation was selected as a fixed modification and N-terminal acetylation and methionine oxidations as variable modifications. Data were processed using trypsin/P as the proteolytic enzymes with up to two missed cleavage sites allowed. Further processing was performed using the 'match between runs' feature and the label-free quantification algorithm. Peptides were identified with an initial precursor mass deviation of up to 7 ppm and a fragment mass deviation of 20 ppm with a minimum length of 7 amino acids. FDR was 0.01 for both the protein and peptide by searching against a reverse database. For TMT-based analyses, reporter ion MS2 (TMT10plex) settings were used. For biotin surface proteome analysis, an additional thioacyl (DSP, $C_3H_4OS$) was used[47].

For DIA proteomics, data processing was performed using DIA-NN[180] (v1.8). Spectral libraries were predicted using the neural network deep learning algorithm used in DIA-NN with trypsin/P, allowing up to one missed cleavage. The precursor change range was set to 1–4, and the $m/z$ precursor range was set to 300–1,800 for peptides consisting of 7–30 amino acids. N-terminal methionine excision and cysteine carbamidomethylation were enabled as fixed modifications, with the maximum number of variable modifications set to zero. The mass spectra were analysed using default settings with a FDR of 1% for precursor identifications and match between runs enabled for replicates. Contaminants and reverse identifications were removed from the resulting output files.

### Lipid extraction

Lipid extraction was performed using a modified single-phase butanol/methanol extraction method as described[181]. Samples (2–5 µg protein from lyophilized sample in 10 µl of water) were mixed with 100 µl of 1:1 butanol:methanol containing internal standards; the samples were vortexed, sonicated on a sonicator bath (1 h) and centrifuged (13,000g, 10 min). The supernatants were transferred into glass vials and stored at −80 °C. Before MS analysis, samples were thawed for 1 h at room temperature, vortexed, sonicated in a sonication bath for 15 min and then left to equilibrate at 20 °C for 2 h before analysis.

## LC–MS (discovery)

Analysis of lipid extracts was performed (in randomized sequence batches) on an Agilent 6495C QQQ mass spectrometer with an Agilent 1290 series high-performance LC (HPLC) system, including a ZORBAX eclipse plus C18 (2.1 × 100 mm 1.8 μm, Agilent) rapid resolution high-throughput column. MS settings and transitions for each lipid class are provided (Supplementary Table 40), adapted from previous methodology[182]. Conditions included: gas temperature 150 °C, gas flow rate 17 l min⁻¹, nebulizer gas pressure 20 psi, sheath gas temperature 200 °C, capillary voltage 3,500 V and sheath gas flow 10 l min⁻¹. Isolation widths for Q1 and Q3 were set to 'unit' resolution (0.7 amu). HPLC conditions included solvent A (50:30:20 water:acetonitrile:isopropanol) and solvent B (1:9:90 water:acetonitrile:isopropanol), with a gradient of 15–100% solvent B over 16 min (total runtime: 20 min), at a flow rate of 0.4 ml min⁻¹ and a column compartment temperature of 45 °C. Solvent A was buffered with 10 mM ammonium formate with medronic acid, while solvent B was buffered with 10 mM ammonium formate.

## LC–MS (validation) for AusDiab and fibroblasts/endothelial cell EVs

Analysis of lipid extracts was performed on an Agilent 6495C QQQ mass spectrometer with an Agilent 1290 series HPLC system similar to the discovery analysis. Mass spectrometer conditions were as follows: gas temperature 200 °C, gas flow rate 17 l min⁻¹, nebulizer gas pressure 20 psi, sheath gas temperature 280 °C, capillary voltage 3,500 V and sheath gas flow 10 l min⁻¹. Isolation widths for Q1 and Q3 were set to 'unit' resolution (0.7 amu). Solvent composition was as per discovery run, with a modified gradient of 15–100% B, over 11 min (total runtime 16 min), at 0.4 ml min⁻¹ and column compartment at 45 °C. Further characterization was performed on PC, PC(O) and PC(P) isobaric and isomeric species to highlight the separation resolution of the rapid resolution high throughput (RRHT) column (Supplementary Fig. 21).

## Lipid data integration and statistical analysis

Raw lipidomic MS data were analysed using MassHunter Quant 10.0 (Agilent Technologies). The peak area of lipid species was normalized to their respective internal standards to generate relative concentration data per sample. For lipids that appear to have isomeric separation on the chromatography, they are designated with the (a) and (b) annotations to highlight different elution orders. Lipid class totals were generated by summing the individual species within each class. MS-based lipidomic RAW data and sample/label annotations have been deposited in the ProteomeXchange Consortium via the MassIVE partner repository and are available under the identifier MSV000094307.

## Structural characterization of lipid isomers and isobars

As PS 36:1 represented a strong signature for EVs, we determined the structure of this PS species in isolated EV's, we reran pooled samples under negative ionization mode under the same gradient conditions, screening for product ions corresponding to the serine head group (788.5 $m/z \rightarrow$ 701.5 $m/z$) and the fatty acyl tails. The final composition was annotated to be PS 18:0_18:1, with signals observed with 283.3 and 281.3 product ions (Supplementary Fig. 20). The method used for these samples monitors CEs in positive ionization mode as ammonium adducts, tracking the COH ion along with an additional water loss (CE 18:0 [M + NH₄] + 670.6/369.3). A matching internal standard (CE 18:0-d6) was used to quantify this lipid species. Retention time, mass and fragmentation patterns were used to confirm its annotation.

## Nanoflow analysis of pEVs

pEVs (~5 μg) were subjected to fixation and permeabilization using eBioscience Foxp3/Transcription Factor Staining Buffer Set (Invitrogen, 00-5523-00). In brief, pEVs (in 50 μl PBS) were incubated with 500 μl fixation and permeabilization buffer on ice for 30 min. Samples were ultracentrifuged at 100,000*g* (1 h at 4 °C), and pellets were resuspended in 100 μl wash buffer. Samples were stained with 5 μl APC Annexin V reagent (BioLegend) and 2 μg of either anti-ADAM 10 antibody (Sigma-Aldrich, AB19026) or rabbit IgG Isotype Control (Invitrogen, 10500C). Samples were incubated at room temperature (gentle end-over mixing) for 1 h. Samples were topped with 900 μl wash buffer and washed twice (ultracentrifuged at 100,000*g* (1 h at 4 °C)) to remove remaining antibodies and potential antibody aggregates. The pellets were resuspended in 100 μl wash buffer containing 0.5 μl of goat anti-rabbit Alexa Fluor 568 Dye secondary antibody (Thermo Fisher Scientific) incubated for 30 min at room temperature (gentle end-over mixing) in the dark. Samples were topped with 900 μl wash buffer and washed twice (ultracentrifuged at 100,000*g* (1 h at 4 °C)). Pellets were resuspended in 100 μl of PBS (0.5% bovine serum albumin) using a 0.22-μm filter. Samples were immediately analysed using Cytek Aurora flow cytometer and SpectroFlo software, as previously described[86]. In brief, controls included isotype control antibodies to assess non-specific binding, along with unstained EV suspensions and buffer-only controls. For purity assessment, SDS was added to labelled EVs at a final concentration of 0.5%, followed by vigorous vertexing for ~30 s before acquisition. High purity is indicated by a marked reduction in vesicle concentration and fluorescence following treatment. The instrument set-up was consistent across all experiments and followed recommendations from the manufacturer. Before the acquisition, the flow cell was washed with Contrad to minimize machine-associated noise. Instrument gating calibration was performed using 90-nm (#64009-15) 125-nm (#64011-15), 150-nm (#64012-15), 200-nm (#64013-15) and equal mix (90–200 nm) beads (Nanobead NIST Traceable Particle Size Standards). The threshold for side scatter was set to 430, and the gain of side scatter was set to 10. The YG3-A channel was used to detect the Alexa Fluor 568 signal, and the R1-A channel was used to detect the APC signal. For all samples, 10,000 events were recorded at the lowest flow rate to minimize the swarming effect. Statistical analysis of flow cytometry values was performed using GraphPad (v 9.1.0) using multiple paired *t*-tests.

## Isolation of EVs using affinity to TIM4

To capture PS-positive EVs from plasma, human TIM4-Fc protein (FUJI-FILM Wako Pure Chemical Corporation) was biotinylated with EZ-Link Maleimide-PEG11-Biotin (Thermo Fisher Scientific) and conjugated to Dynabeads MyOne Streptavidin C1 magnetic beads (65001, Thermo Fisher Scientific) as described[183]. In brief, plasma (200 μl), diluted with 300 μl of PBS, was supplemented with heparin sodium (4U, Thermo Fisher Scientific) and CaCl₂ (final concentration of 2 mM) and incubated with TIM4 affinity beads for 16 hr at 4 °C with gentle rotation. Beads were collected and washed 5× with 1 ml Tween–TBS (0.05% Tween-20, 2 ml CaCl₂). EVs were eluted in 50 μl PBS containing 1 mM EDTA and subjected to DIA-based proteomic analyses as described.

## Data processing and statistical analysis

Software tools used for this study are freely available as open-source R packages (https://www.r-project.org). No newly generated software or custom code were used in this current study, and hence, the codes have not been deposited in a public repository but are available from the corresponding author(s) upon request.

For key analyses, proteome and lipidome datasets were analysed using the R package Differential Enrichment analysis of Proteomics data (DEP)[184]. Proteins identified in at least 70% of one biological group were selected for downstream analysis. Using DEP, the data were background-corrected and normalized by variance stabilizing transformation (vsn), which also performs log₂ transformation. Missing values were then imputed: missing at random data were imputed using the 'knn' method, while missing not at random data were imputed using the 'MinProb' method. Protein-wise linear models combined with empirical Bayes statistics were used for the differential enrichment analysis, whereby the raw *P* values were adjusted to correct for multiple testing

using Benjamini–Hochberg method. Differentially abundant proteins or lipids were clustered by *k*-means clustering using the DEP package. The PCA plot, Pearson correlation matrix, volcano plots, $\log_2$-centred bar plots and overlap bar plots were also generated using DEP. Heatmaps were generated using ComplexHeatmap package[185]. Box plots and scatter plots were generated using RStudio package ggplot2. Cytoscape[186] was used to generate the Ontology map[187] (plugin v3.7.1). Bioconductor package clusterProfiler 4.0 (ref. [188]) was used to perform GO and KEGG pathway enrichment analyses, as well as KEGG gene set enrichment analysis, with default parameters to identify significantly enriched gene sets. The pathway-based data integration and visualization was constructed using R package pathview[189]. For identification of lipid-associated terms enriched in lipidomes, the LION web-based ontology enrichment tool[190] was used.

For conserved protein identification analysis, proteins were classified as detected or not detected across samples (protein abundance was not considered). In surface biotin labelling of EVs, proteins detected in at least six out of nine experiments were considered as pEV surface proteins (protein abundance was not considered). For annotating surface proteins into different categories, SURFY[59]-based categorical annotation of cell surface proteins were used. To identify pEV lipid features, we performed *K*-means clustering of differentially abundant lipids between pEVs versus NonEVs and in vitro EVs versus cells using DEP package.

We used the caretEnsemble R package to assess the ability of these protein features (182 pEV protein features and 42 NonEV protein features, using the naive Bayes algorithm), surface protein features (comprising 151 surface proteins associating with pEV surface and 32 surface proteins associating with NonEVs, using the naive Bayes algorithm) or lipid features (114 pEV lipid features and 52 NonEV lipid features, using the neural network algorithm ('nnet')) to distinguish between distinguish between pEV and NonEV particles. The pEV and NonEV proteome/lipidome datasets were evenly partitioned based on sample type into a training set (70% of the samples) and a validation set (remaining 30% of samples). For protein features based machine learning, proteomes of pEVs (*n* = 16) and pDGS.LD (NonEV particles, *n* = 16) from as 16 plasma samples of the EDCAD cohort served as an independent test set. For lipid features-based machine learning, lipidomes of DGS plasma fractions served as an independent test set. A bootstrapping resampling method was used with 25 resampling iterations to train the models. The performance of the models was evaluated using class probabilities and the two-class summary function. The preprocessing steps of centring and scaling were applied to the predictors before training the models. A confusion matrix comparing predicted classes with actual classes, based on the machine learning algorithm used, was generated to visualize the model's predictive performance. For identification of ADAM10 protein or PS(36:1) and CE(18:0) lipids as biomarker candidates, RFE provided by the caret R package for feature selection using default options was used.

To address the question of source attribution in a data-driven manner, we manually curated tissue- and cell-type-specific protein signatures from three independent sources: (1) the Human Protein Atlas (HPA), which provides tissue- and organ-specific proteins based on immunohistochemistry (Extended Data Fig. 2a,b); (2) the HPA nTPM dataset, an RNA-derived protein expression database across tissues and cells (Extended Data Fig. 2c); and (3) organ-specific protein signatures detected in human plasma, identified via aptamer-based detection[57] (Extended Data Fig. 2d). To systematically assess the presence of bioactive molecules in pEVs, we manually curated and integrated multiple independent datasets that catalogue biologically significant protein classes. Specifically, we leveraged established resources for human transcription factors[61] (Human TFs Database), RNA-binding proteins[60] (RBPDB Database), kinases[58] (KinHub Database) and cell-surface receptors, transporters and signalling proteins[59]. In addition, we incorporated GO terms related to cytokine activity (GO:0005125), chemokine activity (GO:0008009), growth factor activity (GO:0008083) and signal transduction (GO:0007165) to ensure comprehensive coverage of bioactive EV-associated molecules.

As a resource, the Shiny web application (https://evmap.shinyapps.io/evmap/) powered by R and hosted on shinyapps.io, was developed using the R packages shiny, gplots, and ComplexHeatmap. It offers feature selection and visualization tools for EV protein and lipid feature conservation in circulating EVs.

## Statistics and reproducibility

All experiments were performed with at least three independent biological replicates unless otherwise specified. Each replicate was derived from separate cell culture preparations or human plasma samples to ensure biological variability as detailed in the corresponding figure legends. Conserved and differential expression (protein and lipid) analyses were not performed blind due to the conditions of the experiments. No data were excluded from the analyses. The raw *P* values were adjusted to correct for multiple testing using the Benjamini–Hochberg method and are reported in the corresponding figure legends and supplementary tables. The statistical tests were chosen on the basis of the data distribution and experimental design. The quantitative data are presented as the mean or median ± standard error of the mean as indicated in the text or figure legends. Statistical analyses were conducted using R or GraphPad Prism. Proteome and lipidome datasets were vsn-normalized, but normality and variance were not formally tested. No statistical methods were used to predetermine sample sizes; the sample sizes were determined on the basis of previously published studies and pilot experiments. All key findings were independently validated in at least three separate experiments, and reproducibility was confirmed across different experimental set-ups, plasma source or cell lines when applicable.

## Reporting summary

Further information on research design is available in the Nature Portfolio Reporting Summary linked to this article.

## Data availability

The data supporting the findings presented are available within this Resource or its Supplementary Information or source data files. All of the data supporting the findings of this study are further available from the corresponding author upon reasonable request. The raw MS files (proteome and lipidome), sample/label annotation (Source Data 3) and the search/identification files obtained using MaxQuant/DIA-NN have been deposited to the ProteomeXchange Consortium via the MassIVE partner repository and available via MassIVE with identifier MSV000094307. Source data are provided with this paper.

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

## Acknowledgements

The D.W.G. laboratory is supported by research funds from the National Health and Medical Research Council (NHMRC, MRF2015523, APP1141946), National Heart Foundation (NHF, 105072), Helen Amelia Hains Fellowship (D.W.G.) and Department of Defense (PR230065). The Baker Heart and Diabetes Institute acknowledges support by the Victorian State Government Operational Infrastructure funding. The funders had no role in the study design, data collection and analysis, decision to publish or preparation of the manuscript. We thank E. Hanssen and the Bio21 Molecular Science and Biotechnology Institute for assisting cryo-electron microscopy (University of Melbourne). K.H. is supported by an NHMRC Emerging Leadership Fellowship (GNT1197190). Q.H.P., H.F and B.C. are supported through an Australian RFP Scholarship, in addition to the Bright Sparks Foundation. Schematic figures were created with BioRender.com.

## Author contributions

A.R. and D.W.G. conceptualized the idea. A.R. designed the experiments and wrote the manuscript. A.R., Q.H.P., H.F., B.C., J.C., T.D., C.D. and D.W.G. performed experiments. A.R. performed bioinformatics analysis. A.R., K.H. and D.W.G. analysed the results. C.D., J.E.S. and T.H.M. provided resources. A.R. developed the software/app. D.W.G acquired funding. A.R., K.H., B.C., J.E.S., T.H.M., P.M. and D.W.G. reviewed the manuscript for submission.

## Funding

## Competing interests

The authors declare no conflicts of interest.

## Additional information

**Extended data** is available for this paper at https://doi.org/10.1038/s41556-025-01795-7.

**Correspondence and requests for materials** should be addressed to Alin Rai or David W. Greening.

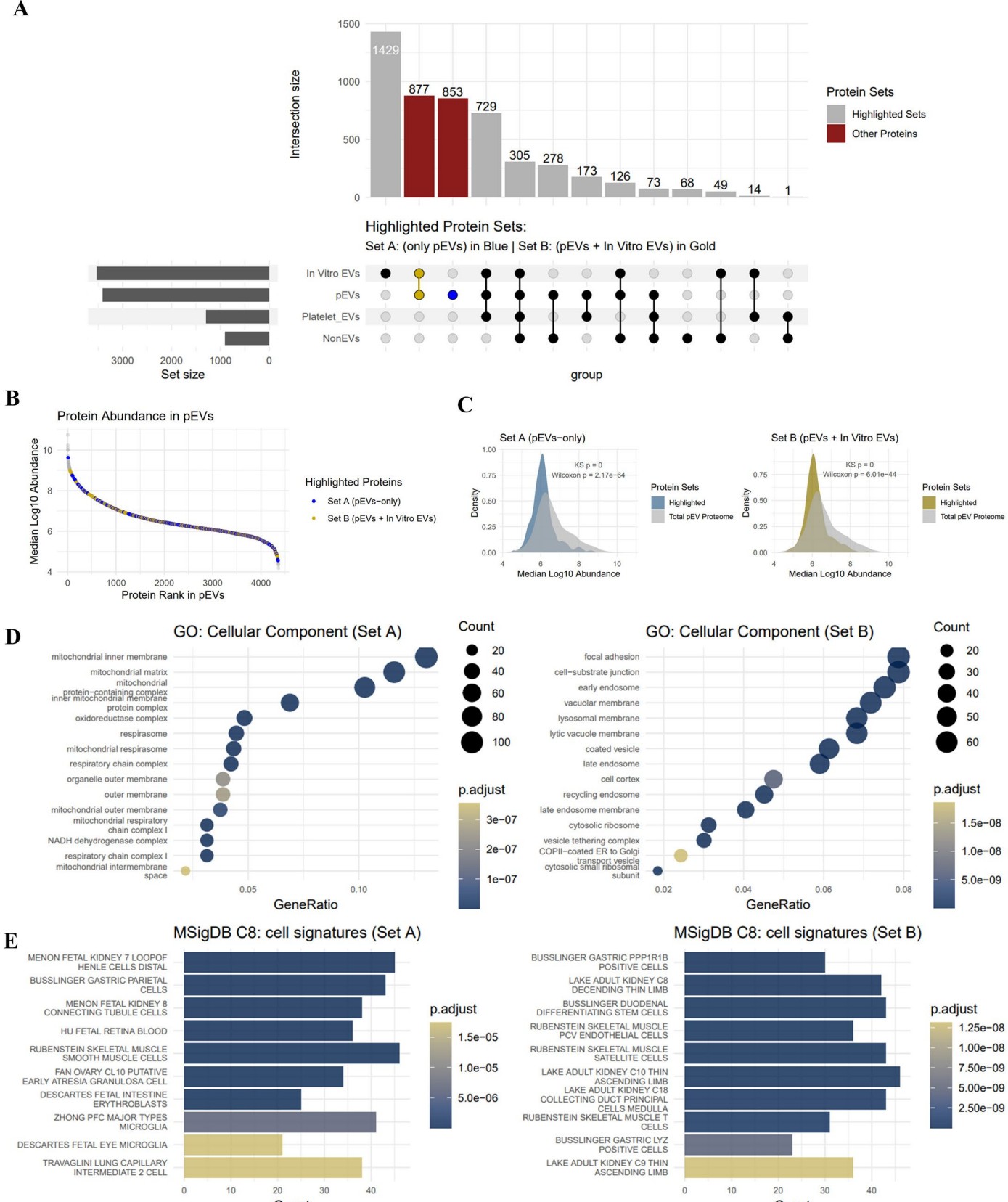

**Extended Data Fig. 1 | Diverse cellular source attribution to pEV proteome.**
**A**. Distribution of activated platelet EV proteins in pEVs and in vitro EVs. Set
A and Set B proteins represent proteins quantified exclusively in pEVs and
in vitro EVs versus activated platelet EVs[53]. **B**. Distribution of Set A and Set B
proteins (abundance) in pEV proteome. **C**. Density plots of Set A and Set B

protein abundance vs pEV proteome abundance. **D**. Gene Ontology (Cellular
Component) analysis of Set A and Set B proteins (Benjamini-Hochberg corrected
p.adjust < 0.05). **E**. Gene Set Enrichment Analysis (using Molecular signature
database[191] for cell signatures (C8 category)) of Set A and Set B proteins
(Benjamini-Hochberg corrected p.adjust < 0.05).

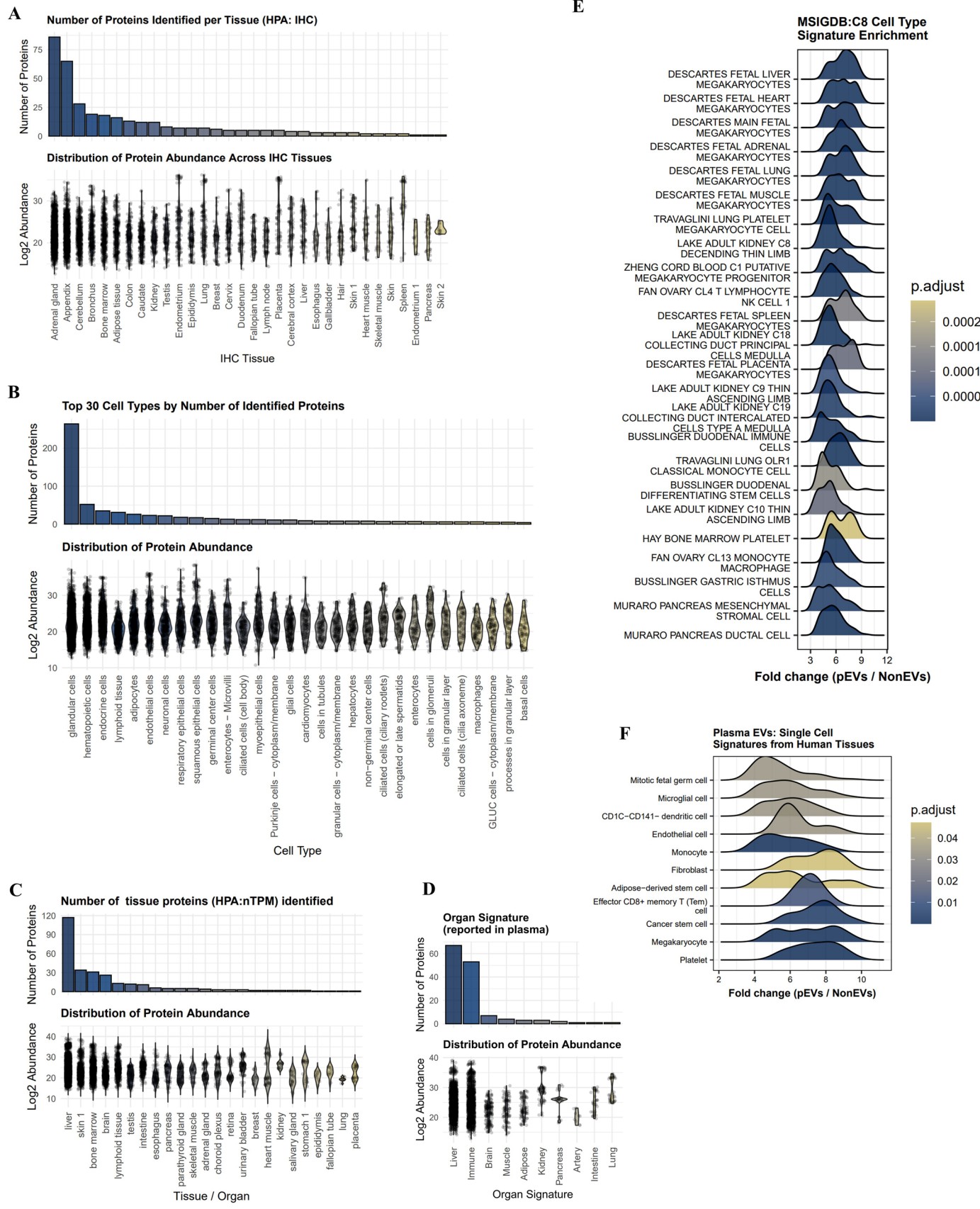

**Extended Data Fig. 2 | See next page for caption.**

**Extended Data Fig. 2 | Quantification of tissue / organ-associated proteins in pEVs. A**. Number of tissue-associated proteins (based on Human Protein Atlas (HPA) immunohistochemistry data (IHC)[56]) quantified in pEV proteome. Abundance distribution of proteins are presented in lower panel. **B**. Number of cell-type-associated proteins (based on HPA IHC) quantified in pEV proteome. Abundance distribution of proteins is presented in lower panel. **C**. Number of tissue-associated proteins (based on HPA normalized transcript per million (nTPM)) quantified in pEV proteome. Abundance distribution of proteins is presented in lower panel. **D**. Number of organ-associated proteins reported in plasma using aptamers quantified in pEV proteome[57]. Abundance distribution of these proteins is presented in lower panel. Violin plots in **A-D** show the distribution of log2 protein abundance (individual points are overlaid). **E**. Gene Set Enrichment Analysis (using Molecular signature database[191] for cell signatures (C8 category) of proteins enriched in pEV versus NonEV proteomes (Benjamini-Hochberg corrected p.adjust < 0.05). **F**. Enrichment of cell signatures based on single-cell sequencing studies of human tissue[191] in proteins enriched pEV versus NonEVs. (Benjamini-Hochberg corrected p.adjust < 0.05).

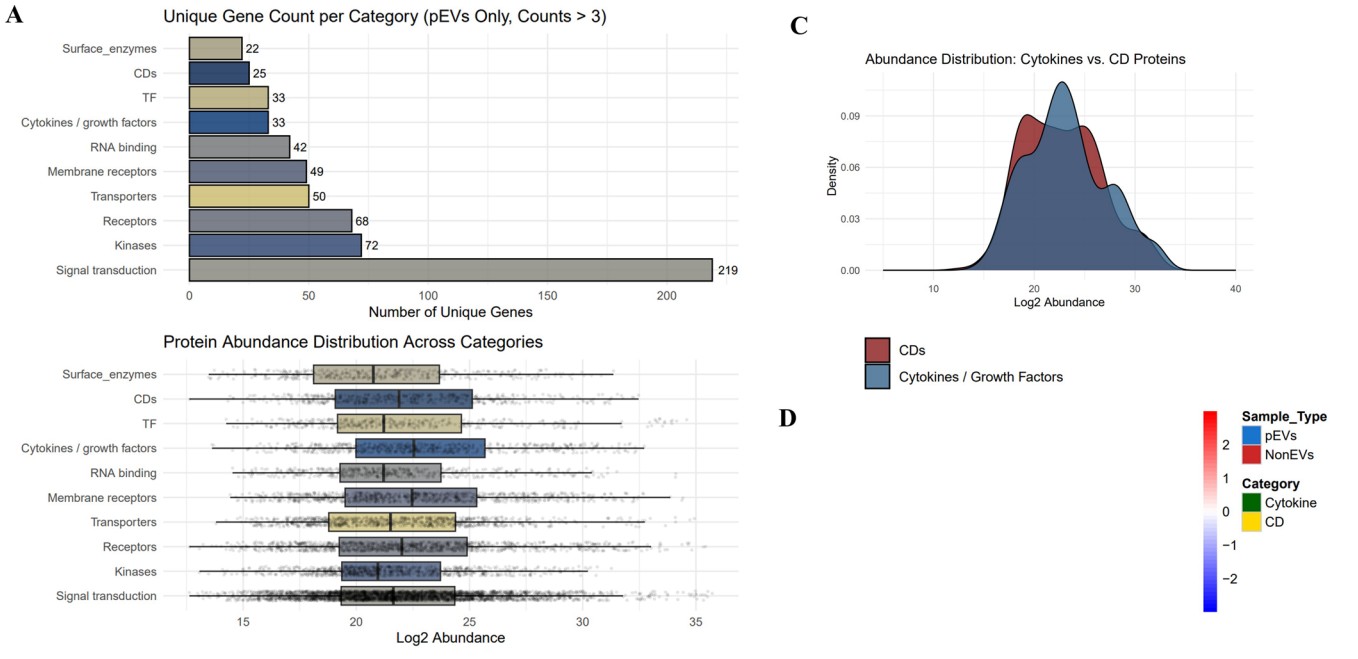

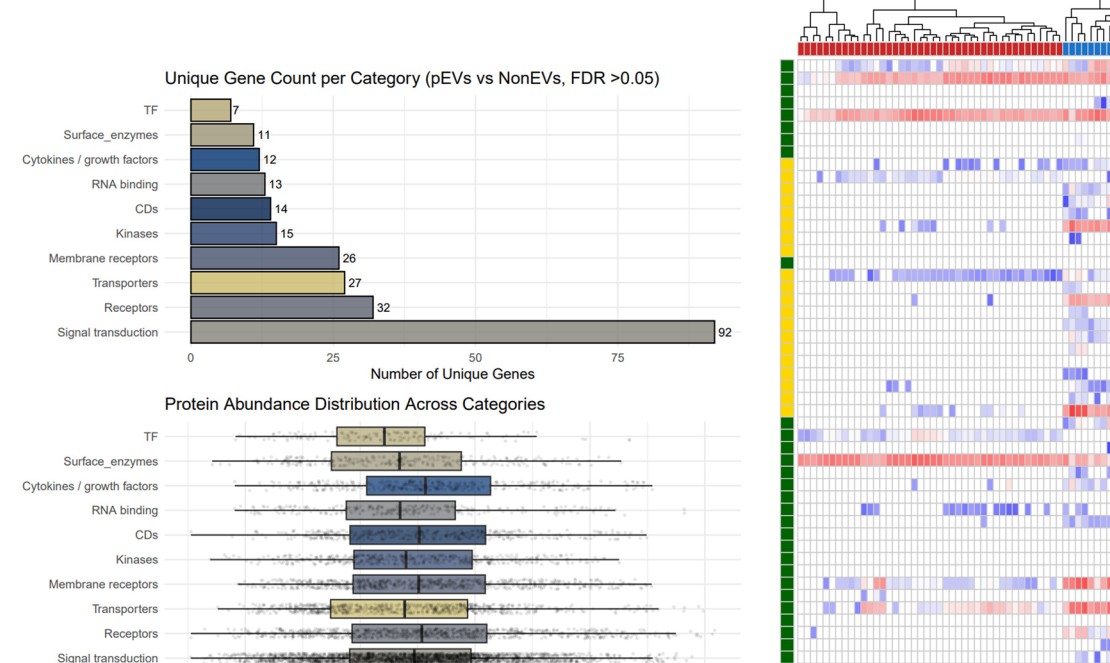

**Extended Data Fig. 3 | Diversity of bioactive cargo in pEVs. A.** Number of different cargo-type quantified in pEV proteomes (in at least 3 out of 38 pEV proteomes, independent plasma samples), with boxplot in lower panel representing protein abundance distribution. TF, transcription factors; CDs, Cluster of Differentiation. Gene Ontology terms related to cytokine activity (GO:0005125), chemokine activity (GO:0008009), growth factor activity (GO:0008083), and signal transduction (GO:0007165) to ensure comprehensive coverage of bioactive EV-associated molecules **B.** Number of different cargo-type quantified in pEV proteomes (but significantly enriched in pEVs vs NonEVs,

FDR < 0.05, n = 38 pEV proteomes, n = 42 NonEV proteome, independent plasma samples), with boxplot in lower panel representing protein abundance distribution. Boxplots in **A-B** show the median (centre line), 25th-75th percentiles (box), minima and maxima within 1.5× interquartile range (whiskers), and outliers beyond. **C.** Abundance distribution of cytokines/growth factors and CD proteins quantified in pEV proteome. **D.** Heatmap of cytokines and CD proteins quantified in pEV (n = 38) and NonEV (n = 42) proteomes (independent plasma samples). Grey boxes in the heatmap represent non-quantification.

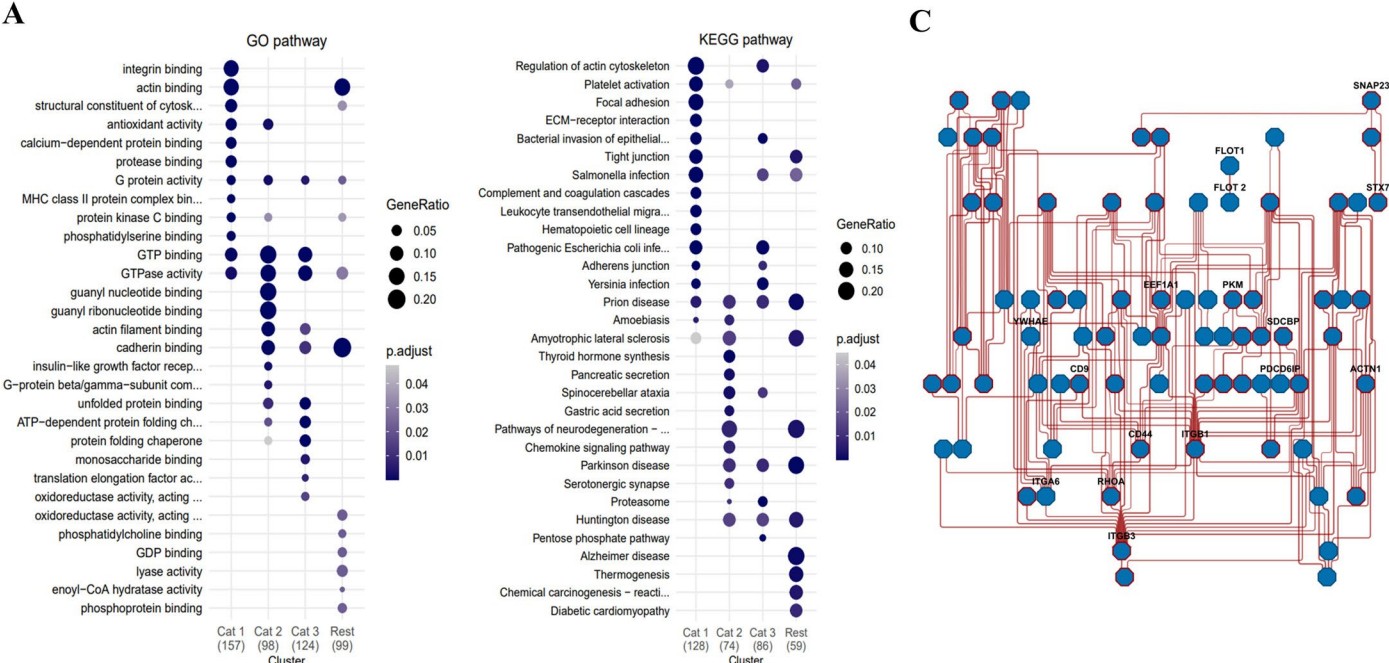

**Extended Data Fig. 4 | Characterization of conserved proteins in pEVs.**
**A**. Gene Ontology pathways and KEGG pathways enriched (p < 0.05) in each cluster. **B**. KEGG endocytosis pathway graph rendered by Pathview where category 1 proteins are indicated as high and category2-3 proteins are indicated as low. **C**. Protein–protein interaction networks of cluster 1 proteins analysed using STRING-DB and Cytoscape. Each node represents a protein. Nodes with red border are enriched for "vesicular pathway".

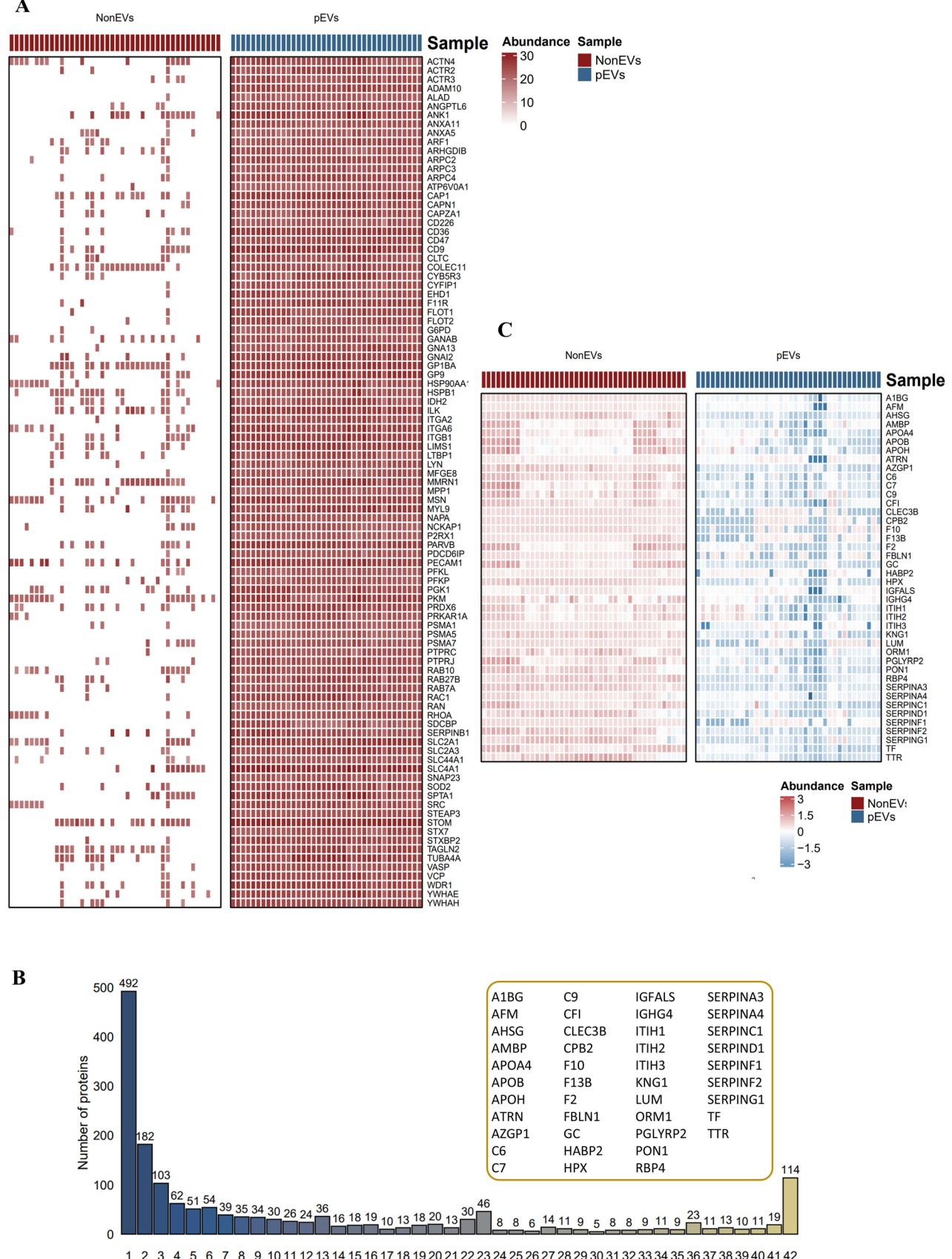

**Extended Data Fig. 5 | Mapping the core proteome of pEVs and NonEVs in human. A.** Heatmap of selected pEV protein features (38 individual plasma samples/proteomes). **B.** Occurrence analysis of proteins in 38 pEV proteomes where Category 1 proteins are ubiquitously quantified (inset). **C.** Heatmap of conserved protein features in NonEVs.

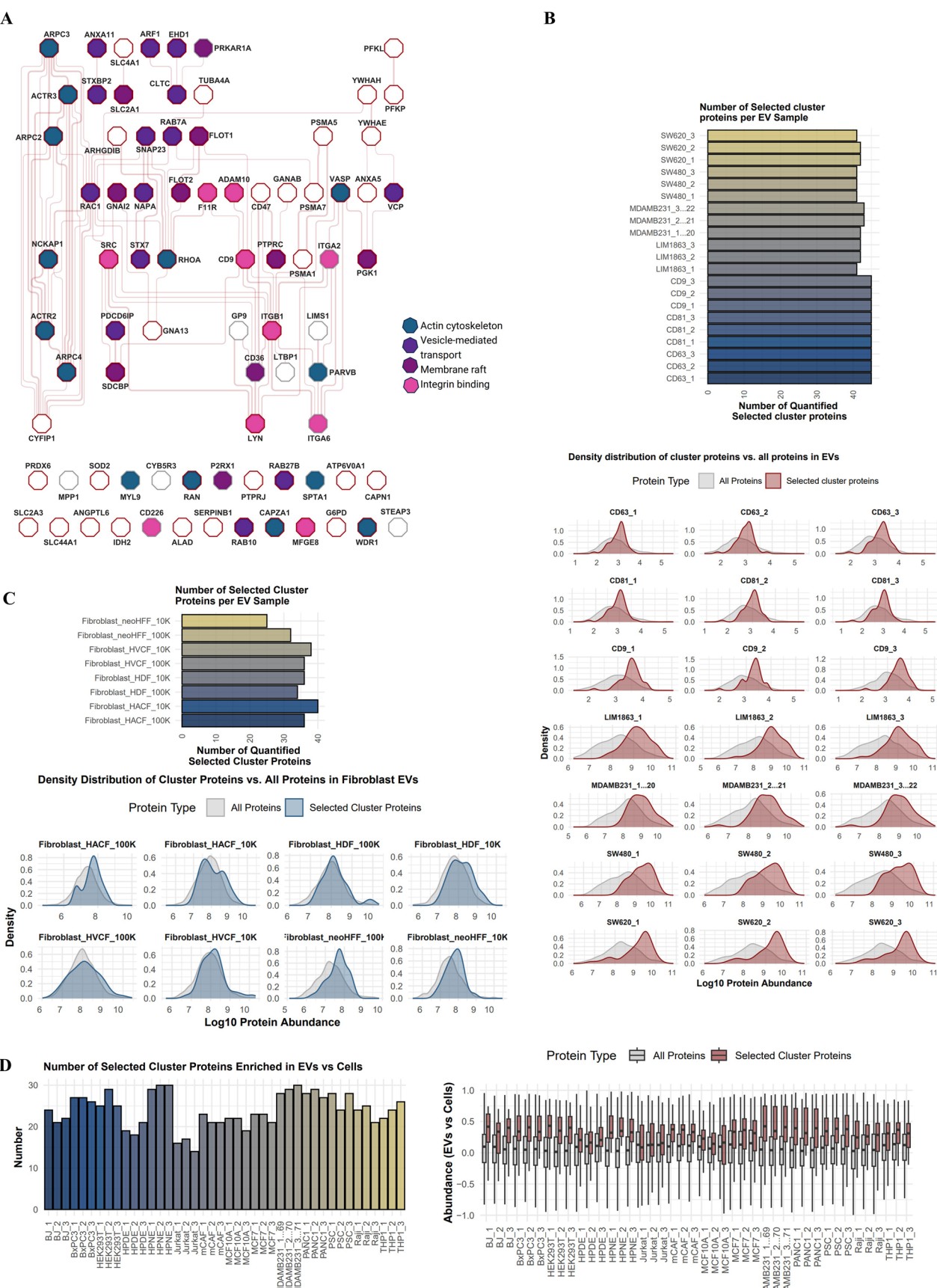

**Extended Data Fig. 6 | See next page for caption.**

**Extended Data Fig. 6 | Coordinated molecular patterns across EV proteomes.**
**A**. Protein–protein interaction networks of selected EV protein features (quantified in <30% of NonEV datasets) analyzed using STRING-DB and Cytoscape; each node represents a protein. Functional enrichment (GO cellular component) was performed using Cytoscape. Nodes with red border are enriched for "Extracellular exosome" term. **B**. Number of proteins belonging to coordinated molecular patterns (in Panel **A**) quantified in in vitro EVs (n = 3 per cell line) or EV subtypes (CD81 + , CD63 + , and CD9+ EVs) prevalent in the circulatory system[70]. Quantification in MS data of individual biological replicates are shown. Lower panel shows distribution of protein abundance of these molecular features against all other proteins in their respective proteome.

**C**. Number of proteins belonging to coordinated molecular patterns (in Panel **A**) quantified in small EVs (100,000 g ultracentrifuged EVs 100 K) or large EVs (10,000 g ultracentrifuged EVs 10 K) from non-transformed cells (fibroblasts). Lower panel shows distribution of protein abundance of these molecular features against all other proteins in their respective proteome. **D**. Number of proteins belonging to coordinated molecular patterns (in Panel **A**) quantified in small EVs from 14 cell lines as reported[30]. Lower panel shows relative abundance of these molecular features in EVs versus their parental cell proteomes. Boxplot show the median (centre line), 25th-75th percentiles (box), minima and maxima within 1.5× interquartile range (whiskers), and outliers beyond.

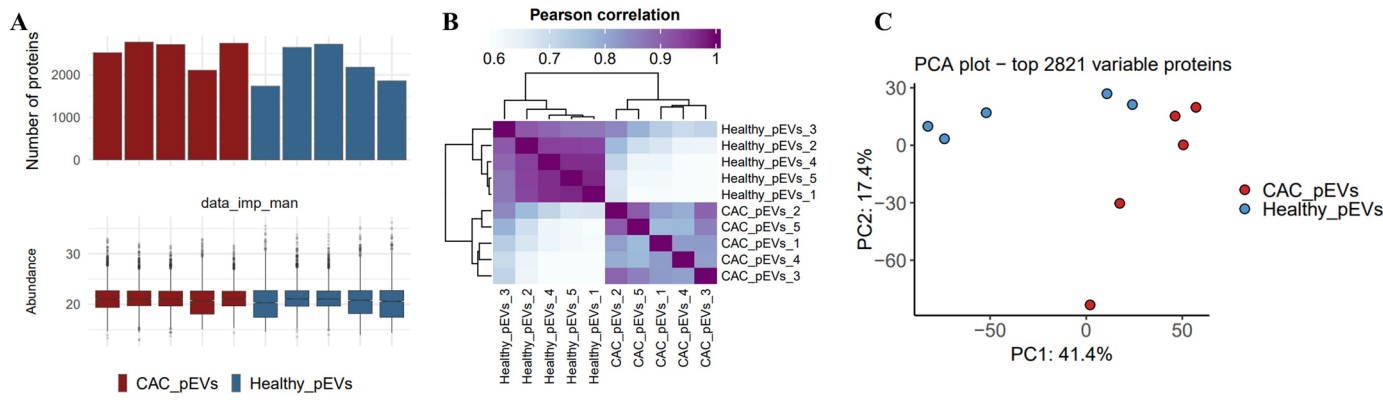

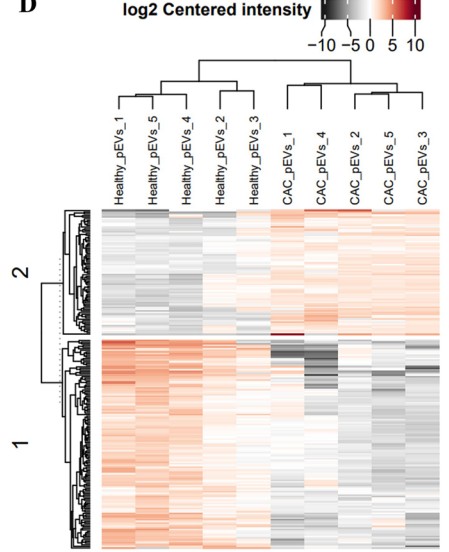

| Description (CAC_UP) | Genes | FDR q-value |
|---|---|---|
| Abnormal cardiovascular system physiology | 20 | 3.93E-09 |
| Abnormality of blood and blood-forming tissues | 16 | 1.83E-05 |
| Respiratory insufficiency | 11 | 3.98E-05 |
| Abnormal metencephalon morphology | 13 | 4.08E-05 |
| Functional abnormality of the gastrointestinal tract | 14 | 4.08E-05 |
| Functional abnormality of the inner ear | 13 | 4.94E-05 |
| Hypertonia | 13 | 8.40E-05 |
| Abnormal myocardium morphology | 10 | 8.40E-05 |
| Abnormality of fluid regulation | 11 | 8.87E-05 |
| Abnormal fundus morphology | 14 | 9.78E-05 |

**Abnormal cardiovascular system physiology**

| | | | | |
|---|---|---|---|---|
| ACAD8 | CST3 | MAN2B1 | SEC63 | TOR1A |
| ACTB | DDX3X | MAP2K1 | SLC25A4 | TPM1 |
| CCM2 | GMPPA | PPA2 | STX5 | TPM2 |
| CHCHD2 | HLA-B | SBDS | TAOK1 | TXNRD2 |

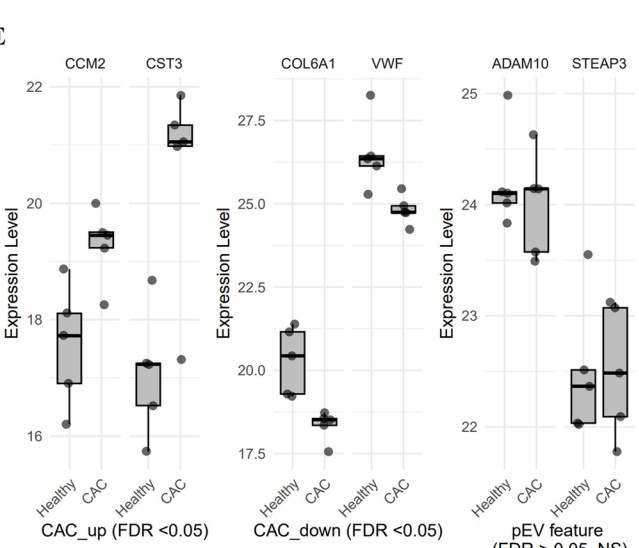

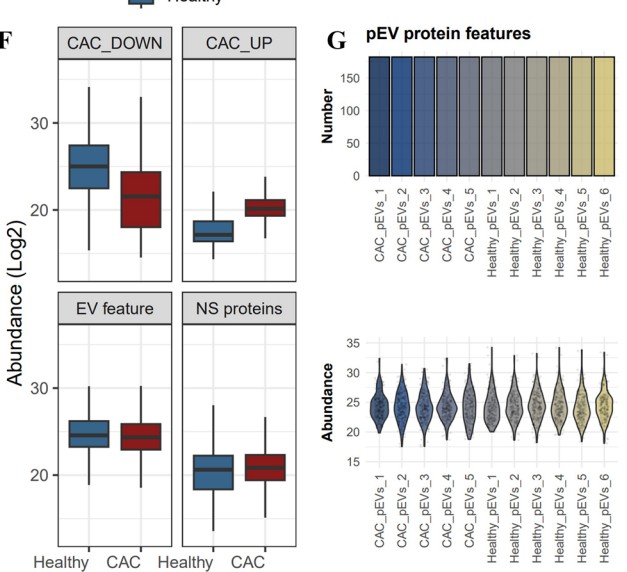

**Extended Data Fig. 7 | See next page for caption.**

**Extended Data Fig. 7 | Proteome analysis of pEVs from individuals with either positive CAC score (CAC group) or zero CAC score (Healthy group).** **A.** Proteins quantified in CAC_pEVs or Healthy_pEVs (n = 5 independent plasma samples in each group). Lower panel indicates normalized proteins abundance (vsn-normalized). Boxplots show the median (centre line), 25th-75th percentiles (box), minima and maxima within 1.5× interquartile range (whiskers), and outliers beyond. **B.** Pearson correlation of quantified proteins. **C.** Principal component analysis of quantified proteins. **D.** Heatmap depicting K-means clustering of differentially abundant proteins (FDR < 0.05) in CAC_pEVs versus Healthy_pEVs: 76 upregulated and 127 downregulated proteins (FDR < 0.05), with GO pathways enriched in upregulated proteins. Proteins of the GO pathway 'Abnormal cardiovascular system physiology' are shown. **E.** Boxplot showing vsn-normalizsed abundance (log2) of indicated proteins in CAC_pEVs and Healthy_pEVs proteomes (n = 5 independent plasma samples in each group). **F.** Boxplot showing distribution of either significantly upregulated (CAC_UP) or down regulated (CAC_DOWN) proteins, or non-significantly dysregulated proteins (NS proteins), and EV proteins features (EV feature) in CAC_pEVs and Healthy_pEVs proteomes (n = 5 independent plasma samples in each group). Boxplots in **E-F** show the median (centre line), 25th-75th percentiles (box), minima and maxima within 1.5× interquartile range (whiskers), and outliers beyond. **G.** pEV protein features quantified in individual CAC_pEVs versus Healthy_pEVs proteomes (n = 5 independent plasma samples in each group). Lower panel shows distribution of pEV protein features in individual proteomes. Violin plot show the distribution of vsn-normalized protein abundance (individual points are overlaid).

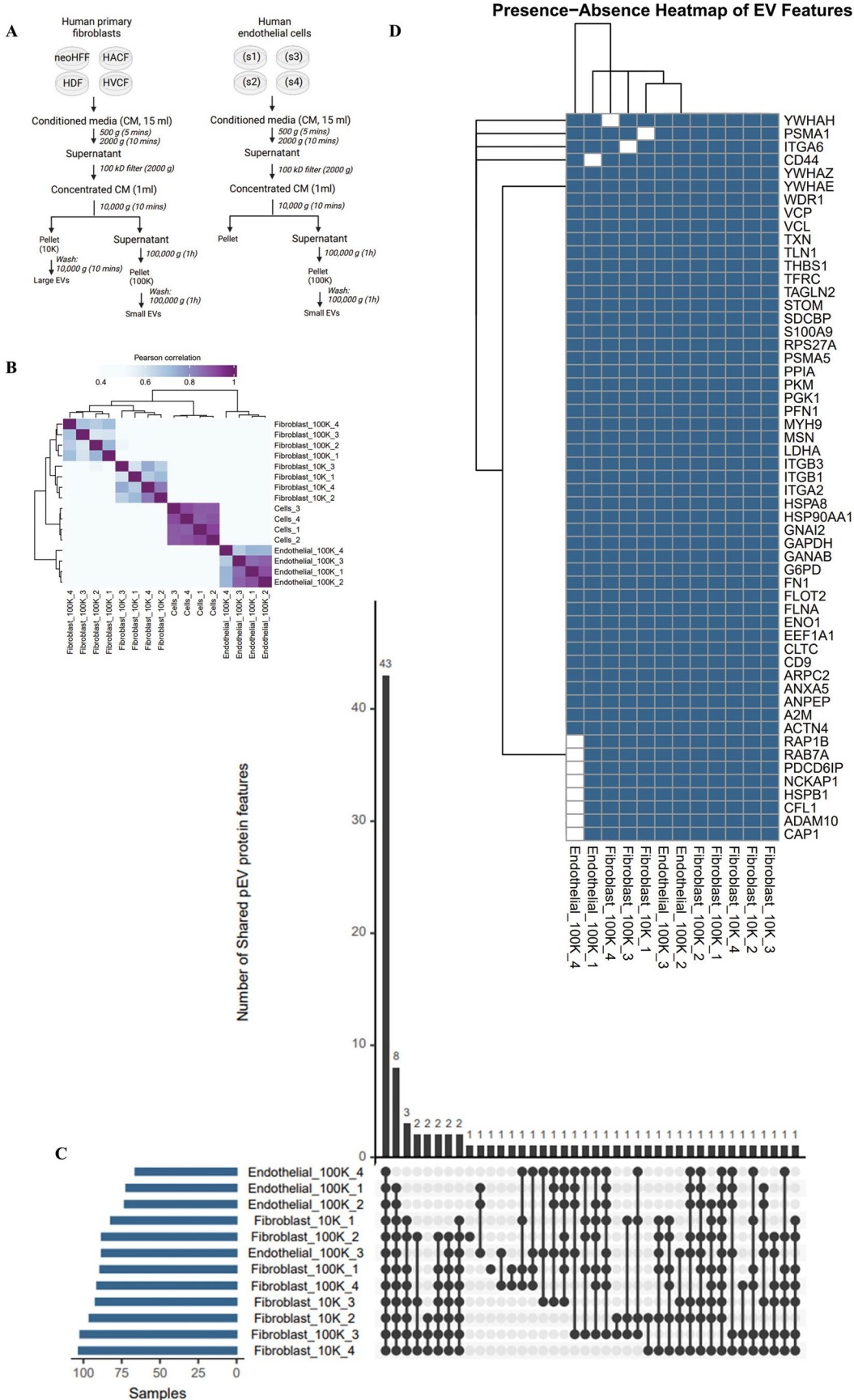

**Extended Data Fig. 8 | See next page for caption.**

**Extended Data Fig. 8 | Conservation of pEV protein features in EVs from non-transformed cells. A**. Workflow for obtaining small EVs (100 K) or large EVs (10 K) from conditioned media of either human primary fibroblasts or human primary endothelial cells. **B**. Pearson correlation of quantified proteins in indicated proteomes using mass spectrometry. Fibroblasts (whole cell lysates) were used as reference proteome for cells. **C**. pEV protein features quantified in small EVs (100 K) or large EVs (10 K) released by fibroblasts or endothelial cells. **D**. Heatmap indicates detection of pEV protein features in small EVs (100 K) or large EVs (10 K) released by fibroblasts or endothelial cells.

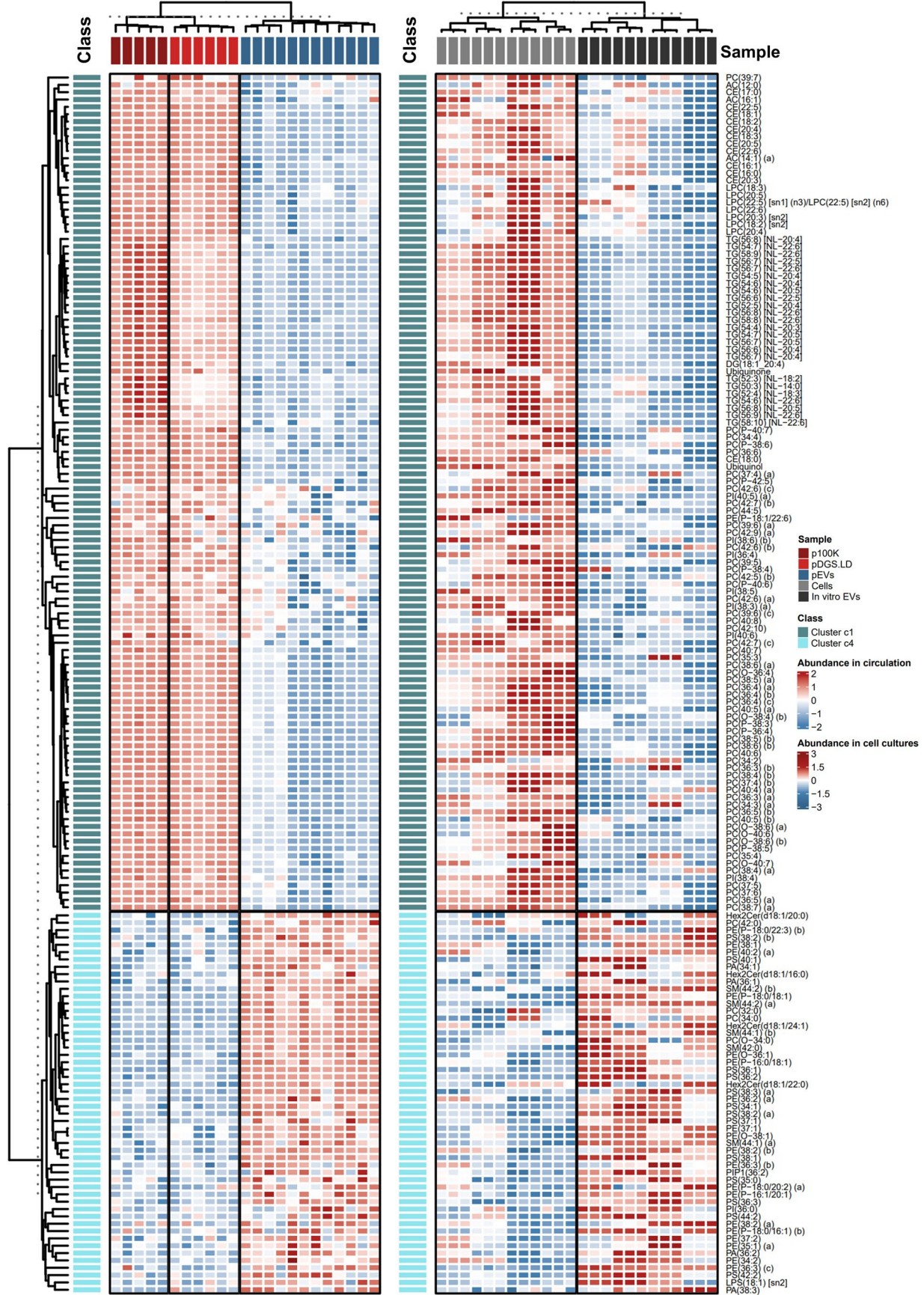

**Extended Data Fig. 9 | Heatmap of EV cluster 1 and cluster 4 lipids in EVs and NonEVs.**

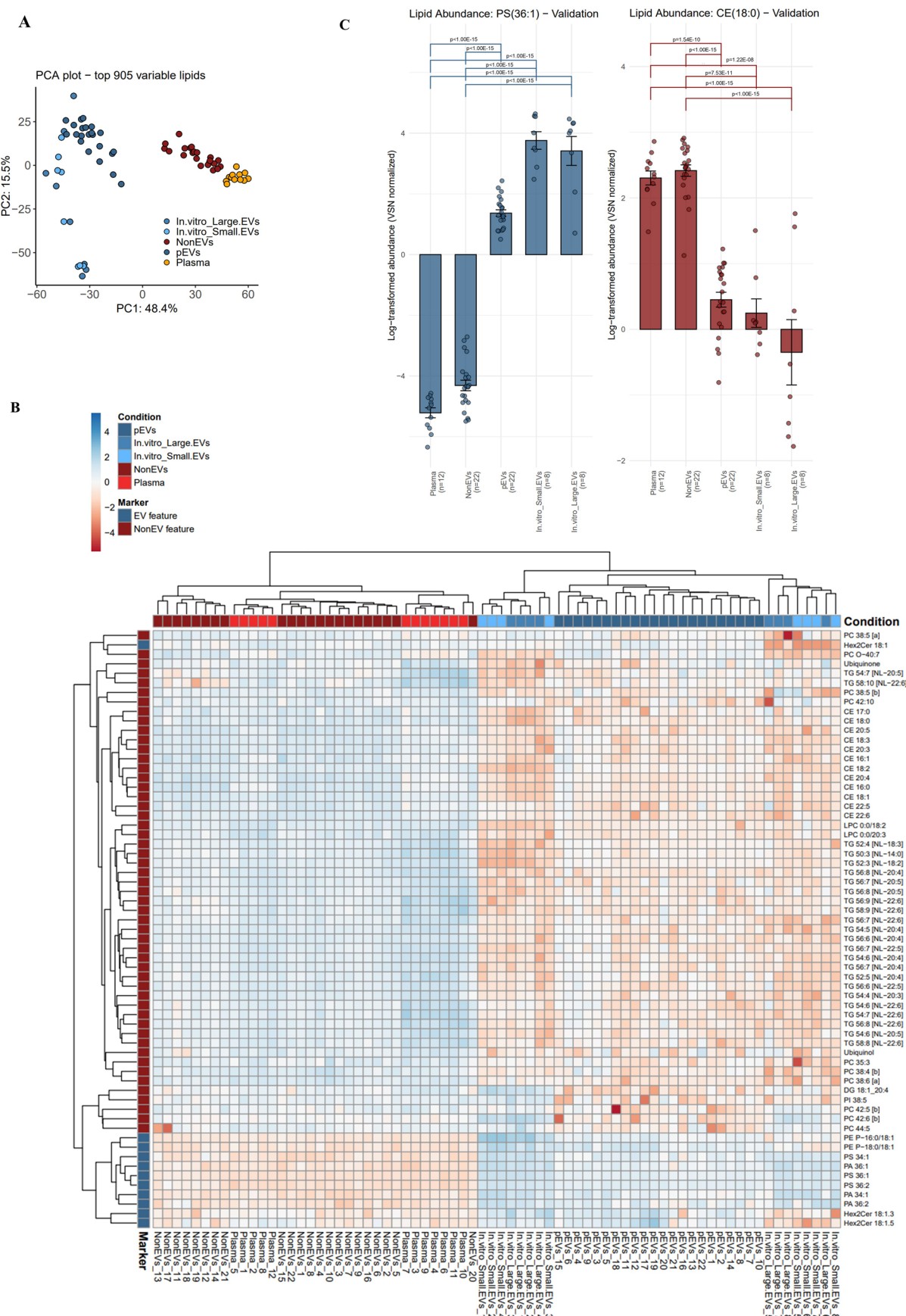

**Extended Data Fig. 10 | See next page for caption.**

**Extended Data Fig. 10 | Conservation of pEV lipid features in EVs from AusDiab validation set and in non-transformed cells. A**. Principal component analysis of quantified lipids in non-transformed EVs (in vitro Small EVs or in vitro Large EVs) or pEVs, NonEVs or neat plasma from AusDiab validation set. **B**. Heatmap depicts distribution of pEV and NonEV lipid features quantified in AusDiab validation set (pEVs, NonEVs or neat plasma) and in non-transformed cell-derived EVs (in vitro Small EVs or in vitro Large EVs). **C**. Relative abundance of pEV lipid features PS(36:1) and CE(18:0) in pEVs lipidome datasets. Lipid features in AusDiab validation set (pEVs, NonEVs or neat plasma) and in non-transformed cell-derived EVs (in vitro Small EVs or in vitro Large EVs) (n represents number of independent plasma samples/ biological replicates in each group). Bars show the group mean vsn-normalized abundance; error bars denote the standard error of the mean. Points are individual samples. One-way ANOVA was performed across groups on normalized values, followed by Tukey's multiple comparisons test (two-sided) to correct for multiple hypothesis testing. Adjusted P values (q values) are reported. Significance thresholds were set at FDR < 0.05.

# Reporting Summary

## Statistics

For all statistical analyses, confirm that the following items are present in the figure legend, table legend, main text, or Methods section.

| n/a | Confirmed |
|-----|-----------|
| ☐ | ☒ The exact sample size (*n*) for each experimental group/condition, given as a discrete number and unit of measurement |
| ☐ | ☒ A statement on whether measurements were taken from distinct samples or whether the same sample was measured repeatedly |
| ☐ | ☒ The statistical test(s) used AND whether they are one- or two-sided<br>*Only common tests should be described solely by name; describe more complex techniques in the Methods section.* |
| ☒ | ☐ A description of all covariates tested |
| ☐ | ☒ A description of any assumptions or corrections, such as tests of normality and adjustment for multiple comparisons |
| ☐ | ☒ A full description of the statistical parameters including central tendency (e.g. means) or other basic estimates (e.g. regression coefficient) AND variation (e.g. standard deviation) or associated estimates of uncertainty (e.g. confidence intervals) |
| ☒ | ☐ For null hypothesis testing, the test statistic (e.g. *F*, *t*, *r*) with confidence intervals, effect sizes, degrees of freedom and *P* value noted<br>*Give P values as exact values whenever suitable.* |
| ☒ | ☐ For Bayesian analysis, information on the choice of priors and Markov chain Monte Carlo settings |
| ☐ | ☒ For hierarchical and complex designs, identification of the appropriate level for tests and full reporting of outcomes |
| ☐ | ☒ Estimates of effect sizes (e.g. Cohen's *d*, Pearson's *r*), indicating how they were calculated |

*Our web collection on statistics for biologists contains articles on many of the points above.*

## Software and code

Policy information about availability of computer code

| Data collection | No software other than custom software or code was used for data collection. Software for data collection included Xcalibur software v4.0, MassHunter Acquisition v10, ZetaView (version 8.06.01 SP1), Cytek Aurora SpectroFlo (vS3.3.0) |
|-----------------|---|
| Data analysis | No software other than custom software or code used for data collection. Software for data analysis included, GraphPad Prism software v.8.1.2/9.3.0, Image J (NIH Image Software), DIA-NN (v1.8). MaxQuant (v1.6.6.0-v1.6.14), MassHunter Quantitative Analysis 9.0, Cytek Aurora SpectroFlo (vS3.3.0), Microsoft Excel 2019 (Office365). |

For manuscripts utilizing custom algorithms or software that are central to the research but not yet described in published literature, software must be made available to editors and reviewers. We strongly encourage code deposition in a community repository (e.g. GitHub). See the Nature Portfolio guidelines for submitting code & software for further information.

## Data

Policy information about availability of data

All manuscripts must include a data availability statement. This statement should provide the following information, where applicable:
- Accession codes, unique identifiers, or web links for publicly available datasets
- A description of any restrictions on data availability
- For clinical datasets or third party data, please ensure that the statement adheres to our policy

Data generated or analyzed during this study are included in this published article (and its supplementary information files), source data or available from Data Repositories. Lipidomics data are available from the NIH Common Fund's National Metabolomics Data Repository (NMDR) website (in addition to quantiitittaive details in Supplementary Tables 31,40). Proteomics data are available from the ProteomeXchange Consortium. All MS-based proteomics data (including sample/label annotation) is deposited to the ProteomeXchange Consortium via the MASSive partner repository and available via MASSive with identifier (MSV000094307).

Hierarchical clustering was performed in Perseus using Euclidian distance and average linkage clustering, with missing values imputed at z-score 0. Proteome and lipidome data sets were analyzed using R package Differential Enrichment analysis of Proteomics data (DEP). Using DEP, the data was background corrected and normalized by variance stabilizing transformation (vsn, which also log2-transforms the data), followed by imputation of missing values, whereby missing at random data were 'knn' imputed and missing not at random data were 'MinProb' imputed. Protein-wise linear models combined with empirical Bayes statistics were used for the differential enrichment analysis, whereby the raw p-values were adjusted to correct for multiple testing using Benjamini-Hochberg method. Differentially abundant proteins or lipids were clustered by k-means clustering using DEP package. The PCA plot, Pearson Correlation matrix, volcano plots, log2 centred bar plots and overlap bar plots were also generated using DEP. Heatmaps were generated using ComplexHeatmap package (https://bioconductor.org/packages/release/bioc/html/ComplexHeatmap.html). Box plots and scatter plots were generated using RStudio package ggplot2 (https://rss.onlinelibrary.wiley.com/doi/abs/10.1111/j.1467-985X.2010.00676_9.x). Cytoscape was used to generate Ontology map (plugin v3.7.1). Bioconductor package clusterProfiler 4.023 was used to perform Ontology or KEGG pathway enrichment analysis, or gene set enrichment (KEGG) analysis, with default parameters used to identify significantly enriched gene sets. The pathway-based data integration and visualization was constructed using R package pathview. For identification of lipid-associated terms enriched in lipidomes, LION web-based ontology enrichment tool was used (www.lipidontology.com). For annotating surface proteins into different categories, SURFY-based categorical annotation of cell surface proteins were employed (https://wlab.ethz.ch/surfaceome/). We employed caretEnsemble R package to assess the ability of protein features (pEV protein features and NonEV protein features, using Naïve Bayes algorithm), surface protein features or lipid features, using Neural Network algorithm ('nnet')) to distinguish between distinguish between pEV and NonEV particles. The Shiny web application (https://evmap.shinyapps.io/evmap/) powered by R and hosted on shinyapps.io, was created using the R packages shiny, gplots, and ComplexHeatmap, and offers feature selections and visualizations for EV protein and lipid feature conservations in circulating EVs.

# Field-specific reporting

Please select the one below that is the best fit for your research. If you are not sure, read the appropriate sections before making your selection.

☒ Life sciences  ☐ Behavioural & social sciences  ☐ Ecological, evolutionary & environmental sciences

For a reference copy of the document with all sections, see nature.com/documents/nr-reporting-summary-flat.pdf

# Life sciences study design

All studies must disclose on these points even when the disclosure is negative.

| | |
|---|---|
| Sample size | Proteomic experiment/file annotation is provided in ProteomeXchange Consortium via the MASSive partner repository and available via MASSive with identifier (MSV000094307) which details the number of biological replicates for each experimental condition (at least n=3). We have further included sample annotation for all proteomic and lipidomic data acquisition in this study in Source Data 3. |
| Data exclusions | All data exclusions are outlined in the methods, legends and/or Source Data/Supplementary information. |
| Replication | For all studies, groups and analyses contained at least three independent biological samples. |
| Randomization | For proteomic and lipidomic sample preparations, samples were blinded and later re annotated for data analysis. |
| Blinding | N/A |

# Reporting for specific materials, systems and methods

We require information from authors about some types of materials, experimental systems and methods used in many studies. Here, indicate whether each material, system or method listed is relevant to your study. If you are not sure if a list item applies to your research, read the appropriate section before selecting a response.

## Materials & experimental systems

| n/a | Involved in the study |
|---|---|
| ☐ | ☒ Antibodies |
| ☐ | ☒ Eukaryotic cell lines |
| ☒ | ☐ Palaeontology and archaeology |
| ☒ | ☐ Animals and other organisms |
| ☐ | ☒ Human research participants |
| ☒ | ☐ Clinical data |
| ☒ | ☐ Dual use research of concern |

## Methods

| n/a | Involved in the study |
|---|---|
| ☒ | ☐ ChIP-seq |
| ☐ | ☒ Flow cytometry |
| ☒ | ☐ MRI-based neuroimaging |

## Antibodies

| | |
|---|---|
| Antibodies used | Details of all antibodies are provided within the data supplement. Albumin (ab207327, Abcam), AGO2 (ab186733, Abcam) were used. Mouse antibodies CD63 (556019, BD Pharmingen), CD81 (555675, BD Pharmingen), APOB100 (3715-3-250, Mabtech), APOA1 (3710-3-1000, Mabtech) were used (1:1000). For flow, Anti-ADAM 10 Antibody (Sigma-Aldrich, AB19026; 2 µg in 100 ul vol), Rabbit IgG Isotype Control (Invitrogen, 10500C) were used. Secondary antibodies used were IRDye 800 goat anti-mouse IgG or IRDye 700 goat anti-rabbit IgG (1:15000, LI-COR Biosciences). For flow, Goat Anti-Rabbit Alexa Fluor 568 Dye secondary antibody (A-11011, |

| Validation | Thermo Fisher Scientific) |
|---|---|

All antibodies used in this study have been validated by the manufacturer;
Albumin (ab207327, Abcam)
https://www.abcam.com/en-au/products/primary-antibodies/albumin-antibody-epr20195-ab207327

AGO2 (ab186733, Abcam)
https://www.abcam.com/en-au/products/primary-antibodies/argonaute-2-antibody-epr10411-ab186733

CD63 (556019, BD Pharmingen)
https://www.bdbiosciences.com/en-au/products/reagents/flow-cytometry-reagents/research-reagents/single-color-antibodies-ruo/purified-mouse-anti-human-cd63.556019

CD81 (555675, BD Pharmingen)
https://www.bdbiosciences.com/en-au/products/reagents/flow-cytometry-reagents/research-reagents/single-color-antibodies-ruo/purified-mouse-anti-human-cd81.555675

APOB100 (3715-3-250, Mabtech)
https://stella.mabtech.com/sites/default/files/product_datasheets/3715-3-250.pdf

APOA1 (3710-3-1000, Mabtech)
https://www.mabtech.com/api/files/product_datasheets/3710-3-1000.pdf

ADAM 10 (Sigma-Aldrich, AB19026)
https://www.sigmaaldrich.com/AU/en/product/mm/ab19026

Rabbit IgG Isotype (Invitrogen, 10500C)
https://www.thermofisher.com/antibody/product/Rabbit-IgG-Isotype-Control/10500C

# Eukaryotic cell lines

Policy information about cell lines

| Cell line source(s) | SW480 (CCL-288, ATCC), MDA MB 231 (HTB-26, ATCC), SW620 (CCL-227, ATCC)  were from American Type Culture Collection (ATCC, VA, USA), and LIM1863 cells from Ludwig Institute for Cancer Research, Melbourne. Primary human cell source include human neonatal foreskin fibroblast cell line (neoHFF) [sourced as gift from P Kaur, Monash University, Australia), human dermal fibroblasts (hDF), adult (Gibco/Thermo Fisher Sci. #C0135C), human atrial cardiac fibroblasts (haCF) (Lonza, #CC-2903) and human ventricular cardiac fibroblasts (Lonza, #CC-2904), human umbilical vein endothelial cells (HUVEC, Lonza, #CC-2519; HUVEC sourced as gift [K Peter, BHDI, Australia], HUVEC-RFP, Angio-Proteomie #cAP-0001) |
|---|---|
| Authentication | Cells were grown in specified media and supplements as per detailed in methods (generation of cell conditioned media) and published protocols, and cells authenticated in our previous publications (PMID: 30582284, PMID: 23585443, PMID: 23230278, PMID: 38938901). Commercially purchased HUVEC, hDF, haCF and MDA MB 231 cells were not authenticated. |
| Mycoplasma contamination | Cells were routinely tested for mycoplasma (neg) contamination by PCR assay annually. |
| Commonly misidentified lines (See ICLAC register) | This cell line is not registered on the list of misidentified cell lines. |

# Human research participants

Policy information about studies involving human research participants

| Population characteristics | There was no population characteristics for Red Cross plasma samples, EDCAD cohort, or AusDiab cohort (other than donation age 18 years+). |
|---|---|
| Recruitment | Red Cross plasma samples were obtained blinded as pooled plasma samples, and plasma samples from individual donors already collected in the EDCAD cohort and AusDiab cohort study used in this study. There was no criteria for participant recruitment. Clinical parameters for EDCAD plasma samples are provided in Supplementary Table 41. |
| Ethics oversight | Human blood plasma samples were obtained from Australian Red Cross Lifeblood or EDCAD study. For Red Cross samples, ethical permits were obtained from the ethical committee of the Australian Red Cross Blood Service Human Research Ethics Committee, and La Trobe University Human Ethics Committee (HEC19485). For EDCAD samples, ethics permit was approved through Human Research Ethics Committee (HREC) at Baker Heart and Diabetes Institute and by the Alfred Hospital Ethics Committee (EDCAD-PMS, #492/20). For AusDiab trial, ethics permit was approved through HREC at Baker Heart and Diabetes Institute and by the Alfred Hospital Ethics Committee (#39/11). |

Note that full information on the approval of the study protocol must also be provided in the manuscript.

# Flow Cytometry

## Plots

Confirm that:

☒ The axis labels state the marker and fluorochrome used (e.g. CD4-FITC).

☒ The axis scales are clearly visible. Include numbers along axes only for bottom left plot of group (a 'group' is an analysis of identical markers).

☒ All plots are contour plots with outliers or pseudocolor plots.

☒ A numerical value for number of cells or percentage (with statistics) is provided.

## Methodology

| | |
|---|---|
| Sample preparation | Plasma EVs (~5 μg) were subjected to fixation and permeabilization using eBioscience™ Foxp3 / Transcription Factor Staining Buffer Set (Invitrogen™, 00-5523-00). Briefly, pEVs (in 50 μl PBS) was incubated with 500 μl fixation and permeabilization buffer on ice for 30 mins. Samples were ultracentrifugation at 100,000 g (1 h at 4 °C) and pellets resuspended in 100 μl wash buffer. Samples were stained with 5 μl APC Annexin V reagent (BioLegend) and either 2 μg of Anti-ADAM 10 Antibody (Sigma-Aldrich, AB19026) or Rabbit IgG Isotype Control (Invitrogen, 10500C). Samples were incubated at room temperature (gentle end-over mixing) for 1h. Samples were topped with 900 μl wash buffer and washed twice (ultracentrifuged at 100,000 g (1 h at 4 °C)) to remove any remaining antibodies and potential antibody aggregates. The pellets resuspended in 100 μl wash buffer containing 0.5 μl of Goat Anti-Rabbit Alexa Fluor 568 Dye secondary antibody (ThermoFisher Scientific) incubated for 30 mins at room temperature (gentle end-over mixing) in dark. Samples were topped with 900 μl wash buffer and washed twice (ultracentrifuged at 100,000 g (1 h at 4 °C)). Pellets resuspended in 100 μl of PBS (0.5% bovine serum albumin) filtered using 0.22 μm filter |
| Instrument | Cytek Aurora flow cytometer |
| Software | SpectroFlo  (vS3.3.0) |
| Cell population abundance | N/A |
| Gating strategy | Instrument gating calibration was performed using 90 nm (#64009-15) 125 nm (#64011-15), 150 nm (#64012-15), 200 nm (#64013-15) and equal mix (90-200 nm) beads (Nanobead NIST Traceable Particle Size Standards). The threshold for side scatter was set to 430, and the gain of side scatter (SSC) were set to 10.  YG3-A channel used for detecting Alex Flour 568 signal, R1-A channel used for detecting APC signal and 10 000 events were recorded for all samples with the slowest flow rate to minimize the swarming effect. |

☒ Tick this box to confirm that a figure exemplifying the gating strategy is provided in the Supplementary Information.

