## [Peer Review File · Nature Cell Biology]

Multi-omics identify hallmark protein and lipid features of small extracellular vesicles circulating in human plasma

Corresponding Author: Professor David Greening

Version 0:

Decision Letter:

*Please delete the link to your author homepage if you wish to forward this email to co-authors.

Dear Professor Greening,

Thank you again for submitting your manuscript, "Multi-omics discovery of hallmark protein and lipid features of circulating small extracellular vesicles in humans", to Nature Cell Biology. It has now been seen by 3 referees, who are experts in EVs (Referee #1); EVs, mass spectrometry (Referee #2); and lipidomics (Referee #3). As you will see from their comments (attached below), they found this work of potential interest but have raised substantial concerns, which in our view would need to be addressed with considerable revisions before we can consider publication in Nature Cell Biology.

Nature Cell Biology editors discuss the referee reports in detail within the editorial team, including the chief editor, to identify key referee points that should be addressed with priority, and requests that are overruled as being beyond the scope of the current study. To guide the scope of the revisions, I have listed these points below. Our standard revision period is six months, and we are committed to providing a fair and constructive peer-review process, so please feel free to contact me if you would like to discuss any of the referee comments further.

The referees' concerns are significant and in our view would need to be addressed thoroughly experimentally; reconsideration of the study for this journal and re-engagement of the referees will depend on the strength of these revisions. We recommend revisions addressing the following reviewer points:

1- The reviewers had significant reservations about the analysis and workflows that need to be addressed in full:

Rev#1 points #3, 5, 6, 7, 8
Rev#2 points #1, 2, 4, 5, 8
Rev#3 points #1-2-3

2- The reviewers additionally requested validation studies to confirm the conclusions:

Rev#1 point #4
Rev#2 points #3, 4, 7
Rev#3 point #4

3- Please also address their technical comments, requests for controls and strengthening of existing data, minor points, requests for discussion and addition of text or text/figure changes/edits.

4- Finally, please pay close attention to our guidelines on statistical and methodological reporting (listed below) as failure to do so may delay the reconsideration of the revised manuscript. In particular, please provide:

We would be happy to consider a revised manuscript that would satisfactorily address these points, unless a similar paper is

published elsewhere or is accepted for publication in Nature Cell Biology in the meantime.

- ensure that it conforms to our format instructions and publication policies (see below and <https://www.nature.com/nature/for-authors>).
- provide a point-by-point rebuttal to the full referee reports verbatim, as provided at the end of this letter.
- provide the completed Reporting Summary (found here <https://www.nature.com/documents/nr-reporting-summary.pdf>). This is essential for reconsideration of the manuscript will be available to editors and referees in the event of peer review. For more information see <http://www.nature.com/authors/policies/availability.html> or contact me.

Nature Cell Biology is committed to improving transparency in authorship. As part of our efforts in this direction, we are now requesting that all authors identified as 'corresponding author' on published papers create and link their Open Researcher and Contributor Identifier (ORCID) with their account on the Manuscript Tracking System (MTS), prior to acceptance. ORCID helps the scientific community achieve unambiguous attribution of all scholarly contributions. You can create and link your ORCID from the home page of the MTS by clicking on 'Modify my Springer Nature account'. For more information please visit www.springernature.com/orcid.

This journal strongly supports public availability of data. Please place the data used in your paper into a public data repository, or alternatively, present the data as Supplementary Information. If data can only be shared on request, please explain why in your Data Availability Statement, and also in the correspondence with your editor. Please note that for some data types, deposition in a public repository is mandatory - more information on our data deposition policies and available repositories appears below.

Link Redacted

We hope that you will find our referees' comments and editorial guidance helpful. Please do not hesitate to contact me if there is anything you would like to discuss. Thank you again for considering the journal for your work.

Best wishes,

Melina

Melina Casadio, PhD
Senior Editor, Nature Cell Biology
ORCID ID: <https://orcid.org/0000-0003-2389-2243>

Reviewers' Comments:

Reviewer #1:

Remarks to the Author:

The manuscript by Rai et al describes authors' formidable effort to comprehensively and systematically catalogue and characterize the proteome and lipidome of the particulate content of human plasma, with special distinction of extracellular vesicles (pEVs) and nonEV particles (pNVPs). The authors collected a large cohort of 110 human plasma samples from

several centres and adopted the high-resolution iodixanol based density gradient separation (DGS) to fractionate the respective particles. The fractions enriched in pEVs and pNVP were then processed for mass spectrometry which yielded approximately 4500 pEV proteins and 829 lipids. Of those, some were highly conserved in pEVs (182 and 52 correspondingly). The authors also performed surfaceome analysis resulting in validation of external membrane proteins. They also performed TMT proteomics for co-expression studies. Pathway analysis revealed that pEV cargo was reassuringly enriched in components of the endosomal system and lipid rafts, while pNVPs contained elements of the complement and coagulation cascades. Notably, machine learning analysis of the profiling output across the samples pointed to ADAM10 as a highly enriched protein (marker) of pEVs, possibly usable for EV purity assessments, along with phosphatidyl serine [PS(36:1)]. The rich database generated by this study served to develop an open-source R/Shiny web tool for the benefit of the EV research community.

Overall, this is a data-rich, systematic and solid benchmarking study of considerable value for those with interest in EV biology, liquid biopsy, analysis of plasma components along with the related fields. The study provides a valuable resource in terms of data, functional pipeline and a R/Shiny web tool. It also addresses long standing needs in terms of baseline plasma EV and NVP characterization, something that may inspire similar efforts in the context of other biofluids where these particles are being sampled and studied. The analysis appears to be thoroughly executed and showcases the expertise of the authors, who are leaders in EV proteomics. The manuscript is also well written, well documented and informative.

Although this is, in many ways, a potentially important, even foundational contribution to the EV research field, there several considerable weaknesses in light of this high profile submission.

1. The paper represents an important scaled up version of prior studies on plasma particles (this is valuable), but it contains relatively little conceptual or experimental novelty. The power of (DGS) in separating EVs from NVP has already been demonstrated (Jeppesen et al 2019), proteomes of plasma and cellular EVs have been extensively studied (Hoshino et al 2020), and protein hubs for EV assembly have been explored to some extent (Luck et al 2020; Kugeratski et al 2021). Properties, numbers and some of the markers of plasma EVs have also been described, as cited by the authors (e.g. Arraud et al 2014). While the stringency with which ADAM10 and PS associate with plasma EVs in the present study is impressive, both of these molecular species have already been found in EVs (Vesiclepedia) with PS having long been used for preparative purposes, even commercially.
2. As such, the manuscript offers a relatively deep, but rather descriptive overview of plasma EV (and NVP) proteome and lipidome and is not asking any specific, or novel biological questions. There is room for this kind of benchmarking, of course, but defining "core" proteome and lipidome says very little about biological properties, activities, functionalities, uptake and other aspects of EVs and NVPs that authors declare as their main objective.
3. It is also presently unclear whether and how the parameters defined by the authors would change and in the context of disease such as inflammation, cardiovascular disease, sepsis, trauma, cancer, thrombosis or other states where plasma EV analysis is already underway. This uncertainty weakens the appeal of the present study by comparison to previously published efforts (e.g. Hoshino et al 2020). How would EV profiles, cargo and 'markers' change under pathological conditions, age, sex and other variables? By compiling samples from different sources all these important variables that would be critical, for example, to understand individualised physiological conditions or patient care become obscured. How would the presently defined bulk 'baseline' help guide future biomarker studies?
4. By the authors' own admission, the present analysis of 'bulk' EV or NVP preparations suppresses the enormous heterogeneity of these particles, not only in terms of the officially recognized 'subtypes', such as exosomes, ectosomes, or their variants (e.g. ARMMs) but also the natural variability within each of these categories, including at the single vesicle level. Important functionalities of small EV subsets could be lost in this process. In fact, it would be of interest to assess whether ADAM10 is expressed in all EVs, or in some EVs but in all plasma samples, and what would either mean.
5. Moreover, plasma EVs come from multiple cellular sources, such as platelets, blood cells, endothelial cells and other compartments. In the present study they are all lumped together in spite of the fundamentally different biological properties of EVs originating from different cells. This is surprising, as high-resolution proteome should allow, at the very least, to distinguish between the signatures of EVs from specific cellular sources, which would be of considerable value and interest. It could be argued that seeking predictable commonalities obscures the molecular determinants of specific EV activities.
6. What is the ability of the mass spectrometry described by the authors to distinguish low abundance, highly bioactive entities, such as growth factor receptors, signal transducing molecules, transcription factors, RNA binding proteins (nothing is said in the paper about circulating RNA) and other important parts of the EV cargo?
7. One of the important sources of the molecular 'noise' among plasma samples and roadblocks in EV/NVP standardization are the pre-analytical variables, especially techniques that prevent activation of platelets during blood collection, preparation and storage (Coumans et al 2017). This is very difficult to achieve when operating across multiple institutions. The present study lacks quality controls as to elimination of activated platelets or induced platelet EVs in samples being analysed, which may result in the reported profiles being essentially those of contaminating platelets.
8. Fig 2F shows the molecular interconnectedness of molecular elements identified within the plasma EV cargo. This would make sense if the respective molecules interacted within the same EV, or at least within the same EV parental cells, which remains undefined in the study. What is the significance of these connections if plasma EVs come from multiple cellular sources?

Reviewer #2:

Remarks to the Author:

Comments to the Authors:

In this manuscript, the authors perform mass spectrometry based proteomic and lipidomic analysis of a large number of extracellular vesicles (EVs) samples isolated from human plasma samples via density gradient ultracentrifugation (110 samples). Signatures of highly conserved proteins and lipids as well as proteins associated with the surface of sEVs were identified. The authors also propose ADAM10 and PS(36:1) as markers to distinguish EVs from non-EVs. The authors provide a comprehensive dataset of the proteome and lipidome of healthy plasma EVs which has not previously been described at this scale and with isolation methods enabling such purity. Several comments should be addressed, as outlined below.

1. The authors isolate EVs with density gradient (DG) wherein plasma is loaded on the top of the gradient; however, this can in some cases lead to incomplete separation of sEVs from 'contaminating' proteins, lipids, etc. The authors should confirm by bottom loaded DG that the identified enriched protein and lipid markers are present.

2. Comparisons between healthy plasma sEVs to sEVs from cell lines are included, but the cell lines employed are predominantly cancer cell lines. The authors should address whether there is greater concordance with non-transformed cell lines, or the features are conserved between non-transformed/healthy and cancer cell-derived EVs. Similarly, the only disease condition included is ECAD; thus, conclusions about applicability to disease states should be tempered.

3. Upon identifying the 182 pEVs signatures, the abundance differences between pEVs and nonEVs need to be considered in addition to the p-values. Although certain proteins exhibit significant p-values, their abundance differences are trivial, casting doubt on their classification as pEV signatures.

4. The identified pEV markers require experimental validation of their abundance in external cohorts.

5. It is not clear why the authors specifically employ Naïve Bayes for proteins and Neural Network for lipids. Did the authors compare the performance of these models to other existing models?

6. It is not necessary to employ machine learning to identify the pEV markers in this study as we can obtain the same information based on comparison analysis. For example, ADAM10 is one of the proteins exclusively detected in pEV samples as shown in Figure 3.

7. For the identified pEV markers, the authors need to explain why they were not universally identified in other studies and the potential impact of differences in the techniques on EV isolation. For example, ADAM10 was not universally detected in both human plasma and cell EV samples in the Hoshino et al study (Cell, 2020).

8. In Fig. 2D, Extended data 2, Supplementary Table 9, amongst the 259 conserved proteins, the authors further distinguish 182 proteins present uniquely or in significantly higher abundance in pEVs compared to NonEVs and was named as "pEV protein features". The criteria or threshold to distinguish these 182 proteins should be included. Similarly, in Extended data 3A, criteria- 2) proteins quantified in over 70% of NonEV samples but less than 30% of pEVs samples (14 proteins) will be included in "NonEV protein features", but there was no clear dividing line that highlighted 70% of NonEV samples and 14 proteins.

Reviewer #3:

Remarks to the Author:

Manuscript i.d.: NCB-A53707

In this manuscript, Greenings and coworkers describe the mass spectrometry-based proteomic and lipidomic analyses of the Extracellular vesicles (EVs) in human plasma. They first performed high-resolution density gradient fractionation of over 110 human plasma samples to isolate circulating EVs, and systematically construct their quantitative proteome (4500 proteins) and lipidome (829 lipids) landscapes. The analysis led them to believe that a highly conserved panel of 182 proteins (ADAM10, STEAP23, STX7) and 52 lipids (PS, PIPs, Hex2Cer, PAs) were discovered. They also claim that a set of 42 proteins and 114 lipids features that served as hallmark features of non-EV particles in plasma have been established. Finally, they pinpoint that PS(36:1) and CE(18:0) are the conserved EV biological markers that precisely differentiates between EV and non-EV particles

This is a gigantic work, containing tons of data for readers to digest; and the goal of the study is ambitious. Frankly, I am overwhelmed by the data that have been presented. Considering myself as an experienced researcher in the lipid field (less is proteomics), it is more appropriate that my critique on this manuscript be focused on the lipidomic analysis, rather than the entire content (e.g., not comment on the proteomic analysis). My comments are listed below.

1. Is the claim of the lipidome (829 lipids, 40 lipid classes) constructed credible (big number is not necessary good)? The LC/MS method as applied by the authors (Agilent QQQ instrument for qualitative and quantitative analysis) appeared to have driven the instrument to the brink of mission impossible. It is not likely that a single LC/MS analysis can result in such a huge number of lipid species, leaving alone a precise quantitation. For tissue lipidome such as plasma, the intensities of lipid species varied drastically among various lipid families and various lipid species in the same lipid family. Are they in the

linear range for quantitation. How confident is the structure identification?

2. The designated lipid species as indicated in nearly the entire paper using, for example, PS(36:1), which only defines the total chain length of FA substituents and their total number of unsaturation. Therefore, isomeric structures, such as isomers with various chain length, double bond location, and region specificity of the fatty acid substituents has been neglected, and thus, significantly reduce the number of lipid species. Therefore, the number (829) identified sounds too large to me.

3. In mass spectrometry, relative intensity (or intensity) for each ion in a complete lipidome is readily available. Although it becomes common practice in the literatures using heat map for presentation, it is just way too ambiguous. Present the key data as “numbers” would be very helpful for readers.

4. To be able to use proteomics and lipidomics assay, in combination with computer software for data handling to analyze a large number of proteins and lipids, and their interactions to pinpoint that PS(36:1) and CE(18:0) are the conserved EV biological markers that differentiates between EV and non-EV particles is an achievement. However, I am surprised by the fact that the detailed structure of PS(36:1) has not been characterized. It would be straightforward to use tandem mass spectrometry to define the structure, including the identity of the fatty acid chains and their position on the glycerol bond, and even the location of double bond. It is also desirable that the structure of CE(18:0) be verified and quantified.

Other comments

1. I am not sure that I know what “glycophosphoserines” is. Is that “glycerophosphoserine” instead? Are you sure its presence in the lipidome?

2. “equipped with nanospray ion source in positive mode” should be “positive ion mode”

3. Reference(s) are required for Table 1. “List of EV and NonEV protein features” and Table 2. “List of EV and NonEV lipid features”.

4. Supplementary Figure 7. Lipidome profiling of circulating EVs is confusing.

5. “Supplementary Figure 8. Lipid species co-enriched in circulating EVs and EVs from cell culture” is also confusing. For example, what does “TG (54:6)[NL-22:6]” mean? There is another TG (54:6) [NL-20:4] in the listing. Are they isomers? I would believe there are more than two isomers. What are the exact structures? I think it is much clearer that the molecular ions (e.g., [M+H]⁺, or other adduct ions such as NH₄⁺ in the positive ion; or [M – H]⁻ or other adduct ions such as [M+ Cl]⁻ in the negative ion mode) be presented, followed by their elemental compositions, and then the structures and the relative intensity. So are the other lipid families and the individual lipid species.

6. The Supplementary Tables have no content (only Table Legends are shown). Thus, there is no way to check (validate) the authenticity of the lipidome as claimed.

AUTHOR AFFILIATIONS – should be denoted with numerical superscripts (not symbols) preceding the names. Full addresses should be included, with US states in full and providing zip/post codes. The corresponding author is denoted by: “Correspondence should be addressed to [initials].”

ABSTRACT AND MAIN TEXT – please follow the guidelines that are specific to the format of your manuscript, as listed in our Guide to Authors (http://www.nature.com/ncb/pdf/nbc_gta.pdf) Briefly, Nature Cell Biology Articles, Resources and Technical Reports have 3500 words, including a 150 word abstract, and the main text is subdivided in Introduction, Results, and Discussion sections. Nature Cell Biology Letters have up to 2500 words, including a 180 word introductory paragraph (abstract), and the text is not subdivided in sections.

ACKNOWLEDGEMENTS – should be kept brief. Professional titles and affiliations are unnecessary. Grant numbers can be

listed.

Methods should be written concisely, but should contain all elements necessary to allow interpretation and replication of the results. As a guideline, Methods sections typically do not exceed 3,000 words. The Methods should be divided into subsections listing reagents and techniques. When citing previous methods, accurate references should be provided and any alterations should be noted. Information must be provided about: antibody dilutions, company names, catalogue numbers and clone numbers for monoclonal antibodies; sequences of RNAi and cDNA probes/primers or company names and catalogue numbers if reagents are commercial; cell line names, sources and information on cell line identity and authentication. Animal studies and experiments involving human subjects must be reported in detail, identifying the committees approving the protocols. For studies involving human subjects/samples, a statement must be included confirming that informed consent was obtained. Statistical analyses and information on the reproducibility of experimental results should be provided in a section titled "Statistics and Reproducibility".

All Nature Cell Biology manuscripts submitted on or after March 21 2016 must include a Data availability statement as a separate section after Methods but before references, under the heading "Data Availability". For Springer Nature policies on data availability see <http://www.nature.com/authors/policies/availability.html>; for more information on this particular policy see <http://www.nature.com/authors/policies/data/data-availability-statements-data-citations.pdf>. The Data availability statement should include:

- Accession codes for primary datasets (generated during the study under consideration and designated as "primary accessions") and secondary datasets (published datasets reanalysed during the study under consideration, designated as "referenced accessions"). For primary accessions data should be made public to coincide with publication of the manuscript. A list of data types for which submission to community-endorsed public repositories is mandated (including sequence, structure, microarray, deep sequencing data) can be found here <http://www.nature.com/authors/policies/availability.html#data>.
- Unique identifiers (accession codes, DOIs or other unique persistent identifier) and hyperlinks for datasets deposited in an approved repository, but for which data deposition is not mandated (see here for details <http://www.nature.com/sdata/data-policies/repositories>).
- At a minimum, please include a statement confirming that all relevant data are available from the authors, and/or are included with the manuscript (e.g. as source data or supplementary information), listing which data are included (e.g. by figure panels and data types) and mentioning any restrictions on availability.
- If a dataset has a Digital Object Identifier (DOI) as its unique identifier, we strongly encourage including this in the Reference list and citing the dataset in the Methods.

We recommend that you upload the step-by-step protocols used in this manuscript to the Protocol Exchange. More details can be found at www.nature.com/protocolexchange/about.

All imaging data should be accompanied by scale bars, which should be defined in the legend.

Cropped images of gels/blots are acceptable, but need to be accompanied by size markers, and to retain visible background signal within the linear range (i.e. should not be saturated). The boundaries of panels with low background have to be demarked with black lines. Splicing of panels should only be considered if unavoidable, and must be clearly marked on the figure, and noted in the legend with a statement on whether the samples were obtained and processed simultaneously. Quantitative comparisons between samples on different gels/blots are discouraged; if this is unavoidable, it should only be performed for samples derived from the same experiment with gels/blots were processed in parallel, which needs to be stated in the legend.

Unprocessed scans of all key data generated through electrophoretic separation techniques need to be presented in a supplementary figure that should be labelled and numbered as the final supplementary figure, and should be mentioned in

every relevant figure legend. This figure does not count towards the total number of figures and is the only figure that can be displayed over multiple pages, but should be provided as a single file, in PDF or TIFF format. Data in this figure can be displayed in a relatively informal style, but size markers and the figures panels corresponding to the presented data must be indicated.

The total number of Supplementary Figures (not including the “unprocessed scans” Supplementary Figure) should not exceed the number of main display items (figures and/or tables (see our Guide to Authors and March 2012 editorial <http://www.nature.com/ncb/authors/submit/index.html#suppinfo>; <http://www.nature.com/ncb/journal/v14/n3/index.html#ed>). No restrictions apply to Supplementary Tables or Videos, but we advise authors to be selective in including supplemental data.

GUIDELINES FOR EXPERIMENTAL AND STATISTICAL REPORTING

REPORTING REQUIREMENTS – We are trying to improve the quality of methods and statistics reporting in our papers. To that end, we are now asking authors to complete a reporting summary that collects information on experimental design and reagents. The Reporting Summary can be found here <https://www.nature.com/documents/nr-reporting-summary.pdf>. If you would like to reference the guidance text as you complete the template, please access these flattened versions at <http://www.nature.com/authors/policies/availability.html>.

We strongly recommend the presentation of source data for graphical and statistical analyses as a separate Supplementary Table, and request that source data for all independent repeats are provided when representative experiments of multiple independent repeats, or averages of two independent experiments are presented. This supplementary table should be in Excel format, with data for different figures provided as different sheets within a single Excel file. It should be labelled and numbered as one of the supplementary tables, titled “Statistics Source Data”, and mentioned in all relevant figure legends.

Version 1:

Decision Letter:

Dear Professor Greening,

Your manuscript "Multi-omics discovery of hallmark protein and lipid features of circulating small extracellular vesicles in humans", has now been seen by 3 referees, who are experts in EVs (Referee #1); EVs, mass spectrometry (Referee #2); and lipidomics (Referee #3), and whose comments are pasted below. In light of their advice, we regret that we cannot offer to publish the study in Nature Cell Biology.

As you will see, although the reviewers find this work interesting, they raise concerns that question the conceptual advance that these findings represent over previous work, and the strength of the data and of the novel conclusions that can be drawn at this stage. Technical concerns about the analysis and interpretation, particularly of the lipidomics data, prevent us from moving forward with your manuscript.

We are very sorry that we could not be more positive on this occasion, but we thank you for the opportunity to consider this work.

With kind regards,

Angela Parrish

Angela R Parrish, PhD
Locum Senior Editor
Nature Cell Biology

Reviewers' comments:

Reviewer #1 (Remarks to the Author):

The revised manuscript by Rai et al along with authors' responses to critique clarify and extend the number of findings being reported. With these thorough revisions and thoughtful comments, the manuscript positions itself as a foundational data repository of solid proteomic and lipidomic profiling of extracellular vesicles (EVs) in human plasma. The technical quality of the work is quite impressive and in accordance with leading standards in the field. It is very likely that the data contained in this manuscript will be used as a major resource by the EV research community and may help benchmark scores of future studies into liquid biopsy of multiple disease states, in ways exemplified by the new analysis of CAC provided by the authors. This is very commendable and of great value.

This said, and with the utmost respect to the authors, and their efforts, the current paper does not drive major new concepts, disruptive technologies, or solutions to fundamental biological questions. Maybe this is not important given other assets, but it should be said. Some aspects of the narrative are also not entirely convincing, and the authors may reconsider revising them, or removing the related data (there is no shortage of those in the paper). The case in point is the depiction of protein interactome (Fig. 2F), which for bulk data combining a highly heterogeneous population of EVs from myriads of contributing cells is completely uninformative, even confusing. The authors may look upon this as "coordinated molecular patterns", which is rather implicit given known pathways of EV biogenesis, but linkages between proteins residing in physically separate biological compartments (EVs or cells) do not signify any real molecular "interaction". This is very simple, and complex commentary does not make this any more convincing.

Reviewer #2 (Remarks to the Author):

In the response to Reviewer #2 (comment 6), the authors state that the machine learning models extract multimarker patterns to discriminate pEVs. However, it appears that all the identified 182 protein features were used as input. How would this strategy translate to routine practice? Would one need to quantify the entire protein panel and then apply the model to classify samples as pEV or not? In addition, the authors note that both Naïve Bayes and neural-network algorithms identify the multi-feature signatures. Does the signature here refer to the entire set of proteins provided to the model, or do these algorithms pinpoint a smaller biomarker subset with the strongest discriminatory power? It's not clear what the value of adding machine learning models would be here.

Reviewer #3 (Remarks to the Author):

Re: NCB-A53707A

Since the lipidomic analysis is based on LC/triple quadrupole mass spectrometer, I am really troubled by the authors' assertion that, e.g. the single transition 772.6 / 184.1 to cover 5 species including two isomers of PC 35:2, two isomers of PC O-36:2 and 1 isomer of PC P-36:1 in plasma / fractions. m/z 184 is a common product ion for PC and sphingomyelin. I really do not believe that the described 5 PC species can be separated by HPLC, in particular, ion mobility was not incorporated for the analysis. As far as I know, all the five species are very rare, and if they exist at all. They would be very minor (very low intensity), and I am sure no way that the feature ions specific for plasmalogen (P-36:1) or 1-O-alkyl PC (PC O-36:2) lipids can be generated by CID on their [M + H]⁺ ions for structural identification or confirmation. PC 35:2 is an odd-chain FA containing phospholipid, which is known to be very rare in mammalian system. Species like PC(37:5)*, PC(37:6)*, PC(39:6) (a)*, PC(39:6) (c)* are also odd chain FA containing PC and are extremely rare. On the other hand, if high resolution mass spectrometry were employed, at the very least, diacyl-PC and alkyl/alkenyl can be differentiated (but not by a triple quadrupole instrument). Thus, unless a solid proof of their existence, they should not be reported. Structural assignments pulled out from computer generated database, simply can't be the final decision for structural assignment. Again, reporting a big number of lipid species without seal of proof is meaningless, and sometimes can be just laughing matters. My point, again, is the fidelity of the reported lipid repertoire.

Other comments:

The color label for Supplementary Figure 19 is very confusing.

Same problems with downloading Table 1-41 (as the first version of the manuscript). Nothing can be found in Supplemental material table 1-41, excepting 3 tables that are not useful, All the other Supplementary Tables, such as 40, Supplementary Table 27, 31 and 32, which are important, are not extractable for review.

Table 2. List of EV and NonEV lipid features, what are those lipids designated with "*" superscript?.

**For Nature Portfolio general information and news for authors, see <http://npg.nature.com/authors>.

Version 2:

Decision Letter:

Our ref: NCB-A53707B-Z

13th August 2025

Dear Dr. Greening,

Thank you for submitting your revised manuscript "Multi-omics discovery of hallmark protein and lipid features of circulating small extracellular vesicles in humans" (NCB-A53707B-Z). It has now been seen by the original referees and their comments are below. The reviewers find that the paper has improved in revision, and therefore we'll be happy in principle to publish it in Nature Cell Biology, pending minor revisions to satisfy the referees' final requests and to comply with our editorial and formatting guidelines.

Thank you again for your interest in Nature Cell Biology Please do not hesitate to contact me if you have any questions.

Sincerely,

Angela R Parrish, PhD
Locum Senior Editor
Nature Cell Biology

Reviewer #1 (Remarks to the Author):

The revised paper by Rai et al addresses, at least to some extent, the concerns that were raised before. I have great respect for the monumental work performed by the authors and the value of their data, as a resource for the community. However, I remain skeptical about the discovery aspect of the study, which reports a refined profile of what is otherwise to some extent known, namely EVs in plasma. On technical merit the paper is very thorough, as far as I can judge. The notion that one can use ADAM10 and PS(36:1) as conserved markers of plasma EVs in healthy individuals is interesting but does not mean that in patients with specific disease states this will remain the case given the possible change in cellular sources of circulating EVs e.g. platelets, inflammatory or cancer cells. The authors chose to remain agnostic as to the contributions of various cells to the EV profile they studied, which makes their analysis, for all its merits, very technical in nature and almost 'questionless'. Whether this is a factor in making publication decisions in this particular journal is a matter of opinion, but ultimately totally within the editorial prerogative.

Reviewer #2 (Remarks to the Author):

My comments have been addressed.

Reviewer #3 (Remarks to the Author):

Manuscript #: NCB-A53707B-Z

Comments:

I am convinced, in fact, very impressed by the nice data (reviewer figures 1-6, and supplementary Figure 20) toward the characterization of the unusual PC species. However, there is inconsistency in Supplementary Figure 19. "Characterization of PS 36:1 in EVs using negative ionization mode. Pooled EV's were run under the same conditions in negative ionization mode, looking for fragments corresponding to the neutral loss of the serine headgroup (red) and all possible fatty acid species, resulting in identification of 18:0 (283.3 m/z) and 18:1 (281.3 m/z)". The very top trace is [M + H]⁺ ion in the positive ion mode. Therefore, the legend should be revised accordingly.

This is a very high quality paper, and I strongly recommend the publication of the paper.

We thank all the reviewers for critical review/suggestions in improving the manuscript.

Blue = response to reviewers queries

Green = specific text changes made to the main manuscript file

Reviewer #1:

The manuscript by Rai et al describes authors' formidable effort to comprehensively and systematically catalogue and characterize the proteome and lipidome of the particulate content of human plasma, with special distinction of extracellular vesicles (pEVs) and nonEV particles (pNVPs). The authors collected a large cohort of 110 human plasma samples from several centres and adopted the high-resolution iodixanol based density gradient separation (DGS) to fractionate the respective particles. The fractions enriched in pEVs and pNVP were then processed for mass spectrometry which yielded approximately 4500 pEV proteins and 829 lipids. Of those, some were highly conserved in pEVs (182 and 52 correspondingly). The authors also performed surfaceome analysis resulting in validation of external membrane proteins. They also performed TMT proteomics for co-expression studies. Pathway analysis revealed that pEV cargo was reassuringly enriched in components of the endosomal system and lipid rafts, while pNVPs contained elements of the complement and coagulation cascades. Notably, machine learning analysis of the profiling output across the samples pointed to ADAM10 as a highly enriched protein (marker) of pEVs, possibly usable for EV purity assessments, along with phosphatidyl serine [PS(36:1)]. The rich database generated by this study served to develop an open-source R/Shiny web tool for the benefit of the EV research community.

Overall, this is a data-rich, systematic and solid benchmarking study of considerable value for those with interest in EV biology, liquid biopsy, analysis of plasma components along with the related fields. The study provides a valuable resource in terms of data, functional pipeline and a R/Shiny web tool. It also addresses long standing needs in terms of baseline plasma EV and NVP characterization, something that may inspire similar efforts in the context of other biofluids where these particles are being sampled and studied. The analysis appears to be thoroughly executed and showcases the expertise of the authors, who are leaders in EV proteomics. The manuscript is also well written, well documented and informative.

Although this is, in many ways, a potentially important, even foundational contribution to the EV research field, there several considerable weaknesses in light of this high profile submission.

1. The paper represents an important scaled up version of prior studies on plasma particles (this is valuable), but it contains relatively little conceptual or experimental novelty. The power of (DGS) in separating EVs from NVP has already been demonstrated (Jeppesen et al 2019), proteomes of plasma and cellular EVs have been extensively studied (Hoshino et al 2020), and protein hubs for EV assembly have been explored to some extent (Luck et al 2020; Kugeratski et al 2021). Properties, numbers and some of the markers of plasma EVs have also been described, as cited by the authors (e.g. Arraud et al 2014). While the stringency with which ADAM10 and PS associate with plasma EVs in the present study is impressive, both of these molecular species have already been found in EVs (Vesiclepedia) with PS having long been used for preparative purposes, even commercially.

We appreciate the reviewer's thoughtful assessment of our study and their recognition of its value in systematically benchmarking plasma extracellular vesicles (pEVs) and non-vesicular particles (nonEVs). We welcome the opportunity to clarify how our study builds upon and extends previous research in the field while addressing key limitations that remained unresolved. Previous studies have been foundational in shaping the current understanding of EVs. Jeppesen et al., 2019 demonstrated the power of density gradient separation (DGS) in isolating EVs from non-vesicular particles, and Hoshino et al., 2020 provided valuable insights into the proteomic content of plasma and cellular EVs. Studies such as Luck et al., 2020 and Kugeratski et al., 2021 explored protein interactions and EV biogenesis, providing essential frameworks for understanding

vesicular assembly mechanisms. While these studies have significantly advanced the field, certain key gaps remained, which our study addresses through expanded molecular profiling, increased methodological rigor, and large-scale validation.

Jeppesen et al., 2019 pioneered high-resolution DGS, identifying CD63, CD81, and CD9 as plasma EV markers while showing that Ago2 and APOA1 were predominantly non-vesicular. These findings complement our isolation strategy, as we also observed these markers in our pEV and nonEV fractions, further supporting the distinction between EVs and non-vesicular particles. However, this previous study was largely limited to cell line-derived EVs (DKO-1, Gli36) and did not explore the proteome or lipidome of human plasma EVs. Our study builds upon these findings by applying a systematic DGS workflow to a large multi-centre cohort ($n > 150$), providing a comprehensive molecular characterization of plasma EVs and nonEVs that extends beyond individual cell lines.

Hoshino et al., 2020 made an important contribution by profiling the proteomes of plasma EVs and demonstrating their relevance in cancer diagnostics. However, their study relied on direct ultracentrifugation for plasma EV isolation, which lacks high-resolution separation and can co-isolate non-vesicular components, as also demonstrated by Jeppesen et al., 2019. While ultracentrifugation is a widely used approach, it does not provide the same degree of separation between vesicular and non-vesicular particles as DGS. Our study complements and extends these findings by employing DGS, which provides greater specificity in distinguishing EVs from non-vesicular particles. Additionally, our study is among the first to apply large-scale lipidomic analyses to plasma EVs, systematically characterizing their molecular compositions. This lipidomic component allows us to pin point novel lipid markers for plasma EVs, such as PS(36:1), which had previously been recognized for its role in EV isolation but had not been established as a robust lipid marker for plasma EV purity assessment.

While large-scale datasets such as ExoCarta have been instrumental in cataloguing EV-associated proteins, they rely primarily on aggregated proteomics data, which may include non-vesicular contaminants. Luck et al., 2020 leveraged such datasets to map EV protein interactions, but Jeppesen et al., 2019 demonstrated that many of the 25 most frequently reported proteins in ExoCarta were more associated with nonEV fractions than purified EVs. Similarly, prior studies such as Luck et al., 2020 and Kugeratski et al., 2021 have explored protein interactions and EV biogenesis, primarily in cell culture systems, but did not directly compare EVs to non-vesicular particles in plasma. Our study provides an independent assessment of EV markers using rigorous validation approaches, demonstrating, for example, the specific enrichment of ADAM10 and PS(36:1) in plasma EVs *versus* nonEVs. This distinction is particularly important, as database-derived EV markers often require further validation to confirm their vesicular origin. Importantly, our surfaceome analysis further refines these datasets by validating plasma EV proteins in the context of membrane localization and accessibility, which is critical for biomarker development and functional studies.

In addition to these methodological advances, our study introduces a machine learning-based framework to verify multiple EV markers that can distinguish between EV and nonEV particles, ensuring greater resilience against biological variability intrinsic to human studies.

Thus, our study builds upon and complements prior work by integrating high-resolution fractionation, large-scale molecular profiling (proteomics, surfaceome, lipidomics), and machine learning-driven marker selection, validating them in human EVs (healthy vs disease, multiple sources), multiple cell lines EVs (cancer cell lines and non-transformed cell lines), different EV types (small and large EVs, CDs). While previous studies have provided essential insights into EV biology, our work addresses key limitations, expands molecular catalogues, and applies rigorous validation across independent cohorts, ultimately advancing plasma EV characterization and standardization efforts.

Essentially, we answer fundamental cell biology question of what makes EVs circulating in human, generating a high-confident protein and lipid map of these signalling organelles.

2. As such, the manuscript offers a relatively deep, but rather descriptive overview of plasma EV (and NVP) proteome and lipidome and is not asking any specific, or novel biological questions. There is room for this kind of benchmarking, of course, but defining ‘core’ proteome and lipidome says very little about biological properties, activities, functionalities, uptake and other aspects of EVs and NVPs that authors declare as their main objective.

We agree with reviewer’s perspective that functional assays are critical for advancing the EV field. However, our study’s primary objective is to define the core proteome and lipidome of plasma EVs, establishing a biologically meaningful reference framework rather than solely presenting a descriptive catalogue. This molecular foundation is essential for functional studies, as it provides a rigorous basis for quality and purity assessment of circulating EVs—a prerequisite for confidently studying EV activities such as uptake, signalling, and biophysical interactions. Without such a benchmark, functional assays risk being confounded by plasma background proteins or non-EV contaminants, which could obscure or misrepresent EV functions.

To enhance biological interpretation, we integrate proteomic and lipidomic data to identify co-regulated networks of EV-associated molecules, linking specific lipid species with key EV proteins. Our pathway and enrichment analyses reveal that core plasma EV components are enriched in endosomal system components, membrane trafficking proteins, and lipid rafts, whereas non-EVs are predominantly associated with complement and coagulation cascades, providing mechanistic insights into EV biogenesis and functional behaviour in circulation. These distinctions reinforce the need for standardized EV purification and characterization and can further inform international EV guidelines, ensuring meaningful functional insights in plasma EV research. To further connect molecular composition with EV functionality, we employ surfaceome profiling, which identifies membrane-associated plasma EV proteins with direct implications for uptake, targeting, and recipient cell interactions. We now also report on quantification of low-abundant and difficult to quantify (Hoshino et al 2020) but highly bioactive and functionally relevant molecules such as chemokines, cytokines and signal transduction molecules (**Reviewer 1, point 6**) within these EVs. These findings provide a rational basis for selecting functional EV markers that can guide studies on EV-mediated signalling pathways and cellular crosstalk.

Importantly, we demonstrate that our identified plasma EV features are conserved across different EV sources, even in human studies where plasma composition is inherently more variable than in controlled cell culture models. By employing a multi-feature, machine learning-based classification framework—rather than relying on single-marker approaches—we provide a robust strategy for distinguishing EVs from non-EV contaminants. Notably, while plasma EVs from individuals with different pathological conditions exhibit distinct proteomes, the core EV features remain conserved (**Reviewer 1, point 3**), underscoring their utility for standardizing EV research while accounting for biological variations. To further strengthen our findings, we validate these molecular features in an independent plasma cohort (AusDiab study, n=22 plasma samples, **Reviewer 2, point 4**) and extend our benchmarking to EVs derived from non-transformed fibroblasts and endothelial cells, encompassing both small and large EVs (**Reviewer 2, point 2**). This validation supports the broader biological relevance of these molecular features, demonstrating their role in shaping EV functionality beyond disease-specific contexts.

While we acknowledge that our study does not include functional assays such as EV uptake or signalling studies, we emphasize that a rigorous molecular definition of plasma EVs—the primary aim of our study—is a necessary foundation for such investigations. Without well-defined molecular benchmarks, functional studies risk being compromised by undefined heterogeneity and contaminants. By establishing a high-confidence molecular reference dataset, our study lays the groundwork for future research to systematically investigate plasma EV biological activities, interactions, and functional properties with molecular precision.

3. It is also presently unclear whether and how the parameters defined by the authors would change and in the context of disease such as inflammation, cardiovascular disease, sepsis, trauma, cancer, thrombosis or other states where plasma EV analysis is already underway. This uncertainty weakens the appeal of the present study by comparison to previously published efforts (e.g. Hoshino et al 2020). How would EV profiles, **cargo**

and ‘**markers**’ change under pathological conditions, age, sex and other variables? By compiling samples from different sources all these important variables that would be critical, for example, to understand individualised physiological conditions or patient care become obscured. How would the presently defined bulk ???baseline??? **help guide future biomarker studies?**

We appreciate the reviewer’s important point regarding how plasma EV profiles (i.e. cargo and “markers”) may change in disease contexts and how our study serves as a foundation for future biomarker studies.

To address this, and to further demonstrate the clinical relevance of our framework, we have now directly compared pEVs from individuals with and without coronary artery calcification (CAC), a well-established marker of future cardiovascular events, primarily ischemia (**Extended Data 9, Supplementary Figure 8, Supplementary Table 21-23**). CAC scoring is ascertained by computed tomography (CT) and can independently predict cardiovascular outcomes⁴⁻⁶, including myocardial infarction and all-cause mortality^{7, 8}, and hence used for risk re-stratification in cardio vascular diseases. Our proteomic analysis reveals that the pEV proteome can distinguish between individuals with CAC (CAC-positive score) and those without (CAC score = 0), highlighting that pEV protein cargo are distinct. We identified 76 upregulated and 127 downregulated proteins (FDR < 0.05), with the upregulated proteins significantly enriched in the GO pathway ‘Abnormal cardiovascular system physiology’. Notably, this includes proteins such as CST3, TPM1, TXNRD2, and MAP2K1, all of which have well-established roles in cardiovascular disease (CVD) and vascular calcification. CST3 (Cystatin C) is an important marker of arterial stiffness and atherosclerosis, playing a key role in vascular remodelling and calcification. TPM1 (Tropomyosin 1) regulates vascular smooth muscle cell function, which is directly implicated in vascular stiffening and plaque formation. TXNRD2 (Thioredoxin Reductase 2) plays a protective role against oxidative stress, a key driver of coronary calcification and endothelial dysfunction. MAP2K1 (MEK1) is a central regulator of the MAPK/ERK signalling pathway, influencing vascular inflammation and extracellular matrix remodelling, both of which contribute to CAC progression. The enrichment of these proteins in CAC-associated pEVs underscores their potential role in disease-specific EV signalling and highlights the ability of pEVs to reflect disease processes in circulation. Furthermore, our study demonstrates that pEVs can resolve low-abundance proteins in plasma (**Reviewer 1, point 6**), which is particularly relevant given that disease-associated proteins are often 1000-fold less abundant than highly abundant plasma proteins present in nonEVs. Notably, these low-abundance proteins include tissue-specific signatures (**Reviewer 1, point 5**), further supporting the potential of “high purity” pEVs as a superior platform for disease biomarker discovery.

Importantly, despite the proteomic differences observed between CAC and non-CAC individuals, the core set of 182 pEV marker proteins remained highly conserved across both groups, indicating that the fundamental pEV molecular signature is stable across individuals, even in the context of disease (**Extended Data 9F-G**). This supports the robustness of our pEV characterization and its applicability across different physiological conditions. In contrast, nonEV particles and neat plasma proteomes were unable to distinguish CAC from non-CAC individuals, underscoring the power of highly purified pEVs (separated from nonEV particles) in revealing disease-associated changes in the plasma proteome.

While our study does not explicitly examine variables (a future but a huge undertaking beyond the scope of this study) such as age, sex, inflammation, cancer, thrombosis, or other pathological states, the preservation of core molecular identity in pEVs while revealing disease-specific differences in EDCAD study reinforces their potential as a well-defined reference against which changes can be meaningfully interpreted. We anticipate that molecular markers we define also provide a scalable framework that will facilitate longitudinal and disease-specific EV research, allowing future studies to assess how EV signatures deviate in disease settings.

We have made the following inclusion in the Results (page 7, line 257):

Importantly, we show that pEV protein cargo differs between individuals with and without coronary artery calcium (CAC) deposits in our EDCAD cohort (**Extended Data 9A-C, Supplementary Table 21-22**). Occurrence and development of calcification is a complex biological process, which is regulated by multiple factors, including EVs that are regarded as nidus for calcification by providing mineral nucleation sites⁷¹. CAC

scoring, a predictor of future cardiovascular events, is determined by computed tomography. We identified 76 upregulated and 127 downregulated proteins (FDR < 0.05) in pEVs from individuals with positive CAC scores (**Extended Data 9D-E, Supplementary Table 21-22**), with the upregulated proteins significantly enriched in the GO pathway ‘Abnormal cardiovascular system physiology’ (**Extended Data 9D, Supplementary Table 23**). Notably, this includes (Cystatin C (CST3)⁷², TPM2⁷³, and TPM1⁷⁴, and TXNRD2⁷⁵, all of which have well-established roles in cardiovascular disease and vascular calcification. Despite these proteomic differences between CAC and non-CAC pEV proteomes, the core set of 182 pEV marker proteins remained highly conserved across both groups (**Extended Data 9F-G, Supplementary Table 24**), indicating that the fundamental pEV molecular signature is stable across individuals, potentially even in the context of disease (**Figure 3D**). This preservation of core molecular identity in pEVs across CAC and non-CAC individuals, while simultaneously capturing disease-specific molecular shifts in the EDCAD study, reinforces their potential as a well-defined reference for plasma EV research including disease biomarker discovery. In contrast, NonEV particles and neat plasma proteomes were unable to distinguish CAC from non-CAC individuals, underscoring the power of highly purified pEVs (separated from NonEV particles) in revealing disease-associated changes in the plasma proteome (**Supplementary Figure 8**).

We have made the following inclusion in the Discussion (page 13, line 544):

Importantly, despite disease-associated proteomic changes, the core set of 182 plasma EV protein features remained highly conserved across both CAC and non-CAC individuals in the EDCAD cohort, underscoring their robustness as molecular hallmarks of circulating EVs. This warrants future investigation on molecular stability across diverse disease states to establish the use of these core pEV features as a reliable foundation for human plasma EV studies in large population-based cohorts, enabling consistent characterization and cross-study comparability.

Table inclusions:

Supplementary Table 21. Proteome analysis of pEVs, NonEVs, and neat plasma from individuals (EDCAD cohort) with either positive CAC score (CAC group) or zero CAC score (Healthy group).

Supplementary Table 22. Differential abundant protein analysis of pEVs from individuals (EDCAD cohort) with either positive CAC score (CAC group) or zero CAC score (Healthy group).

Supplementary Table 23. Gene Ontology pathways enriched in differential abundant proteins in pEVs from individuals (EDCAD cohort) with either positive CAC score (CAC group) or zero CAC score (Healthy group).

Extended data 9. Proteome analysis of pEVs from individuals with either positive CAC score (CAC group) or zero CAC score (Healthy group). **A.** Proteins quantified in CAC_pEVs or Healthy_pEVs. Lower panel indicates normalized proteins abundance (vs2-normalized) **B.** Pearson correlation of quantified proteins. **C.** Principal component analysis of quantified proteins. **D.** Heatmap depicting K-means clustering of differentially abundant proteins (FDR < 0.05) in CAC_pEVs versus Healthy_pEVs: 76 upregulated and 127 downregulated proteins (FDR < 0.05), with GO pathways enriched in upregulated proteins. Proteins of the GO pathway ‘Abnormal cardiovascular system physiology’ are shown. **E.** Boxplot showing vs2-normalized abundance (log2) of indicated proteins in CAC_pEVs and Healthy_pEVs proteomes. **F.** Boxplot showing distribution of either significantly upregulated (CAC_UP) or down regulated (CAC_DOWN) proteins, or non-significantly dysregulated proteins (NS proteins), and EV proteins features (EV feature) in CAC_pEVs and Healthy_pEVs proteomes. **G.** pEV protein features quantified in individual CAC_pEVs versus Healthy_pEVs proteomes. Lower panel shows distribution of pEV protein features in individual proteomes.

Supplementary Figure 8. Proteome analysis of pEVs, NonEVs and neat plasma from EDCAD set. Individuals with either positive CAC score (CAC group) or zero CAC score (Healthy group) are indicated. **A.** Proteins quantified in pEVs, NonEVs and neat plasma from individuals with either positive CAC score (CAC group) or zero CAC score (Healthy group). **B.** Box plot depicting normalised intensities (Z-scored) of indicated proteins. **C.** Heatmap of Pearson correlation of quantified proteins. **D.** Principal component analysis of quantified proteins.

4. By the authors' own admission, the present analysis of 'bulk' EV or NVP preparations suppresses the enormous heterogeneity of these particles, not only in terms of the officially recognized 'subtypes' such as exosomes, ectosomes, or their variants (e.g. ARMMs) but also the natural variability within each of these categories, including at the single vesicle level. Important functionalities of small EV subsets could be lost in this process. In fact, it would be of interest to assess whether ADAM10 is expressed in all EVs, or in some EVs but in all plasma samples, and what would either mean.

We concur that bulk pEV analysis may obscure important functionalities of individual EV subpopulations, which remains a limitation of our study. As the reviewer points out, the heterogeneity of EVs extends not only to recognized subtypes such as exosomes and ectosomes but also to variability at the single-vesicle level. While ADAM10 is conserved in different circulating exosome subpopulations in humans (based on surface expression of tetraspanins, CD9, CD63, CD81⁶¹, Figure 3), the reviewer raises an important question whether ADAM10 is present in all EVs or only in some, and whether it is universally detected across all plasma samples.

To address this, we performed Aurora Cytek-based single-particle analysis of pEVs using an ADAM10 antibody or IgG control, with instrument gating calibrated using 90–200 nm beads (Figure 6H, Supplementary Figure 16). To assess phosphatidylserine (PS) exposure as a vesicle marker, we also labelled EVs with Annexin V stain, which binds PS lipids. Our results revealed that ADAM10 signals were detected in ~40% of pEVs, with a range of 20–75% across samples, a distribution comparable to the PS lipid signal using Annexin V. To verify that these signals were EV-derived, we subjected EVs to solubilization with SDS, which eliminated the ADAM10 signal, confirming its association with vesicular structures. Thus, our data show that ADAM10 is not universally expressed across all EVs, supporting the notion that EV heterogeneity extends to protein content at the single-vesicle level.

However, when analysed at the population level, mass spectrometry profiling across over 140 pEV proteomes from different individuals consistently detected ADAM10 in all samples, suggesting that while ADAM10 is present in plasma EVs universally, it is heterogeneously distributed across individual vesicles. This observation aligns with prior studies indicating that ADAM10 does not regulate the production of all small EV subpopulations (CD81/CD63-positive EVs, PMID: 29721369). Furthermore, previous research on melanoma-derived EVs has shown that mature ADAM10 is enriched in specific vesicle fractions, whereas high-molecular-weight ADAM10 C-terminal fragments are selectively present in other EV fractions (PMID: 30690120). This suggests that ADAM10-containing EVs may represent distinct vesicle subsets formed through specific sorting mechanisms, possibly via the ESCRT pathway.

While the precise biological function of ADAM10 in EVs remains incompletely understood, prior evidence suggests that it plays an active role in extracellular proteolysis and intercellular communication rather than simply serving as a structural EV marker. Notably, a recent study (PMID: 33484965) identified ADAM10-dependent cleavage of PTGFRN (Prostaglandin F2 receptor negative regulator) during EV biogenesis, implicating ADAM10-containing EVs in proteolytic processing events. Other findings indicate that mature ADAM10 is released in EVs by lymph-node mesenchymal stromal cells, where it retains enzymatic activity within the extracellular microenvironment. These observations collectively suggest that ADAM10-containing EVs are biophysically distinct and functionally specialized, reinforcing the notion that EV heterogeneity is not random but biologically regulated.

Despite the inherent limitations of bulk EV analysis, it provides critical insights into systemic EV function that may not be captured by single-vesicle studies alone. Bulk proteomic and lipidomic profiling enables detection of conserved molecular signatures that would otherwise be difficult to identify in rare or highly heterogeneous subpopulations. Our findings also demonstrate that despite vesicle-level heterogeneity, ADAM10 is consistently detected across all individuals, reinforcing its biological relevance in plasma EVs. Furthermore, the fact that only a subset of EVs contain ADAM10 suggests a regulated sorting mechanism rather than stochastic incorporation, highlighting the importance of further investigating EV subtype-specific functionalities.

While single-vesicle flow cytometry and high-resolution imaging techniques would be ideal for further dissecting EV subpopulation heterogeneity, such approaches were beyond the scope of this study. However, we agree that future research should explore the differential packaging of ADAM10 into distinct EV subsets, which remains an area of ongoing interest in our laboratory.

Thus, our study provides strong evidence that ADAM10 is a consistent component of plasma EVs but is not universally present across all vesicles, reinforcing that EV heterogeneity is a fundamental property of these particles. Prior studies support that ADAM10-containing EVs are functionally distinct and may participate in extracellular proteolysis and intercellular signalling, rather than simply serving as a structural component of all EVs. While bulk analysis offers essential insights into conserved molecular features, we recognize that single-vesicle studies will be needed to further refine our understanding of ADAM10's role in EV biology, and we anticipate that future research will address this intriguing question in greater detail.

We have made the following inclusion in the Results (page 10, line number 415):

Furthermore, single-vesicle analysis using Cytek Aurora flow cytometry confirmed the presence of ADAM10 on a subset of plasma EVs (Figure 6H, Supplementary Figure 16), with ~40% of pEVs exhibiting ADAM10⁺ signals, comparable to PS signal detected using Annexin V staining (Figure 6H, Supplementary Figure 16C, Supplementary Table 39). SDS detergent treatment showed strong reduction in fluorescence signal intensity and count, suggesting their EV-origin⁸⁶ (Supplementary Figure 16B-C). These findings suggest that while ADAM10 is a component of plasma EVs across individuals, it is not universally present across all vesicles, consistent with previous report⁸⁷.

We have made the following inclusion in the Discussion (page 11, line number 462):

The selective presence of ADAM10 (and PS lipids) in only a subset of EVs highlights the non-random nature of EV cargo packaging, supporting models of regulated biogenesis pathways such as ESCRT-dependent sorting⁸⁹. Evidence that ADAM10 retains proteolytic activity in EVs⁹⁰ suggests that these vesicles may serve as active mediators of extracellular remodeling and intercellular communication, expanding their functional relevance beyond passive biomarker carriers. The differential association of ADAM10 with specific EV subtypes, and its absence from some canonical small EV populations⁸⁷, underscores the existence of biophysically and functionally distinct EV in circulation. Together, these findings support a growing consensus that EV heterogeneity is biologically regulated rather than stochastic, with specific protein profiles reflecting distinct functional roles in physiological and pathological contexts^{87,89,90}.

Figure 6. ADAM10 protein and phosphatidylserine lipid are biological markers for circulating EVs in humans. H. Scatter plot showing detection of ADAM10-positive and PS positive pEVs, which was sensitive to SDS detergent (0.5%) solubilization.

Supplementary Figure 16. Spectral flow cytometry (Cytek Aurora)-based pEV analysis for ADAM10 and PS expression A. Scatter plot depicts instrument gating calibration using 90 nm, 125 nm, 150 nm, 200

nm and equal mix (90-200 nm) beads. Bottom panel indicated buffer alone control. YG3-A channel used for detecting Alex Flour 568 signal. R1-A channel used for detecting APC signal. **B.** Cytek Aurora-based pEV analysis. Scatter plot showing detection of ADAM10 antibody-AF568 positive (detected using YG3-A channel) and PS positive (Annexin V-APC, detected using R1-A channel) in pEVs. IgG used as control for ADAM10 antibody. SDS detergent (0.5%) solubilization was performed to indicate EV origin of these signals. **C.** Bar plot depicting ADAM10+ (AF548 signal), PS+ (APC signal) or overlapping (ADAM10 and PS +) signal.

Methods inclusion:

Nanoflow analysis of plasma EVs

Plasma EVs (~5 µg) were subjected to fixation and permeabilization using eBioscience™ Foxp3 / Transcription Factor Staining Buffer Set (Invitrogen™, 00-5523-00). Briefly, pEVs (in 50 µl PBS) was incubated with 500 µl fixation and permeabilization buffer on ice for 30 mins. Samples were ultracentrifugation at 100,000 g (1 h at 4 °C) and pellets resuspended in 100 µl wash buffer. Samples were stained with 5 µl APC Annexin V reagent (BioLegend) and either 2 µg of Anti-ADAM 10 Antibody (Sigma-Aldrich, AB19026) or Rabbit IgG Isotype Control (Invitrogen, 10500C). Samples were incubated at room temperature (gentle end-over mixing) for 1h. Samples were topped with 900 µl wash buffer and washed twice (ultracentrifuged at 100,000 g (1 h at 4 °C)) to remove any remaining antibodies and potential antibody aggregates. The pellets resuspended in 100 µl wash buffer containing 0.5 µl of Goat Anti-Rabbit Alexa Fluor 568 Dye secondary antibody (ThermoFisher Scientific) incubated for 30 mins at room temperature (gentle end-over mixing) in dark. Samples were topped with 900 µl wash buffer and washed twice (ultracentrifuged at 100,000 g (1 h at 4 °C)). Pellets resuspended in 100 µl of PBS (0.5% bovine serum albumin) filtered using 0.22 µm filter. Samples were immediately analysed using Cytek Aurora flow cytometer and SpectroFlo® software, as previously described⁸⁶. Briefly, controls included isotype control antibodies were used to test for unspecific binding, in addition to unstained EV suspension, and buffer-only controls. For purity assessment, Sodium dodecyl sulfate detergent (SDS) was added to labelled EVs at a final concentration of 0.5%, followed by vigorous vortexing for ~30 s before acquisition. High purity would be indicated by a strong reduction in vesicle concentration and fluorescence after the treatment. The instrument set-up was consistent across all experiments and followed recommendations from the manufacturer. Prior to the acquisition, the flow cell was washed with Contrad to minimize machine-associated noise. Instrument gating calibration was performed using 90 nm (#64009-15) 125 nm (#64011-15), 150 nm (#64012-15), 200 nm (#64013-15) and equal mix (90-200 nm) beads (Nanobead NIST Traceable Particle Size Standards). The threshold for side scatter was set to 430, and the gain of side scatter (SSC) were set to 10. YG3-A channel used for detecting Alex Flour 568 signal, R1-A channel used for detecting APC signal and 10 000 events were recorded for all samples with the slowest flow rate to minimize the swarming effect. Statistical analysis of flow cytometry values was performed using GraphPad (v 9.1.0) using multiple paired t-tests.

Table inclusions:

Supplementary Table 39. Cytek aurora-based pEV analysis for ADAM10 and PS expression (Supplementary Figure 16)

5. Moreover, plasma EVs come from multiple cellular sources, such as platelets, blood cells, endothelial cells and other compartments. In the present study they are all lumped together in spite of the fundamentally different biological properties of EVs originating from different cells. This is surprising, as high-resolution proteome should allow, at the very least, to distinguish between the signatures of EVs from specific cellular sources, which would be of considerable value and interest. It could be argued that seeking predictable commonalities obscures the molecular determinants of specific EV activities.

As the reviewer aptly points out, plasma-derived EVs originate from multiple cell types, including platelets, blood cells, endothelial cells, and others, forming a dynamic and heterogeneous pool. However, precise source attribution remains technically and biologically challenging, particularly in the absence of prior isolation or enrichment strategies. While it is theoretically possible to infer EV cellular origins using protein markers, confidently distinguishing EVs from different cell types in bulk plasma remains non-trivial. The primary limitation lies in the lack of high-confidence “**universally exclusive markers**” for each cellular source, which

constrains classification accuracy even in targeted approaches such as immunoaffinity purification or single-vesicle profiling. High-confidence attribution of EV origin within this complex and dynamic pool also requires addressing multiple key considerations. Beyond identifying universally exclusive markers, these include: (i) whether the identified marker is actively packaged into EVs, (ii) whether it is surface-expressed with correct topology (i.e., accessible binding sites for targeting molecules such as antibodies or aptamers), and (iii) whether it has been rigorously validated as cell-type-specific in the EV context. Given that these requirements demand extensive experimental validation and marker exclusivity testing, such analyses fall beyond the scope of the current study but remain an important future direction.

To address the question of source attribution in a data-driven manner, we manually curated tissue- and cell-type-specific protein signatures from three independent sources (**Extended Data 2, Supplementary Table 11-12**): (1) The Human Protein Atlas (HPA), which provides tissue- and organ-specific proteins based on immunohistochemistry (**Panel A**); (2) HPA nTPM dataset, an RNA-derived protein expression database across tissues and cells (**Panel B–C**); and (3) Organ-specific protein signatures detected in human plasma, identified via aptamer-based detection (PMID: 38057571) (**Panel D**). By integrating these datasets with our high-resolution pEV proteomes, we observed proteins derived from diverse tissues, including the adrenal glands, brain, bone marrow, kidney, liver, and heart muscle, all present at comparable abundances (**Panel A–C**). At the cellular level, proteins from glandular cells were the most abundant in pEVs, likely reflecting their high secretory capacity, followed by proteins associated with hematopoietic cells, endocrine cells, endothelial cells, lymphoid cells, adipocytes, neural cells, epithelial cells, and cardiomyocytes (**Panel B**). Similarly, organ-specific signatures previously detected in plasma using aptamer-based approaches were also quantified in our pEV proteomes, including proteins from the liver, skin, bone marrow, and brain (**Panel D**). To further validate these findings, Gene Set Enrichment Analysis revealed significant enrichment of cell-type-specific protein signatures for endothelial cells, fibroblasts, hepatocytes, cardiomyocytes, and kidney-derived cells (adjusted $p < 0.05$) (**Panel E–F**). Importantly, these tissue-enriched proteins were largely absent in non-EV fractions, reinforcing the importance of EV purity in capturing biologically relevant signals. Thus, while precise cellular origin deconvolution remains complex, our dataset provides high-resolution detection and quantification of tissue-specific EV-associated proteins, serving as molecular leads for future investigations aimed at further subclassifying plasma EVs by their sources.

The reviewer raises the concern that seeking predictable commonalities in pEVs may obscure the molecular determinants of specific EV activities. We acknowledge this perspective; however, we emphasize that identifying shared molecular signatures in plasma EVs is not merely an abstraction but a critical approach for defining conserved pathways and intercellular communication mechanisms, which are essential in both physiological and pathological processes. Rather than obscuring specificity, recognizing these commonalities allows us to define robust and reproducible molecular markers that provide biological insights across diverse EV subpopulations, and enables their application in biomarker discovery, diagnostics, and therapeutic monitoring. Furthermore, our surfaceome profiling offers additional potential to refine EV classification in future studies by identifying functionally relevant membrane-associated proteins that may serve as enrichment markers for source-specific EV characterization.

Looking forward, we anticipate that our dataset can be leveraged alongside advanced methodologies such as single-vesicle flow cytometry or affinity capture to achieve higher-resolution classification of plasma EVs based on their cellular origins. Additionally, label-free computational deconvolution methods could enable quantitative estimation of the relative contributions of different cell types to the circulating EV pool. Our current dataset lays a critical foundation for such efforts, serving as a benchmark for identifying both shared and unique EV features across different cellular origins. These findings will ultimately enhance our ability to interpret the biological and functional roles of plasma-derived EVs.

We have made the following inclusion in the Results (page 5, line number 165):

Indeed, we quantified diverse cell-, tissue-, and organ-associated proteins in pEVs^{56,57}, suggesting that our dataset provides a snapshot of the diverse vesicular population in circulation (**Extended Data 2A-D, Supplementary Table 11**): many of these cell signatures were also enriched in pEVs vs NonEVs (**Extended Data 2E-F, Supplementary Table 12**).

We have made the following inclusion in the Discussion (page 13, line number 525):

In regards to source attribution, protein signatures associated with diverse cell types were represented in pEVs, including endothelial cells, fibroblasts, hepatocytes, cardiomyocytes, kidney cells, and hematopoietic cells (such as platelets), corroborating previous reports^{15,54,55,108,109}. In healthy individuals, circulating EVs reportedly arise from hematopoietic cells (such as platelets, erythrocytes, and leukocytes) and endothelial cells^{54,55}. While definitive source attribution of plasma EVs remains technically challenging due to the lack of universally exclusive surface markers, recent computational deconvolution algorithms offer promising solutions to estimate the relative contribution of different cell types^{108,109}. Complementary proteomic studies have also identified tissue-specific EV-associated proteins, providing valuable leads for understanding EV origin and function in systemic circulation¹⁵. In parallel, high-sensitivity techniques such as single-vesicle flow cytometry and antibody-based surface profiling^{110,111} offer additional means to enrich and classify EVs based on cellular source, which may be used alongside our reference EV proteome dataset to further refine classification and improve translational application. Our dataset offers a high-confidence resource of tissue- and cell-enriched protein features in circulating EVs that can be harnessed to guide the development of affinity-based enrichment strategies, improve computational deconvolution models, and support source-specific EV biomarker discovery. In future studies, combining these molecular leads with advanced single-EV technologies and disease-specific profiling could yield deeper insights into tissue-specific EV biogenesis, intercellular communication, and their roles in health and disease.

Tables:

Supplementary Table 11. Quantification of tissue / organ-associated proteins (Extended Data 2) and bioactive cargo (Extended Data 3) in pEVs

Supplementary Table 12. Cell signatures (Molecular Signature Database) in pEVs

Extended data 2. Quantification of tissue / organ-associated proteins in pEVs. **A.** Number of tissue-associated proteins (based on Human Protein Atlas (HPA) immunohistochemistry data (IHC)⁵⁶) quantified in pEV proteome. Abundance distribution of proteins are presented in lower panel. **B.** Number of cell-type-associated proteins (based on HPA IHC) quantified in pEV proteome. Abundance distribution of proteins is presented in lower panel. **C.** Number of tissue-associated proteins (based on HPA normalized transcript per million (nTPM)) quantified in pEV proteome. Abundance distribution of proteins is presented in lower panel. **D.** Number of organ-associated proteins reported in plasma using aptamers quantified in pEV proteome⁵⁷. Abundance distribution of these proteins is presented in lower panel. **E.** Gene Set Enrichment Analysis (using Molecular signature database¹³³ for cell signatures (C8 category) of proteins enriched in pEV vs NonEV proteomes. **F.** Enrichment of cell signatures based on single-cell sequencing studies of human tissue¹³³ in proteins enriched pEV vs NonEVs.

6. What is the ability of the mass spectrometry described by the authors to distinguish low abundance, highly bioactive entities, such as growth factor receptors, signal transducing molecules, transcription factors, RNA binding proteins (nothing is said in the paper about circulating RNA) and other important parts of the EV cargo?

We appreciate the reviewer's insightful question regarding the ability of MS to detect low-abundance, highly bioactive molecules in pEVs, including growth factor receptors, signal transduction molecules, transcription factors, and RNA-binding proteins. While these molecules are often present at lower concentrations compared to structural EV components, their functional relevance makes them critical to understanding EV-mediated signalling.

To systematically assess the presence of these functional molecules in pEVs, we manually curated and integrated multiple independent datasets that catalogue biologically significant protein classes. Specifically, we leveraged established resources for human transcription factors (PMID: 29425488, Human TFs Database), RNA-binding proteins (PMID: 21036867, RBPDB Database), kinases (PMID: 28056780, KinHub Database), and cell-surface receptors, transporters, and signalling proteins (PMID: 30373828). In addition, we incorporated Gene Ontology terms related to cytokine activity (GO:0005125), chemokine activity (GO:0008009), growth factor activity (GO:0008083), and signal transduction (GO:0007165) to ensure comprehensive coverage of bioactive EV-associated molecules.

Using this systematic approach, we identified a diverse range of low abundance yet functionally significant molecules that were significantly enriched in pEVs compared to non-EV plasma fractions (FDR < 0.05). Among the chemokines and cytokines, we detected CMTM5, CCL5, INHBE, PF4, PPBP, ADIPOQ, GMFG, GPI, MIF, TGFB1, TIMP1, and TYMP, all of which play essential roles in immune signalling, inflammation, and tissue remodelling. The significant enrichment of these molecules in pEVs reinforces the idea that EVs selectively package immunomodulatory cargo, which may contribute to intercellular communication and systemic regulation.

In addition to immunoregulatory molecules, our dataset contained a broad array of signal transduction proteins, further supporting the role of pEVs in mediating cell signalling and communication. Notably, we identified YWHAB, PRKCB, RAC2, RALA, RAP2B, RHEB, ANXA1, ARRB1, CAMK1, CD226, FAS, GNA13, GNAI2, GRB2, GNB1, MAPK1, PTK2B, STK4, TXNRD1, PRKACB, STAM, AKAP9, LYN, PECAM1, STAT3, and SRC, which are involved in diverse cellular pathways, including kinase signalling cascades, cytoskeletal remodelling, and immune activation. These findings underscore that pEVs serve as transporters of bioactive molecules, actively participating in regulating cellular processes beyond passive cargo transfer.

We also investigated RBPs, given their role in post-transcriptional gene regulation and RNA stability. Our analysis confirmed the enrichment of ACO1, EIF2S1, EIF3B, GAPDH, HNRNPK, MCTS1, PCBP1, PCBP2, PTBP1, RPS3, SYNCRIP, TMEM63A, and ZC3HAV1, suggesting that pEVs may act as vehicles for RNA regulatory complexes. This aligns with emerging evidence that EVs influence gene expression in recipient cells by transferring RBPs along with their associated RNA cargo, adding another layer of functional significance to their role in intercellular communication.

We also identified several TFs in pEVs, suggesting selective incorporation into vesicles as part of cell-to-cell communication mechanisms. We identified LTF, NFKB1, CHCHD3, NME2, PA2G4, STAT3, and STAT5B as significantly enriched in pEVs. The detection of NFKB1 and STAT3/STAT5B is particularly noteworthy given their central roles in immune regulation, inflammation, and cancer progression. Their presence in pEVs suggests that vesicle-mediated transfer of transcriptional regulators may contribute to systemic signalling and disease pathogenesis.

Additionally, a density plot of chemokine abundance revealed a distribution pattern closely resembling that of canonical EV proteins (e.g., CD proteins), supporting the notion that these molecules are integral EV

components. Furthermore, heatmap clustering confirmed that the majority of chemokines, growth factors, and signalling molecules exhibited selective enrichment in pEVs, paralleling patterns observed for well-established EV markers.

These findings collectively demonstrate our high-resolution mass spectrometry is capable of detecting low-abundance but biologically significant molecules in pEVs, including cytokines, chemokines, kinases, signal transduction proteins, RNA-binding proteins, and transcription factors. The fact that these molecules are significantly enriched in pEVs compared to non-EVs, and exhibit abundance patterns similar to canonical EV proteins, strongly supports active incorporation of these bioactive signalling molecules. While our study primarily focuses on proteomic and lipidomic characterization, these results highlight the functional potential of pEVs in transferring regulatory molecules and set the stage for future investigations into EV-mediated intercellular signalling in humans.

We have made the following inclusion in the Results (page 5, line number 169):

Furthermore, our high-resolution mass spectrometry quantified low-abundant but biologically significant molecules in pEVs, including signal transduction proteins, cytokines and chemokines, kinases⁵⁸, cell-surface receptors, transporters, and CD proteins⁵⁹, RNA-binding proteins⁶⁰, and transcription factors⁶¹ (**Extended Data 3A-B, Supplementary Table 11**). These include RNA binding proteins such as HNRNPK⁶² and PCBP2⁶³, TFs such as NME2⁶⁴, kinases such as SRC⁶⁵, chemokines such as TGFB1⁶⁶, and receptors such as integrins⁶⁷, previously reported to be functionally active cargo of EVs. These molecules (e.g., cytokines) exhibit similar abundance to canonical CD proteins typical to EVs, strongly supporting their active incorporation (**Extended Data 3C-D**).

We have made the following inclusion in the Discussion (page 12, line number 485):

Moreover, several bioactive cargo quantified in pEVs - such as TGFB1⁶⁶, the RNA binding protein HNRNPK⁶², integrins⁶⁷, SRC kinases⁶⁵ - are closely linked to lipid raft assembly and trafficking⁹⁷⁻¹⁰⁰. Importantly, classical lipid raft-modulating proteins such as CAV1 and CAVIN1 - known to regulate membrane microdomain composition and influence the EV proteome^{101,102} - regulate miRNA cargo loading into EVs by modulating HNRNPK within the membrane raft⁶². Elevated levels of EV-associated HNRNPK have also been detected in body fluids from patients with metastatic prostate and colorectal cancers, underscoring its clinical relevance.

Extended data 3. Diversity of bioactive cargo in pEVs. **A.** Number of different cargo-type quantified in pEV proteomes (in at least 3 out of 42 pEV proteomes), with box-plot in lower panel representing protein abundance distribution. TF, transcription factors; CDs, Cluster of Differentiation. Gene Ontology terms related to cytokine activity (GO:0005125), chemokine activity (GO:0008009), growth factor activity (GO:0008083), and signal transduction (GO:0007165) to ensure comprehensive coverage of bioactive EV-associated molecules **B.** Number of different cargo-type quantified in pEV proteomes (but significantly enriched in pEVs vs NonEVs, FDR <0.05), with box-plot in lower panel representing protein abundance distribution. **C.** Abundance distribution of cytokines/growth factors and CD proteins quantified in pEV proteome. **D.** Heatmap of cytokines and CD proteins quantified in pEV and NonEV proteomes. Grey boxes in the heatmap represent non-quantification.

7. One of the important sources of the molecular ‘noise’ among plasma samples and roadblocks in EV/NVP standardization are the pre-analytical variables, especially techniques that prevent **activation of platelets** during blood collection, preparation and storage (Coumans et al 2017). This is very difficult to achieve when operating across multiple institutions. The present study lacks quality controls as to elimination of activated platelets or induced platelet EVs in samples being analysed, which may result in the reported profiles being essentially those of contaminating platelets.

We appreciate the reviewer’s important question regarding the potential impact of pre-analytical variables, particularly platelet activation, on pEV profiles. As highlighted in previous studies (e.g., Coumans et al., 2017), platelet-derived EVs are a natural component of plasma and contribute to the circulating EV pool. We acknowledge that controlling pre-analytical factors across multiple institutions in multicenter studies is complex; however, we implemented stringent blood collection and processing protocols to minimize additional platelet activation and ensure consistency across sample cohorts.

To mitigate platelet activation, we followed established protocols for blood collection, handling, and storage, as recommended in previous studies (Hoshino et al., 2020; Jeppesen et al., 2019). Specifically, blood was collected and processed at room temperature immediately following venipuncture, and plasma was isolated using carefully controlled centrifugation steps to prevent additional platelet activation. Samples obtained from the Australian Red Cross (ARC) were processed in our lab, while samples from the EDCAD and AusDiab cohorts (now included in the revised study) were processed at the Baker Heart and Diabetes Institute, ensuring tightly controlled pre-analytical variables at each collection site. As an additional quality control measure, plasma samples exhibiting excessive cloudiness were discarded, and hemocytometer-based cell counting confirmed the absence of residual platelets or other contaminating cells in plasma prior to EV isolation.

Beyond pre-analytical controls, we conducted a comprehensive proteomic analysis of activated platelet EVs, identifying 235 proteins significantly enriched in platelet-derived EVs (p-adjusted <0.05) (PMID: 38556629). While platelet-derived EVs are naturally present in plasma, comparison of these proteins with our high-confidence pEV dataset revealed that of the 2,418 proteins conserved in pEV proteomes, only 8.6% (208 proteins) overlapped with activated platelet EV markers, indicating that our dataset is not dominated by platelet EV proteins (PMID: 38556629). Additionally, we examined the presence of non-platelet tissue-derived proteins in pEVs, including Cardiac Troponin I (TNNI3), a well-established cardiac muscle-specific marker commonly used in heart failure and ischemic heart disease diagnostics. This protein was robustly detected in plasma EVs (present in 70% of samples) but was absent in activated platelet EVs, further supporting the conclusion that pEVs capture tissue-derived molecular signatures beyond platelet-derived components. Notably, we previously reported EV proteomes from intact human heart tissue (PMID: 33861516), in which TNNI3 was similarly detected, reinforcing the biological specificity of pEV markers.

To further evaluate the contribution of platelet EVs to our dataset, we examined key platelet activation markers, such as platelet factor 4 (PF4/CXCL4), a protein highly enriched in activated platelet EVs (PMID: 38556629). While PF4 was detected in pEVs, it was absent in in vitro-derived EVs and did not rank among the most abundant pEV proteins, suggesting that while platelet EVs contribute to the pEV proteome, they do not dominate it. We also performed a comparative analysis using UpSet plots, demonstrating that pEVs contain distinct protein subsets (Set A and Set B) that are not shared with platelet EVs. Further, waterfall plots showed that pEV-unique proteins span a range of high, mid, and low-abundant molecules, reinforcing their biological relevance. Density plots confirmed that Set A and Set B proteins exhibit a distribution pattern that closely mirrors the overall pEV proteome, and KS tests indicated that these proteins are not rare outliers but rather integral components of pEVs.

Additionally, we explored the cellular, tissue, and organ signatures represented in our pEV proteomes, providing strong evidence for molecular diversity beyond platelet-derived proteins. Gene Ontology (GO) Cellular Component enrichment analysis revealed that pEV proteins are enriched in classical EV-associated compartments, such as endosomes, mitochondria, and focal adhesions, rather than platelet-specific compartments. Additionally, MSigDB C8 enrichment analysis identified non-platelet cell-type associations,

including endothelial cells, monocytes, and tissue-derived EV markers, highlighting that pEVs reflect a broad and diverse extracellular vesicle landscape.

We acknowledge that platelet EVs are a normal component of circulating plasma EVs, and their presence is expected in any plasma-derived EV study. However, our pre-analytical controls, proteomic comparisons, and enrichment analyses strongly suggest that pEV profiles are representative of diverse cellular origins and are not dominated by platelet EV contamination. The identification of multiple organ-derived proteins, non-platelet immune markers, and distinct pEV-unique protein subsets reinforces the biological relevance of our dataset. Future work using single-vesicle characterization approaches (e.g., high-resolution flow cytometry or super-resolution microscopy) may further refine our understanding of platelet EV contributions within the broader plasma EV pool.

We have made the following inclusion in the Results (page 5, line number 161):

Importantly, although activated platelet EV proteins such as CLEC1B, PF4 and PPBP (which we previously reported⁵³) contribute towards the pEV proteome landscape (**Supplementary Fig. 4, Supplementary Table 9**), we observed an abundance of non-platelet EV proteins (**Extended Data 1A-C**), suggesting that the pEV proteome represent diverse cellular source^{15,54,55} (**Extended Data 1D-E, Supplementary Table 10**).

Tables:

Supplementary Table 9. Detection of activated platelet EV proteins in current dataset

Supplementary Table 10. Non-platelet EV proteins in pEV proteomes (Extended Data 1)

Supplementary Figure 4. Quantification of platelet EV proteins in circulating EV proteome. A. Detection of platelet proteins (CLEC1B, PF4, PPBP) in platelet EVs in our previous report⁷⁷. **B.** Detection of platelet EV proteins (CLEC1B, PF4, PPBP) in in vitro EV, NonEV or pEV proteomes. **C.** Detection of platelet EV proteins (CLEC1B, PF4, PPBP) in non-transformed cell-derived EVs. **D.** Detection of platelet EV proteins (CLEC1B, PF4, PPBP) in pEVs and NonEV proteomes from AusDiab Validation set.

A

B

C

D

E

Extended data 1. Diverse cellular source attribution to pEV proteome. **A.** Distribution of activated platelet EV proteins in pEVs and in vitro EVs. Set A and Set B proteins represent proteins quantified exclusively in pEVs and in vitro EVs versus activated platelet EVs⁷⁷. **B.** Distribution of Set A and Set B proteins (abundance) in pEV proteome. **C.** Density plots of Set A and Set B protein abundance vs pEV proteome abundance. **D.** Gene Ontology (Cellular Component) analysis of Set A and Set B proteins. **E.** Gene Set Enrichment Analysis (using Molecular signature database⁷⁸ for cell signatures (C8 category)) of Set A and Set B proteins.

8. Fig 2F shows the molecular interconnectedness of molecular elements identified within the plasma EV cargo. This would make sense if the respective molecules interacted within the same EV, or at least within the

same EV parental cells, which remains undefined in the study. What is the significance of these connections if plasma EVs come from multiple cellular sources?

We appreciate the reviewer's thoughtful question regarding the significance of the molecular interconnectedness presented in Figure 2F, given that plasma EVs originate from multiple cellular sources. While we acknowledge that the interactions depicted in this network may not necessarily occur within the same EV, they nonetheless provide biologically meaningful insights into conserved functional modules and coordinated pathways within the plasma EV pool.

To address this, we systematically investigated these interconnected molecular features at the cellular proteome level or within specific EV subpopulations (e.g., CD9/CD81/CD63-positive EVs) (Extended Data 7). Our data demonstrate that pEV molecular features are highly conserved across EVs derived from multiple cell types, including SW480, SW620, MDA-MB-231, and LIM1863 cancer cell-derived EVs, as well as non-transformed fibroblast-derived EVs (human foreskin fibroblasts, human dermal fibroblasts, and human atrial and ventricular fibroblasts). Furthermore, these features are enriched across CD9-, CD81-, and CD63-positive human plasma EVs, suggesting that they are not exclusive to a single EV subpopulation but rather represent shared EV-associated molecular signatures across different EV types and sources. Density plot analyses further support this by confirming that these pEV features are abundant and widely distributed within EV proteomes, reinforcing their biological relevance.

Additionally, our findings align with previous work by Kalluri and colleagues, where these molecular features were shown to be highly conserved across EVs from 14 different cell types. Importantly, they exhibited relative enrichment in EVs compared to their parental cells, indicating that these molecules are not simply reflective of cellular background but are actively selected and enriched within EVs. This cross-EV conservation across multiple subpopulations (small vs. large EVs, CD9/CD81/CD63 EVs) suggests that these molecular features represent fundamental EV-associated signatures that likely contribute to shared functional roles in intercellular communication.

Even though plasma EVs originate from diverse cellular sources, analysing these interconnected molecular networks provides valuable insights into functional modules that reflect systemic biological processes. Many of these molecular interactions are likely involved in shared signalling pathways across multiple cell types, capturing common EV biogenesis pathways, metabolic adaptation, and extracellular communication.

While we recognize that single-vesicle proteomics or EV subtype-specific profiling would be required to precisely determine which molecular interactions occur within the same EV, the network analysis presented here offers a valuable systems-level perspective. These conserved molecular interactions provide functional insights into how EVs from multiple sources contribute to broader biological processes, and they may serve as important indicators of systemic physiological and pathological states detectable in plasma. As a next step, future investigations will apply single-vesicle analyses and EV subpopulation-specific profiling to further dissect these molecular relationships at higher resolution.

We have made the following inclusion in the Results (page 6, line number 213):

Upon closer inspection, these interconnected molecular features were also conserved across individual EV proteomes at the cellular level or within specific EV subpopulations (i.e., CD9/CD81/CD63-positive EVs) (**Extended Data 7A, Supplementary Table 15**). These molecular features were conserved in EVs from primary human fibroblasts (**Extended Data 7B**) and also enriched in EVs compared to the parental cell proteomes³⁰ (**Extended Data 7C**).

We have made the following inclusion in the Discussion (page 13, line number 512):

Nevertheless, bulk analyses can still offer insights into shared molecular programs and biologically conserved pathways. For instance, the molecular interconnectedness observed in our network analysis (Figure 2F) reflects functionally coherent modules enriched in the plasma EV pool. Although these interactions may not necessarily occur within the same EV, they nonetheless reveal coordinated molecular patterns that are conserved across EV subtypes and cell types. Importantly, we validated that many of these molecular features

are enriched in EVs derived from multiple cell types - including cancer cells and primary fibroblasts - as well as across CD9-, CD63-, and CD81-positive plasma EVs, suggesting they are not artifacts of averaging but representative of fundamental EV-associated signatures. Our findings align with prior studies demonstrating that such molecules are actively enriched in EVs relative to their cells of origin and conserved across cell-derived EVs³⁰. Thus, while single-vesicle resolution would offer even deeper insight, bulk EV analysis can still uncover biologically meaningful and source-representative features that contribute to systemic intercellular communication.

Table:

Supplementary Table 15. Conservation of interconnected molecular features (Fig. 2)

Extended data 7. Conservation of interconnected molecular features (Fig. 2F) across individual EV proteomes at cellular level or within specific EV subpopulations. A. Number of interconnected molecular features quantified in In vitro EVs (n=3 per cell line) or EV subtypes (CD81+, CD63+, and CD9+ EVs)

prevalent in the circulatory system⁷⁰. Quantification in MS data of individual biological replicates are shown. Lower panel shows distribution of protein abundance of these molecular features against all other proteins in their respective proteome. **B.** Number of interconnected molecular features quantified in small EVs (100,000 g ultracentrifuged EVs 100K) or large EVs (10,000 g ultracentrifuged EVs 10K) from non-transformed cells (fibroblasts). Lower panel shows distribution of protein abundance of these molecular features against all other proteins in their respective proteome. **C.** Number of interconnected molecular features quantified in small EVs from 14 cell lines as reported³⁰. Lower panel shows relative abundance of these molecular features in EVs versus their parental cell proteomes.

Reviewer #2:

In this manuscript, the authors perform mass spectrometry based proteomic and lipidomic analysis of a large number of extracellular vesicles (EVs) samples isolated from human plasma samples via density gradient ultracentrifugation (110 samples). Signatures of highly conserved proteins and lipids as well as proteins associated with the surface of sEVs were identified. The authors also propose ADAM10 and PS(36:1) as markers to distinguish EVs from non-EVs. The authors provide a comprehensive dataset of the proteome and lipidome of healthy plasma EVs which has not previously been described at this scale and with isolation methods enabling such purity. Several comments should be addressed, as outlined below.

1. The authors isolate EVs with density gradient (DG) wherein plasma is loaded on the top of the gradient; however, this can in some cases lead to incomplete separation of sEVs from 'contaminating' proteins, lipids, etc. The authors should confirm by bottom loaded DG that the identified enriched protein and lipid markers are present.

We agree with the reviewer that density gradient separation (DGS) does not achieve absolute EV purity, as some degree of co-isolation with non-EV components is inevitable. As suggested, we have now performed bottom-loaded DGS of plasma. For this, 500 μ L of plasma was carefully loaded at the bottom of a pre-formed discontinuous DGS using a long syringe to avoid disturbing the gradient. Ultracentrifugation was performed at 100,000 g for 18 hours (4°C), as in the top-loaded DGS method. After centrifugation, 12 equal fractions (1 mL each) were collected, diluted with 2 ml PBS, and subjected to another round of ultracentrifugation at 100,000 g for 1 hour (4°C). The resulting pellets were resuspended in 50 μ l PBS and analysed by Western blotting (Supplementary Fig. 1F).

Although bottom-loaded DGS successfully resolved EVs (as indicated by CD63 signal around fraction 6), we observed overlapping signals of non-EV components, specifically albumin (ALB, detected in all fractions) and high-density lipoprotein particles (APOA1, detected in fractions 5-12), co-fractionating with EV proteins. This suggests that, compared to top-loaded DGS, bottom-loaded DGS does not achieve the same level of resolution for intact plasma, likely due to the higher viscosity and protein density at the bottom of the gradient, which affects separation efficiency. Similar observations have been reported in previous studies (PMID: 32944179, PMID: 35524458).

Regarding the reviewer's concern about the incomplete separation of sEVs from 'contaminating' proteins and lipids, we acknowledge that no currently available protocol achieves absolute EV purity, particularly from plasma-derived samples. Moreover, a seminal study (PMID: 32944179) demonstrated that commonly reported plasma protein "contaminants" are, in fact, part of a naturally occurring protein corona surrounding EVs, a finding that has reshaped our understanding of circulating EVs. This distinction between co-isolated plasma proteins and the intrinsic protein corona of EVs presents a significant challenge in defining what should be considered a "contaminant."

Given these considerations, our aim was to achieve a sufficiently high level of EV purity with minimal non-EV particle contamination, rather than absolute isolation. The Western blots in Figure 1 (b, f-h) and cryo-EM images in Figure 1c confirm that our preparations are enriched for EVs with minimal detectable non-EV

particles. Moreover, these protein features are also present in EV isolated from cells cultures In vitro. Additionally, we avoided pelleting EVs under excessive high-speed centrifugation, which could induce shear stress and protein aggregation, a common issue when isolating plasma EVs.

The high purity of our EV preparations is further supported by MS-based proteomics, which identified over 5,000 proteins, predominantly low-abundance EV proteins. If significant levels of non-EV proteins were present, they would have severely interfered with the detection and quantification of EV-specific proteins and lipids, as commonly observed in ultracentrifugation-alone protocols. To further confirm purity, we compared the relative abundances of established EV markers, such as SDCBP (syntenin-1) and phosphatidylserine, between EV and non-EV fractions from the same plasma, reinforcing that our EV isolates contain minimal non-EV contaminants.

Thus, while we acknowledge that DGS cannot completely separate sEVs from plasma proteins and lipids, we emphasize that our approach achieves a high degree of EV purity, as also highlighted by the reviewer as a key strength of our study.

We have made the following inclusion in the Results (page 4, line number 135):

To assess separation efficiency, we also performed bottom-loaded DGS by carefully loading plasma at the bottom of the gradient before ultracentrifugation (**Supplementary Fig. 1F**). While EVs were recovered in similar fractions as in the top-loaded approach (CD63 signal in fraction 6), bottom-loaded DGS resulted in greater co-fractionation of non-EV components, including ALB and APOA1, across multiple fractions. This suggests that, compared to top-loaded DGS, bottom-loaded DGS does not achieve the same resolution in plasma samples.

Supplementary Figure 1F. Characterization of EVs from human plasma. F. Bottom-up density gradient separation (DGS) of human plasma (0.5 ml). Western blot analysis of twelve DGS fractions (vol:vol matched) of human plasma with antibodies against indicated proteins.

2. Comparisons between healthy plasma sEVs to sEVs from cell lines are included, but the cell lines employed are predominantly cancer cell lines. The authors should address whether there is greater concordance with non-transformed cell lines, or the features are conserved between non-transformed/healthy and cancer cell-derived EVs. Similarly, the only disease condition included is ECAD; thus, conclusions about applicability to disease states should be tempered.

As suggested by the reviewer, we have now examined the proteome and lipidome profiles of EVs isolated from non-transformed cell lines (**namely, human primary fibroblasts and endothelial cells**) to assess the conservation of pEV protein and lipid features. To isolate EVs from these cells, we subjected conditioned media to differential centrifugation (500 g for 5 min and 2,000 g for 10 min), followed by concentration using Amicon 100 kD filters (2,000 g). The concentrated supernatant was further centrifuged at 10,000 g (30 min) to pellet large EVs (10K EVs), while the resultant supernatant underwent ultracentrifugation at 100,000 g (1 h) to pellet small EVs (100K EVs). The isolated EVs were washed in PBS (1 mL, 10,000 g for large EVs and 100,000 g for small EVs) before resuspension in 50 μ L PBS and subsequent proteomic and lipidomic analyses. In assessing the conservation of pEV protein features, we observed that the proteomes of fibroblast-derived EVs (10K and 100K) were distinct from those of their parental fibroblasts, while endothelial cell-derived EVs (100K) exhibited even greater divergence. Among the 182 pEV protein features, 135 were detected in this dataset, with 43 proteins showing 100% conservation in both fibroblast- and endothelial-derived EVs, and an

additional 85 proteins being conserved in at least 7 out of 12 EV proteomes. Notably, conserved proteins include ADAM10, integrins, Rabs, Annexin A5, CD markers (CD9, CD44), and SDCBP (which is well-established EV marker). Proteins not detected in this dataset primarily included immunoglobulins and complement proteins, which are either plasma-specific or components of the plasma EV protein corona, further supporting the notion that these proteins are characteristic of circulating EVs rather than cell line-derived EVs.

Regarding disease relevance, we acknowledge that our study primarily focused on EDCAD as a model system. While our findings demonstrate the robust identification of pEV features in healthy plasma, further validation in additional disease contexts will be necessary to fully assess the generalizability of these markers across different pathophysiological conditions. Nonetheless, our new analyses provide strong evidence that a substantial subset of pEV protein features is conserved in non-transformed cell-derived EVs, further reinforcing their biological relevance beyond cancer-derived EVs.

To extend our findings beyond protein markers, we performed lipidomic analysis, interrogating 905 lipids to assess the conservation of pEV lipid features in non-transformed cell-derived EVs. In this analysis, we re-quantified 323 lipids from the discovery dataset, which included 12 out of 52 pEV lipid features and 56 out of 114 NonEV lipid features. We analysed two independent plasma validation sets from AusDiab: Validation Set 1 (n=10 plasma samples, with pEVs and NonEVs) and Validation Set 2 (n=12 plasma samples, including pEVs, NonEVs, and lipidomic analysis of neat plasma itself). Additionally, we performed lipidomic analysis of EVs from non-transformed fibroblasts (n=4) and endothelial cells (n=4), isolating small (n=4 per cell type) and large EVs (n=4 per cell type).

In this dataset, we first confirmed that pEVs exhibit distinct lipidomic profiles compared to NonEVs and neat plasma, with clear clustering patterns observed between pEVs in PCA and Pearson correlation matrix analysis. Next, we performed ANOVA (FDR-controlled) and retained only lipids with FDR < 0.05 for downstream comparisons.

To further visualize these findings, we generated a heatmap of the pEV and NonEV lipid features, which clearly demonstrated differential lipid expression between pEVs and NonEVs across independent samples. Additionally, we selected two key lipid features of interest—PS (36:1) as a pEV-enriched lipid and CE (18:0) as a NonEV-enriched lipid—and evaluated their expression using bar plots. PS (36:1) expression was significantly higher in pEVs compared to NonEVs and plasma, with no significant difference between pEVs and in vitro EVs, indicating its conservation across EV sources. Conversely, CE (18:0) expression was significantly higher in NonEVs and plasma, with no difference between NonEVs and neat plasma, further supporting its classification as a NonEV-enriched lipid feature.

We have made the following inclusion in the Results (page 7, line number 281):

To assess the broader biological relevance of pEV protein features beyond plasma-derived EVs, we analysed EV proteomes from non-transformed human fibroblasts and endothelial cells (**Extended Data 10, Supplementary Table 25**). Among the 182 pEV protein features, 135 were detected in this dataset, with 43 proteins showing 100% conservation across fibroblast- and endothelial-derived EVs. Conserved proteins included ADAM10, integrins, Rabs, Annexin A5, CD markers (CD9, CD44), and SDCBP, reinforcing their role as core EV components across different biological sources. Proteins absent from non-transformed cell-derived EVs primarily included immunoglobulins and complement proteins, supporting their plasma-specific nature or association with the EV protein corona.

We have made the following inclusion in the Results (page 9, line number 350):

The relative abundance of these features was also conserved in EVs from primary human fibroblasts and endothelial cells (**Extended Data 12**). These findings highlight the robustness of our identified EV lipid markers across independent populations and multiple EV sources.

We have made the following inclusion in the Results (page 10, line number 423):

In our validation set, PS(36:1) lipid was significantly enriched in pEVs and In vitro EVs, reinforcing its role as a conserved pEV lipid marker (**Extended Data 13**). In contrast, CE(18:0) lipid was enriched in NonEVs and neat plasma. These findings confirm that pEV protein and lipid markers are conserved beyond plasma-derived EVs, reinforcing their biological significance and potential as robust EV classification markers.

Extended data 10. Conservation of pEV protein features in EVs from non-transformed cells. A. Workflow for obtaining small EVs (100K) or large EVs (10K) from conditioned media of either human primary fibroblasts or human primary endothelial cells. **B.** Pearson correlation of quantified proteins in indicated proteomes using mass spectrometry. Fibroblasts (whole cell lysates) were used as reference proteome for cells. **C.** pEV protein features quantified in small EVs (100K) or large EVs (10K) released by

fibroblasts or endothelial cells. **D.** Heatmap indicates detection of pEV protein features in small EVs (100K) or large EVs (10K) released by fibroblasts or endothelial cells.

Extended data 12. Conservation of pEV lipid features in EVs from AusDiab validation set and in non-transformed cells. **A.** Principal component analysis of quantified lipids in non-transformed EVs (in vitro Small EVs or in vitro Large EVs) or pEVs, NonEVs or neat plasma from AusDiab validation set. **B.** Heatmap depicts distribution of pEV and NonEV lipid features quantified in AusDiab validation set (pEVs, NonEVs or neat plasma) and in non-transformed cell-derived EVs (in vitro Small EVs or in vitro Large EVs).

Extended data 13. Relative abundance of pEV lipid features PS (36:1) and CE (18:0) in pEVs lipidome datasets. Lipid features in AusDiab validation set (pEVs, NonEVs or neat plasma) and in non-transformed cell-derived EVs (in vitro Small EVs or in vitro Large EVs).

Methods:

Human fibroblasts and endothelial cells (80% confluent in T75 culture flasks) were cultured in DMEM supplemented with 0.5%v/v insulin transferrin selenium (Invitrogen) and 1% Pen/Strep) to generate conditioned media. To isolate EVs, we subjected conditioned media to differential centrifugation (500 g for 5 min and 2,000 g for 10 min at 4 °C), followed by concentration using Amicon® Ultra Centrifugal Filter, 100 kDa MWCO (low speed centrifugation at 2,000 g at 4 °C). The filter was washed with 5 time with 1 ml ice-cold PBS. The concentrated retentate was retrieved and further centrifuged at 10,000 g (30 min at 4 °C) to pellet crude large EVs (10K EVs), while the resultant supernatant underwent ultracentrifugation at 100,000 g (1 h at 4 °C) to pellet crude small EVs (100K EVs). The isolated EVs were washed in PBS (1 ml, 10,000 g for large EVs and 100,000 g for small EVs) before resuspension in 50 µl PBS and stored at -80°C until further use (subsequent proteomic and lipidomic analyses).

Tables:

Supplementary Table 25. Quantification of pEV protein features in EVs from non-transformed cells

Supplementary Table 32. Lipidomics analysis in EVs from AusDiab validation set and non-transformed cells.

Supplementary Table 33. VSN-normalized lipid abundances in EVs from AusDiab validation set and non-transformed cells.

3. Upon identifying the 182 pEVs signatures, the abundance differences between pEVs and NonEVs need to be considered in addition to the p-values. Although certain proteins exhibit significant p-values, their abundance differences are trivial, casting doubt on their classification as pEV signatures.

To address this, we would like to clarify the robustness of our data analysis and selection criteria. We demonstrate that all 182 pEV protein features exhibit 100% conservation in our discovery pEV proteome dataset (Fig. 2-3, Supplementary Table 7). In contrast, 97 of these features were not detected in 50% or more of the NonEVs proteome datasets, reinforcing their specificity to pEVs. This highlights that, beyond statistical significance and fold change, we applied an additional conservation-based filtering criterion to ensure that these proteins are consistently associated with pEVs. The log₂ fold change values used in our analysis represent the difference in protein abundance between pEVs and NonEVs, where positive values indicate enrichment in pEVs. In Figure 3, we demonstrate that the relative abundances of these proteins successfully differentiate pEVs from NonEVs. Notably, 179 out of 182 pEV proteins exhibit a log₂ fold change greater than 0.58, which corresponds to a >1.5-fold increase in linear scale, a commonly accepted threshold for biological significance in proteomics studies. This further reinforces that our identified pEV markers are not only statistically significant but also biologically meaningful.

To ensure statistical rigor, we applied Benjamini-Hochberg (BH) correction to control for false discovery rates (FDR), reducing the likelihood of false-positive identifications. Across both pEV (182 proteins) and NonEVs (42 proteins) features, 220 protein features exhibit an FDR < 0.001. This highly stringent threshold indicates that fewer than 0.1% of the identified features are expected to be false positives, demonstrating strong statistical confidence in these protein identifications. Importantly, the relative abundances of pEV protein features were not only statistically robust but also functionally relevant. Using machine learning-based classification models, we show that these features effectively distinguish pEVs from NonEVs.

To further validate our findings, we assessed the presence of these pEV markers in an external plasma cohort (see response to Point 4) and in non-transformed cell-derived EVs (Extended Data 2). Additionally, we provide functional validation, demonstrating that ADAM10, one of our proposed pEV markers, is detectable on pEVs using flow cytometry. Furthermore, we show that ADAM10 can be used for immuno-capture-based isolation of EVs from plasma, highlighting the potential utility of our identified markers in experimental and clinical applications.

These combined analyses provide strong evidence that our identified pEV markers are not only statistically significant but also biologically relevant and reproducible across independent datasets, machine learning models, and experimental approaches.

We have made the following inclusion in the Results (page 5, line number 199):

For pEV protein feature selection, we first selected proteins that were 100% conserved in pEV proteomes (i.e., present in all 38 proteome datasets, Category 1 proteins), which resulted in 259 proteins. Next, of these 259 proteins, we selected proteins with fold change >1.5 and FDR < 0.001 in pEV vs. NonEVs, resulting in 182 proteins, which we refer to as ‘pEV protein features’ (Fig. 2D, Extended Data 5, Supplementary Table 7). For NonEV protein feature selection, we first selected proteins that were 100% conserved in NonEV proteomes (i.e., present in all 42 proteome datasets), which resulted in 114 proteins. Next, of these 114 proteins, we selected proteins with fold change < -1.5 and FDR < 0.001 in pEV vs. NonEVs, resulting in 42 proteins, which we refer to as ‘NonEV protein features’ (Extended Data 6, Supplementary Table 7).

Tables:

Supplementary Table 20. Adjusted p values and fold change of proteins between pEVs versus NonEVs in discovery set (Figure 2) and AusDiab set.

Supplementary Table 34. Adjusted p values and fold change of lipids between pEVs versus NonEVs in discovery set (Figure 4) and AusDiab set.

4. The identified pEV markers require experimental validation of their abundance in external cohorts.

As requested by the reviewer, we have now performed experimental validation of pEV marker abundance in an independent external cohort. To do this, we isolated pEVs and NonEVs using top-down DGS from plasma samples obtained from the Australian Diabetes, Obesity, and Lifestyle Study (AusDiab), a large, population-based cohort study. We analysed matched samples (pEVs, n=12; NonEVs, n=12) using our proteomics and lipidomics pipelines to assess the reproducibility of our findings.

In this external cohort, we first confirmed that pEVs and NonEVs exhibit distinct molecular profiles, as demonstrated in figure below (Panels A-C), with clear separation based on principal component analysis (PCA) and Pearson correlation matrix. Importantly, proteins significantly enriched in pEVs (Panels D-F) included well-established EV markers and Gene Ontology terms, further supporting the specificity of our approach.

Regarding pEV markers (Extended data 8), we quantified 177 out of 182 pEV protein features and all 42 NonEV protein features in this dataset (Panels A-B). Statistical analysis using ANOVA t-tests with Benjamini-Hochberg (BH) correction showed that 156 pEV proteins (87.36%) were significantly enriched in pEVs compared to NonEVs (FDR < 0.05, log₂ fold change > 0.67, equivalent to a >1.6-fold increase in linear scale).

Similarly, 39 out of 42 NonEVs proteins (92.86%) were significantly enriched in NonEVs compared to pEVs (FDR < 0.05, log₂ fold change < -0.67), supporting their specificity as NonEVs markers.

To further evaluate the reproducibility of our findings, we compared log₂ fold-change values for all significant pEV and NonEVs protein features (FDR < 0.05 in both datasets) between our discovery dataset (Figure 3C) and the new validation dataset (AusDiab cohort). This yielded a Pearson correlation of $r = 0.851$ ($p < 2 \times 10^{-16}$), indicating a strong and highly significant positive correlation between the two datasets. Linear regression further showed a slope of 0.7 (95% CI: 0.635–0.757) and $R^2 = 0.72$, meaning that 72% of the variance in the validation dataset can be explained by the discovery dataset. The strong correlation and statistical significance confirm the high reproducibility of our identified markers, reinforcing their biological relevance across independent cohorts.

These findings provide compelling evidence that our identified pEV markers are statistically robust, biologically relevant, and reproducible across independent populations.

To extend our findings beyond protein markers, we performed lipidomic analysis, interrogating 905 lipids to assess the conservation of pEV lipid features in AusDiab plasma samples as well as non-transformed cell-derived EVs (Supplementary Fig. 12-13, Extended data 12-13). In this analysis, we re-quantified 323 lipids from the discovery dataset, which included 12 out of 52 pEV lipid features and 56 out of 114 NonEV lipid features. We analysed two independent plasma validation sets from AusDiab: Validation Set 1 (n=10 plasma samples, with pEVs and NonEVs) and Validation Set 2 (n=12 plasma samples, including pEVs, NonEVs, and lipidomic analysis of neat plasma itself). Additionally, we performed lipidomic analysis of EVs from non-transformed fibroblasts (n=4) and endothelial cells (n=4), isolating small (n=4 per cell type) and large EVs (n=4 per cell type).

In this dataset, we first confirmed that pEVs exhibit distinct lipidomic profiles compared to NonEVs and neat plasma, with clear clustering patterns observed between pEVs in PCA and Pearson correlation matrix analysis. Next, we performed ANOVA (FDR-controlled) and retained only lipids with FDR < 0.05 for downstream comparisons. We then assessed the reproducibility of lipid fold changes (pEVs vs NonEVs for 323 lipids (present in discovery dataset) for across datasets, highlighting pEV and NonEV lipid features. When comparing the discovery dataset with Validation Set 1, we obtained a slope of 1.21, $R^2 = 0.81$, Pearson $r = 0.901$, and p-value < $2e-16$, demonstrating strong correlation and reproducibility. A similar trend was observed for Validation Set 2, with a slope of 1.21, $R^2 = 0.77$, Pearson $r = 0.878$, and p-value < $2e-16$. These findings confirm that pEV lipid features are highly reproducible across datasets.

To further visualize these findings, we generated a heatmap of the pEV and NonEV lipid features, which clearly demonstrated differential lipid expression between pEVs and NonEVs across independent samples. Additionally, we selected two key lipid features of interest—PS (36:1) as a pEV-enriched lipid and CE (18:0) as a NonEV-enriched lipid—and evaluated their expression using bar plots. PS (36:1) expression was significantly higher in pEVs compared to NonEVs and plasma, with no significant difference between pEVs and in vitro EVs, indicating its conservation across EV sources. Conversely, CE (18:0) expression was significantly higher in NonEVs and plasma, with no difference between NonEVs and neat plasma, further supporting its classification as a NonEV-enriched lipid feature.

These findings provide compelling evidence that our identified pEV protein and lipid markers are statistically robust, biologically relevant, and reproducible across independent populations, reinforcing their potential as reliable EV biomarkers.

We have made the following inclusion in the Results (page 7, line number 246) for proteomic section:

To validate our identified pEV markers, we analyzed plasma from an independent external cohort (AusDiab, n=12 pEVs, n=12 NonEVs) using the same EV isolation and proteomic pipeline (**Supplementary Table 17**). Comparative proteomic analysis revealed that 177 out of 182 pEV protein features and all 42 NonEV protein features were detected, with 87.4% (156/182) of pEV proteins and 92.9% (39/42) of NonEV proteins

significantly enriched in their respective fractions (FDR < 0.05) (**Supplementary Table 18, Extended Data 8**), with similar enrichment in Gene Ontology pathways in our discovery cohort (**Supplementary Table 19**).

Comparative analysis of fold change of pEV and NonEV protein features between the discovery and validation datasets (**Supplementary Table 20**) demonstrated a strong positive correlation (Pearson $r = 0.851$, $p < 2 \times 10^{-16}$), confirming the reproducibility of pEV protein features across independent populations (**Figure 3C**).

Extended data 8. Heatmap of pEV and NonEV protein features in AusDiab validation set.

Figure 3. Conservation of circulating EV protein features. **C.** Scatter plot of fold change (FDR < 0.05) EV protein features and NonEV protein features in discovery set and validation set (AusDiab set). Rest of the proteins are indicated with grey points. Pearson $r = 0.851$, $p < 2 \times 10^{-16}$. **D.** Density plot of proteins in pEV proteomes from individuals with either positive CAC score (CAC group) or zero CAC score (Healthy group). CAC_DOWN: proteins significantly (FDR < 0.05) downregulated in pEVs from CAC vs Healthy. CAC_UP: proteins significantly (FDR < 0.05) upregulated in pEVs from CAC vs Healthy. EV feature: 182 pEV protein features. NS_proteins: proteins with similar abundance between pEVs from CAC vs Healthy.

Tables:

Supplementary Table 17. Proteome analysis of pEVs, NonEVs and neat plasma from AusDiab validation cohort and EVs from non-transformed cells

Supplementary Table 18. Differential abundant protein analysis of pEVs versus NonEVs in AusDiab set

Supplementary Table 19. Gene Ontology analysis of differential abundance analysis of proteins identified in pEVs versus NonEVs in AusDiab set

Supplementary Table 20. Adjusted p values and foldchange of proteins between pEVs versus NonEVs in discovery set (Figure 2) and Ausdiab set.

We have made the following inclusion in the Results (page 9, line number 350) for lipidomic section:

To further assess the conservation of pEV lipid features, we performed lipidomic profiling of pEVs and NonEVs isolated from two sets of plasma from AusDiab study ($n=10$ plasma samples for validation set 1 & $n=12$ plasma samples for validation set 2) and non-transformed (primary human fibroblasts and endothelial cells) cell-derived EVs (**Supplementary Table 32, Extended Data 12, Supplementary Fig. 12**). In this validation lipidomic workflow, we re-quantified 12 out of 52 pEV lipid features and 48 out of 114 NonEV lipid features and report their differential abundance analysis (**Supplementary Table 32-33, Supplementary Fig. 13**). Comparison of fold changes between these lipid features across the discovery and validation datasets showed a strong correlation (**Figure 5G, Supplementary Table 34**, Pearson $r = 0.901$, $p < 2 \times 10^{-16}$ for validation set 1, and Pearson $r = 0.878$, $p < 2 \times 10^{-16}$ for validation set 2), confirming the reproducibility of pEV lipid features. The relative abundance of these features was also conserved in EVs from primary human fibroblasts and endothelial cells (**Extended Data 12, Supplementary Fig. 13**). These findings highlight the robustness of our identified EV lipid markers across independent populations and multiple EV sources.

Extended data 12. Conservation of pEV lipid features in EVs from AusDiab validation set and in non-transformed cells. **A.** Principal component analysis of quantified lipids in non-transformed EVs (in vitro Small EVs or in vitro Large EVs) or pEVs, NonEVs or neat plasma from AusDiab validation set. **B.** Heatmap depicts distribution of pEV and NonEV lipid features quantified in AusDiab validation set (pEVs, NonEVs or neat plasma) and in non-transformed cell-derived EVs (in vitro Small EVs or in vitro Large EVs).

Figure 5. Conserved lipid features of circulating EVs in humans. G. Scatter plot of fold change (FDR < 0.05) EV lipid features and NonEV lipid features in discovery set and two validation sets (AusDiab set). Rest of the lipids are indicated with grey points.

Tables:

Supplementary Table 32. Lipidomics analysis in EVs from AusDiab validation set and non-transformed cells.

Supplementary Table 33. VSN-normalized lipid abundances in EVs from AusDiab validation set and non-transformed cells.

Supplementary Table 34. Adjusted p values and foldchange of lipids between pEVs versus NonEVs in discovery set (Figure 4) and Ausdiab set.

5. It is not clear why the authors specifically employ Naïve Bayes for proteins and Neural Network for lipids. Did the authors compare the performance of these models to other existing models?

We appreciate the reviewer's insightful question regarding our use of Naïve Bayes for protein classification and Neural Networks for lipid classification. Our primary aim was not to develop a final predictive model but to provide a proof-of-principle demonstration that the identified protein and lipid features can effectively distinguish extracellular vesicle (EV) components from non-EV material.

To support robust model selection, we employed the caretEnsemble framework, which allows for the systematic training and evaluation of multiple machine learning algorithms in parallel. For both the proteomics and lipidomics datasets, we tested a range of classifiers, including Random Forest, Support Vector Machines (linear, svmLinear), Partial Least Squares (pls), k-Nearest Neighbours (knn), C5.0, Naïve Bayes, and Neural Networks (nnet). Model performance was assessed using cross-validation and compared across accuracy, sensitivity, specificity, and area under the ROC curve.

While several models were able to distinguish EVs from NonEVs in training and validation sets, performance in the independent test set varied (Figure below; panels A–B for cross-validation, and panels C–D for test set). For the proteomic data, Naïve Bayes (Accuracy: 97%, panel C) outperformed other models (e.g., Neural Network: 59% Accuracy), likely due to its robustness in high-dimensional, sparsely distributed datasets - where the assumption of feature independence helps mitigate overfitting. In contrast, the lipidomics dataset benefited from a Neural Network classifier (Accuracy: 97%, kappa: 89%), which outperformed other models, including Naïve Bayes, likely due to the presence of non-linear relationships in lipidomic data that are better captured by neural architectures. Additionally, Neural Networks yielded higher probability scores for EV classification compared to linear models such as SVM.

Importantly, our machine learning analysis was designed to illustrate that the EV-enriched protein and lipid features exhibit distinct, classifiable patterns when compared to non-EV counterparts. We anticipate that further optimization, including feature selection, hyperparameter tuning, and ensemble modelling, could enhance performance, and we consider this a valuable direction for future studies.

6. It is not necessary to employ machine learning to identify the pEV markers in this study as we can obtain the same information based on comparison analysis. For example, ADAM10 is one of the proteins exclusively detected in pEV samples as shown in Figure 3.

We appreciate the reviewer's perspective that comparative analysis alone can highlight pEV markers such as ADAM10. Indeed, the presence/absence of certain proteins (e.g., ADAM10 in pEVs) provides strong evidence for their enrichment. However, our application of machine learning was not intended as a replacement for standard comparison analysis but rather as a complementary approach to identify broader multi-marker patterns in pEV classification.

While individual markers such as ADAM10 can be identified based on their exclusive presence in pEVs, extracellular vesicle (EV) composition is inherently heterogeneous, and many biologically relevant EV features are not strictly binary (presence/absence) but rather differentially enriched. Machine learning enables us to leverage multi-marker patterns across the entire proteome and lipidome, detecting subtle but statistically meaningful relationships between features that may not be evident through standard comparison analysis alone.

Additionally, machine learning allowed us to assess the collective discriminatory power of pEV-associated features, moving beyond individual proteins toward a more systems-level understanding of pEV composition. By systematically evaluating multiple classification models using caretEnsemble, we demonstrated that Naïve Bayes (proteins) and Neural Networks (lipids) effectively captured multi-feature signatures that distinguish pEVs from NonEVs. This does not mean that individual markers like ADAM10 lack importance, but rather that a combination of features can provide a more robust classification framework for identifying pEVs across different sample conditions.

Thus, our use of machine learning serves as a proof of principle, illustrating that pEV proteomic and lipidomic signatures contain information that can be used for classification beyond binary presence/absence comparisons. This highlights the potential of data-driven approaches for improving the characterization of EVs in future studies, while still recognizing the value of traditional comparison methods for identifying individual markers.

7. For the identified pEV markers, the authors need to explain why they were not universally identified in other studies and the potential impact of differences in the techniques on EV isolation. For example, ADAM10 was not universally detected in both human plasma and cell EV samples in the Hoshino et al study (Cell, 2020).

While many of the identified pEV protein features, including ADAM10, have been reported in previous EV studies, their detection is often influenced by differences in EV isolation techniques, dataset variability, and analytical depth. For example, ultracentrifugation-based methods, such as those used in Hoshino et al., 2020, may lead to co-isolation of non-vesicular proteins, potentially masking lower-abundance or selectively enriched EV proteins. This is also true in our case when analysing proteomes of ultracentrifuged EVs. In contrast, our high-resolution DGS-based separation combined with deep proteomic profiling provides enhanced specificity for distinguishing EV-associated proteins from plasma background, likely contributing to the increased detection of certain markers in our dataset. These methodological differences underscore the importance of standardized EV isolation approaches in ensuring reproducibility across studies.

8. In Fig. 2D, Extended data 2, Supplementary Table 9, amongst the 259 conserved proteins, the authors further distinguish 182 proteins present uniquely or in significantly higher abundance in pEVs compared to NonEVs and was named as 'pEV protein features'. The criteria or threshold to distinguish these 182 proteins should be included.

Similarly, in Extended data 3A, criteria- 2) proteins quantified in over 70% of NonEV samples but less than 30% of pEVs samples (14 proteins) will be included in 'NonEV protein features' but there was no clear dividing line that highlighted 70% of NonEV samples and 14 proteins.

To ensure clarity, we have made the following changes in the manuscript:

Point 1: The criteria or threshold to distinguish 182 pEV protein features: We first selected proteins that were 100% conserved in pEV proteomes (i.e. in all 38 proteome datasets, Category 1 proteins), which resulted in 259 proteins. Next, of these 259 proteins, we selected proteins with fold change >1.5 (p value <0.0001) in pEV vs NonEV, resulting in 182 proteins- we refer to them as 'pEV protein features'.

Point 2: The criteria or threshold to distinguish 42 NonEV protein features: We first selected proteins that were 100% conserved in NonEV proteomes (i.e. in all 42 proteome datasets), which resulted in 114 proteins. Next, of these 114 proteins, we selected proteins with fold change <-1.5 (p value <0.0001) in pEV vs NonEVs, resulting in 42 proteins- we refer to them as 'NonEV protein features'.

We have made the following inclusion in the Results (page 6, line number 199):

For pEV protein feature selection, we first selected proteins that were 100% conserved in pEV proteomes (i.e., present in all 38 proteome datasets, Category 1 proteins), which resulted in 259 proteins. Next, of these 259 proteins, we selected proteins with fold change >1.5 and p-val <0.0001 in pEV vs. NonEVs, resulting in 182

proteins, which we refer to as ‘pEV protein features’ (Fig. 2D, Extended Data 5, Supplementary Table 7, FDR < 0.05 as reported in Supplementary Table 20).

For NonEV protein feature selection, we first selected proteins that were 100% conserved in NonEV proteomes (i.e., present in all 42 proteome datasets), which resulted in 114 proteins. Next, of these 114 proteins, we selected proteins with fold change < -1.5 and p-val < 0.0001 in pEV vs. NonEVs, resulting in 42 proteins, which we refer to as ‘NonEV protein features’ (Extended Data 6, Supplementary Table 7, FDR < 0.05 as reported in Supplementary Table 20).

Reviewer #3:

In this manuscript, Greenings and coworkers describe the mass spectrometry-based proteomic and lipidomic analyses of the Extracellular vesicles (EVs) in human plasma. They first performed high-resolution density gradient fractionation of over 110 human plasma samples to isolate circulating EVs, and systematically construct their quantitative proteome (4500 proteins) and lipidome (829 lipids) landscapes. The analysis led them to believe that a highly conserved panel of 182 proteins (ADAM10, STEAP23, STX7) and 52 lipids (PS, PIPs, Hex2Cer, PAs) were discovered. They also claim that a set of 42 proteins and 114 lipids features that served as hallmark features of non-EV particles in plasma have been established. Finally, they pinpoint that PS(36:1) and CE(18:0) are the conserved EV biological markers that precisely differentiates between EV and non-EV particles

This is a gigantic work, containing tons of data for readers to digest; and the goal of the study is ambitious. Frankly, I am overwhelmed by the data that have been presented. Considering myself as an experienced researcher in the lipid field (less is proteomics), it is more appropriate that my critique on this manuscript be focused on the lipidomic analysis, rather than the entire content (e.g., not comment on the proteomic analysis). My comments are listed below.

1. Is the claim of the lipidome (829 lipids, 40 lipid classes) constructed credible (big number is not necessary good)? The LC/MS method as applied by the authors (Agilent QQQ instrument for qualitative and quantitative analysis) appeared to have driven the instrument to the brink of mission impossible. It is not likely that a single LC/MS analysis can result in such a huge number of lipid species, leaving alone a precise quantitation. For tissue lipidome such as plasma, the intensities of lipid species varied drastically among various lipid families and various lipid species in the same lipid family. Are they in the linear range for quantitation. How confident is the structure identification?

We agree with the reviewer that a larger number is not necessarily better. However, by leveraging scheduled multiple reaction monitoring along with chromatography, obtaining measurements for 829 species is not outside the realm of possibility and this method was developed for this project. We have added a supplementary table (Supplementary Table 40) highlighting the fragmentation and conditions for each of the lipid classes. The broad range of coverage comes from chromatographic separation of lipid isomers, where a single MRM transition can encompass multiple species, and classes, e.g. our single transition 772.6 / 184.1 covers 5 species including two isomers of PC 35:2, two isomers of PC O-36:2 and 1 isomer of PC P-36:1 in plasma / fractions.

We have adapted this method to handle the small lipid content found in isolated EVs and fractions. While we agree that the absolute quantitation would be near-impossible (we do not have standards for all 829 lipids), the relative quantification would enable comparisons between

For structural identification and confidence, we report nomenclature based on what we can infer from the experiments conducted here. For example, the PE(P) class fragment in an acyl and alkenyl specific manner (DOI: 10.1016/j.jasms.2004.07.009), thus is reported with full *sn* isomer acyl composition e.g. PE(P-16:0/18:2), while many of the other glycerophospholipid classes are measured as sum compositions (e.g. PC 36:2). Any isomeric separation would be denoted as [a] or [b] until further characterised.

2. The designated lipid species as indicated in nearly the entire paper using, for example, PS(36:1), which only defines the total chain length of FA substituents and their total number of unsaturation. Therefore, isomeric structures, such as isomers with various chain length, double bond location, and region specificity of the fatty acid substituents has been neglected, and thus, significantly reduce the number of lipid species. Therefore, the number (829) identified sounds too large to me.

We agree that many of the species reported here are annotated as sum composition. For these species, while the sum composition is presented, they are isometrically separated by elution time and have been differentially annotated with the (a) and (b) notations, e.g. PI(40:4) (a) and PI(40:4) (b).

To address the coverage of the platform, we have provided the annotation and relative concentrations of all the individually reported species, the sum class totals and the reported values of lipids measured in Supplementary Table 31.

3. In mass spectrometry, relative intensity (or intensity) for each ion in a complete lipidome is readily available. Although it becomes common practice in the literatures using heat map for presentation, it is just way too ambiguous. Present the key data as numbers would be very helpful for readers.

We have added the full lipidome in Supplementary Table 27, 31 and 32.

Specifically, we have included in the Source Data:

Main Figures	Figure Panel	Source data location
5	Panel A	Supplementary Table 29
5	Panel C	Source data 2
5	Panel F	Supplementary Table 31
5	Panel G	Supplementary Table 34

Extended Data	Figure Panel	Source data location
11		Supplementary Table 30
12		Supplementary Table 33
13		Supplementary Table 33

Supplementary Figures	Figure Panel	Source data location
10	Panel A	Supplementary Table 28 (column A-O)
10	Panel B-C	Supplementary Table 30 (column AU-BQ)
10	Panel D	Supplementary Table 28 (column A-O)
10	Panel E	Supplementary Table 30 (column BR-CO)
11	Panel A-B	Supplementary Table 30 (column A-AN)
13		Supplementary Table 33

4. To be able to use proteomics and lipidomics assay, in combination with computer software for data handling to analyze a large number of proteins and lipids, and their interactions to pinpoint that PS(36:1) and CE(18:0) are the conserved EV biological markers that differentiates between EV and non-EV particles is an achievement. However, I am surprised by the fact that the detailed structure of PS(36:1) has not been characterized. It would be straightforward to use tandem mass spectrometry to define the structure, including the identity of the fatty acid chains and their position on the glycerol bond, and even the location of double bond. It is also desirable that the structure of CE(18:0) be verified and quantified.

Characterization of PS : We have re-run the EV samples and have utilized negative ionisation mode screening for the head group neutral loss (-87 Da) and the fragment ions corresponding acyl tails. Signal was observed only for 18:0 and 18:1 corresponding to PS 18:0_18:1. This has been added as Supplementary Figure 19.

Additional methods were added into the manuscript:

Line 828, page 19, As PS 36:1 represented a strong signature for EVs, we determined the structure of this PS species in isolated EV's, we re-ran pooled samples under negative ionisation mode under the same gradient conditions screening for product ions corresponding to the serine head group (788.5 m/z -> 701.5 m/z) and the fatty acyl tails. The final composition was annotated to be PS 18:0_18:1 with signals observed with 283.3 and 281.3 product ions.

Line 834, page 19, Characterisation of CE: The method run for these samples monitors CE's in positive ionisation mode as an ammonium adduct monitoring for the cholesterol ion with additional water loss (CE 18:0 [M+NH₄]⁺ 670.6 / 369.3). We have a matching internal standard (CE 18:0 d6) to provide quantitation for this lipid species. Retention time, mass and fragmentation confirms the current annotation.

Other comments

1. I am not sure that I know what glycoposphoserines is. Is that glycerophosphoserine instead? Are you sure its presence in the lipidome?

This was an error and should have read phosphatidylserine, it is the normal glycerophospholipid and not anything unusual. We have corrected this on line 342 page 9.

2. Equipped with nanospray ion source in positive mode should be positive ion mode.

We have clarified this in the methodology and have rephrased it to 'Equipped with a nano spray ion source operating in positive ion mode' (Line 714, page 17). We should clarify that this is "Nano liquid chromatography–tandem mass spectrometry" for our proteomics pipeline.

3. Reference(s) are required for Table 1. of EV and NonEV protein features and Table 2. List of EV and NonEV lipid features.

As requested, this has been updated for Tables 1-2

4. Supplementary Figure 7. Lipidome profiling of circulating EVs is confusing.

We have provided the legend to clarify this figure (now as Supplementary Figure 9)

In regard to the old Supplementary Figure 7. Lipidome profiling of circulating EVs (which now is "Supplementary Figure 9" in this revised version), we provide greater clarity on the figure legend in the revised manuscript:

Old version: **Lipidome profiling of circulating EVs.** **A.** Lipid abundance (MS-based intensity) for In vitro EVs, pEVs, p100K, pDGS.LD, and cells (WCL). **B.** Heatmap of Pearson correlation matrix of lipid abundance between datasets.

Revised version:

Supplementary Figure 9: Lipidome profiling of circulating EVs. **A.** The lipid abundance in indicated lipidomes (pEVs; n=12, p100K; n= 6, pDGS.LD; n=5, In vitro EVs; n=12, cells; n=12) was subjected to variance stabilization normalization (vsN) and depicted as boxplot. **B.** Heatmap of Pearson correlation matrix of lipid abundance (vsN-normalized) between lipidomes.

5. Supplementary Figure 8. Lipid species co-enriched in circulating EVs and EVs from cell culture??? is also confusing. For example, what does TG (54:6) [NL-22:6] mean? There is another TG (54:6) [NL-20:4] in the listing. Are they isomers? I would believe there are more than two isomers. What are the exact structures? I think it is much clearer that the molecular ions (e.g., [M+H]⁺, or other adduct ions such as NH₄⁺ in the positive ion; or [M+H]⁻ or other adduct ions such as [M+Cl]⁻ in the negative ion mode) be presented, followed by their elemental compositions, and then the structures and the relative intensity. So are the other lipid families and the individual lipid species.

We have clarified part of this with (Supplementary Table 41). The [NL-22:6] corresponds to the specific fragment lost monitored for the MRM transition. TG (54:6) [NL-22:6] measurement indicates a triglyceride with 54 carbons and 6 double bonds, containing at least one 22:6. While there are more isomers than the two measured for TG 54:6, we only targeted the few to cover the major key biological variances across different TG species.

We have added Supplementary Table with the molecular adducts measured for each of the lipid classes (Supplementary Table 40) with the species and their classes provided in Supplementary Table 27.

6. The Supplementary Tables have no content (only Table Legends are shown). Thus, there is no way to check (validate) the authenticity of the lipidome as claimed.

Table missing

We thank the reviewer for bringing this to our attention. We confirm that all Supplementary Tables (which from our understanding needed to be downloaded individually, just as the Legends Table) were provided in the original submission; however, we acknowledge that there may have been a formatting or file upload/access issue that led to only the table legends being visible. In this revised version, we have ensured that all Supplementary Tables are now fully included and accessible, including Supplementary Table 40, which details the conditions for tandem mass spectrometry analysis of lipid species. These comprehensive tables provide the full dataset supporting our lipidomic analysis and allow for independent validation of the results as presented in the manuscript.

Reviewers' comments:

Reviewer #1 (Remarks to the Author):

The revised manuscript by Rai et al along with authors' responses to critique clarify and extend the number of findings being reported. With these thorough revisions and thoughtful comments, the manuscript positions itself as a foundational data repository of solid proteomic and lipidomic profiling of extracellular vesicles (EVs) in human plasma. The technical quality of the work is quite impressive and in accordance with leading standards in the field. It is very likely that the data contained in this manuscript will be used as a major resource by the EV research community and may help benchmark scores of future studies into liquid biopsy of multiple disease states, in ways exemplified by the new analysis of CAC provided by the authors. This is very commendable and of great value.

This said, and with the utmost respect to the authors, and their efforts, the current paper does not drive major new concepts, disruptive technologies, or solutions to fundamental biological questions. Maybe this is not important given other assets, but it should be said. Some aspects of the narrative are also not entirely convincing, and the authors may reconsider revising them, or removing the related data (there is no shortage of those in the paper). The case in point is the depiction of protein interactome (Fig. 2F), which for bulk data combining a highly heterogeneous population of EVs from myriads of contributing cells is completely uninformative, even confusing. The authors may look upon this as “coordinated molecular patterns”, which is rather implicit given known pathways of EV biogenesis, but linkages between proteins residing in physically separate biological compartments (EVs or cells) do not signify any real molecular “interaction”. This is very simple, and complex commentary does not make this any more convincing.

We sincerely thank Reviewer #1 for their generous assessment of the technical rigor and foundational value of our study. We are grateful that the reviewer recognizes the potential of this work as a long-term resource for the EV research community, and we appreciate the thoughtful feedback aimed at improving clarity.

Regarding the concern about the depiction of the protein interactome (old Fig. 2F), we concur with the reviewer that the observed enrichment of functionally related proteins does not imply direct physical interactions, especially in the context of heterogeneous EV populations. In the revised manuscript, we clarified that these associations reflect correlative, functional enrichment that are consistent with known EV biogenesis pathways.

In response to this critique, we have made the following targeted revisions:

- a) We have fundamentally reframed the description of this figure to explicitly state that it reflects coordinated molecular patterns, not physical or spatial protein - protein interactions.
- b) We have softened all language in the text that previously implied molecular interaction or network connectivity.
- c) We have relocated the original Fig. 2F to the supporting Extended Data section (now referred to as Extended Data 7A) to de-emphasize any unintended interpretation of mechanistic interaction.

Revised manuscript text (main Results section, page 6, line 203):

“These pEV protein features showed coordinated molecular patterns across functional groups associated with “vesicular transport”, “actin cytoskeleton regulation”, and “membrane-raft assembly” (see Extended Data 7A for enrichment-based associations). Upon closer inspection, these correlated molecular patterns were conserved across individual EV proteomes at the cellular level or within specific EV subpopulations (i.e., CD9/CD81/CD63-positive EVs) (Extended Data 7B, Supplementary Table 15). Moreover, these enrichment trends were observed in EVs from primary human fibroblasts (Extended Data 7C) and were enriched in EVs relative to parental cell proteomes³⁰ (Extended Data 7D).”

We have also modified the Discussion (Page 13, line 518):

“For instance, enrichment patterns observed in our study (Extended Data 7A) reflects functionally associated protein modules enriched in the plasma EV pool. While these associations do not imply direct molecular interactions or spatial co-localization within individual EVs, they highlight coordinated expression patterns that are recurrent across EV subtypes and cell types. “

We trust this revised framing more clearly communicates our intent and aligns with the reviewer’s comment. We appreciate the reviewer’s guidance on this point and believe these adjustments enhance the overall clarity and integrity of the manuscript.

Reviewer #2 (Remarks to the Author):

In the response to Reviewer #2 (comment 6), the authors state that the machine learning models extract multimarker patterns to discriminate pEVs. However, it appears that all the identified 182 protein features were used as input. How would this strategy translate to routine practice? Would one need to quantify the entire protein panel and then apply the model to classify samples as pEV or not? In addition, the authors note that both Naïve Bayes and neural-network algorithms identify the multi-feature signatures. Does the signature here refer to the entire set of proteins provided to the model, or do these algorithms pinpoint a smaller biomarker subset with the strongest discriminatory power? It’s not clear what the value of adding machine learning models would be here.

We thank the reviewer for this insightful question regarding the translational feasibility and added value of our machine learning (ML) framework.

We agree that, in the benchmarking phase of our study, classifiers were initially trained on the full set of 182 quantified proteins to evaluate the maximal discriminatory potential of plasma EVs. However, our workflow goes beyond this comprehensive input by applying recursive feature elimination to systematically reduce dimensionality and identify a robust minimal signature. This process highlighted the protein ADAM10 and the lipid species PS(36:1) and CE(18:0) as sufficient to accurately classify samples as pEV or non-pEV (**Supplementary Fig. 14C** below left panel, **Fig. 6 F**, below right panel). These key features were subsequently experimentally validated in our study (**Fig. 6 G-H**), underscoring their biological relevance and clinical translatability.

Therefore, routine implementation would not require quantifying the entire protein/lipid panel. Instead, only these minimal markers - readily quantifiable using targeted assays such as ELISA, targeted mass spectrometry, or aptamer-based sensing platforms (e.g., fluorescent, electrochemical, or nanopore-based) - would be needed. This significantly reduces assay complexity, cost, and turnaround time, while maintaining high classification accuracy.

Moreover, the machine learning models were critical for identifying multi-marker patterns that could not be captured using traditional univariate methods. Several individual proteins can have low discriminatory power in isolation but contributed meaningfully when combined, highlighting the advantage of ML in detecting subtle, synergistic signals.

In addition, our model outputs a ranked feature list (e.g., top 30 features by importance, **Supplementary Table 35**), providing the research community with a valuable resource. This allows others to prioritize markers based on commercially available or in-house developed reagents (e.g., aptamers, antibodies) or explore alternative subsets tailored to specific disease contexts or technical platforms.

To enhance clarity in the manuscript, we have made following inclusions to reiterate that machine learning framework yields a biologically validated, minimal signature of EV markers (Results section, page 9, line 370 & page 10, line 385):

“To facilitate routine implementation and enhance translational feasibility, we applied Recursive Feature Elimination to reduce dimensionality and identify a minimal subset of features that retained high classification accuracy.”

“To enhance translational feasibility, we applied RFE to reduce dimensionality and identify a minimal lipid signature with high classification accuracy.”

We have also made following inclusion in the Discussion (Page 11, line 450):

“In addition to this resource, we identified a minimal set of biomarkers - ADAM10 (protein), PS(36:1), and CE(18:0) (lipids) - with strong discriminatory power between EVs and NonEVs. This reduced marker set is compatible with targeted, scalable assays such as ELISA or targeted mass spectrometry, making it highly practical for clinical translation. Furthermore, the ranked feature list generated by our machine learning framework provides a valuable resource for the research community, enabling prioritization of alternative markers based on available reagents or disease-specific applications.”

We believe this addresses the reviewer’s concerns and emphasizes the practical utility and added value of our ML-driven approach.

Reviewer #3 (Remarks to the Author):

Since the lipidomic analysis is based on LC/triple quadrupole mass spectrometer, I am really troubled by the authors' assertion that, e.g. the single transition 772.6 / 184.1 to cover 5 species including two isomers of PC 35:2, two isomers of PC O-36:2 and 1 isomer of PC P-36:1 in plasma / fractions. m/z 184 is a common product ion for PC and sphingomyelin. I really do not believe that the described 5 PC species can be separated by HPLC, in particular, ion mobility was not incorporated for the analysis. As far as I know, all the five species are very rare, and if they exist at all. They would be very minor (very low intensity), and I am sure no way that the feature ions specific for plasmalogen (P-36:1) or 1-O-alkyl PC (PC O-36:2) lipids can be generated by CID on their [M + H]⁺ ions for structural identification or confirmation. PC 35:2 is an odd-chain FA containing phospholipid, which is known to be very rare in mammalian system. Species like PC(37:5)*, PC(37:6)*, PC(39:6) (a)*, PC(39:6) (c)* are also odd chain FA containing PC and are extremely rare. On the other hand, if high resolution mass spectrometry were employed, at the very least, diacyl-PC and alkyl/alkenyl can be differentiated (but not by a triple quadrupole instrument). Thus, unless a solid proof of their existence, they should not be reported. Structural assignments pulled out from computer generated database, simply can't be the final decision for structural assignment. Again, reporting a big number of lipid species without seal of proof is meaningless, and sometimes can be just laughing matters. My point, again, is the fidelity of the reported lipid repertoire.

Original reviewer comments are red highlighted)

We thank the reviewer for raising important concerns regarding the fidelity of our reported lipid repertoire. We respectfully offer the following clarifications and supporting data to address these points, and to underscore that the lipid identifications presented in our study are the product of extensive experimental validation, not solely database-driven annotation or theoretical assignments. We have added an extra supplementary figure (**Supplementary Figure 20**) compiling the results from below but have addressed the queries with individual figures.

Reviewer comment: Since the lipidomic analysis is based on LC/triple quadrupole mass spectrometer, I am really troubled by the authors' assertion that, e.g. the single transition 772.6 / 184.1 to cover 5 species including two isomers of PC 35:2, two isomers of PC O-36:2 and 1 isomer of PC P-36:1 in plasma / fractions. m/z 184 is a common product ion for PC and sphingomyelin. I really do not believe that the described 5 PC species can be separated by HPLC, in particular, ion mobility was not incorporated for the analysis.

As the LC-QQQ instrument runs on unit resolution, there are more species observed per given scan, given the lack of high-resolution distinction between isobars. This is extensively published in literature. We agree that 184 is a common product ion for PC and SM as they both contain the phosphocholine head group. This is similar for PC O and PC P thus not surprisingly yield similar product ions under CID.

Below are chromatograms from a pooled human plasma sample (10µl of plasma, extracted into 100µl of butanol/methanol with a 1µl injection volume). This has been run on both our LC-QQQ (Agilent 6495C, **Reviewer Figure 1**), LC-QTOF (Agilent 6546, **Reviewer Figure 2**) and

using LC-Orbitrap (HFX, Thermo, **Reviewer Figure 3**), using the same chromatographic conditions. **LC conditions** – Water/ACN/IPA + 10mM ammonium formate and 5 μ M medronic acid, with Zorbax Eclipse Plus RRHD C18 2.1 x 100mm.

Reviewer Figure 1 – Transition (772.6/184.1) corresponding to PC 35:2 / PC O-36:2 / PC P-36:1 using pooled human plasma (0.1 μ l on column injection) run on an Agilent 6495C with chromatographic conditions as previously described (Agilent Infinity II HPLC).

Reviewer Figure 2 – EIC's corresponding to PC 35:2 and PC O-36:2 / PC P-36:1 using pooled human plasma (0.1 μ l on column injection) run on an Agilent 6546 QTOF with chromatographic conditions as previously described. Resolution is approximately 60,000 FWHM. The m/z 772.5851 (C₄₃H₈₃NO₈P) putatively corresponds to PC 35:2 (Top panel) while 772.6215 (C₄₄H₈₇NO₇P) corresponds to PC O-36:2 or PC P-36:1 (Panel).

Reviewer Figure 3 – EIC's corresponding to PC 35:2 and PC O-36:2 / PC P-36:1 using pooled human plasma (0.1 μ l on column injection) run on an HFX Orbitrap and a Vanquish analytical HPLC with chromatographic conditions as previously described. Analysis was run as full MS1 scan with resolution set to **240,000 FWHM**. The EIC 772.6215 (C₄₄H₈₇NO₇P) corresponds to PC O-36:2 or PC P-36:1 (top panel) while EIC 772.5851 (C₄₃H₈₃NO₈P) putatively corresponds to PC 35:2 (bottom panel).

While PC O-36:2 and PC P-36:1 are indistinguishable by exact mass (isomers) and have similar product ions (184) when protonated $[M+H]^+$, they are separated by elution time under reverse phase conditions. To confirm, we've previously exploited their sensitivity to acid hydrolysis to validate this (**Reviewer Figure 4**). This was published previously [1] and cited in this manuscript.

Acid Hydrolysis for Verification of Plasmalogen Species: To confirm several atypical plasmalogen species, two identical samples were analysed following acid hydrolysis of one sample. Plasmalogens are susceptible to acid hydrolysis which completely removes the alkenyl chain resulting in their corresponding lyso species. Lipid extracts of pooled plasma samples were dried down in glass vials and exposed to concentrated HCl vapour for 5 minutes, before reconstitution in butanol:methanol (1:1) and subsequent lipidomic analysis by LC-MS/MS.

Reviewer Figure 4 – Transition (772.6/184.1) corresponding to PC 35:2 / PC O-36:2 / PC P-36:1 using pooled human plasma (0.1 μ l on column injection) run on an Agilent 6490 with chromatographic conditions as previously described. Red trace, HCl vapor treated, Blue trace, untreated control

Our group had led the characterisation of odd and branched glycerophospholipids and is described in full detail. We can identify two major species when characterising peaks identified in our chromatographic run, using described approaches by Hsu et al [2, 3] using lithium alongside CID to obtain acyl specific fragmentation (**Reviewer Figure 5**) [1].

Reviewer Figure 5 – Lithium adducts for acyl-specific characterisation. Lithium acetate was added to the running solvent. Species corresponding to PC 35:2 was measured as their lithium adduct $[M+Li]^+$ m/z 778.6, with the loss of the phosphocholine headgroup as the product ion (m/z 595.2). Product ions corresponding to the acyl composition 17:0 (449.2) and 18:2 was observed (439.2) split across two distinct chromatographic peaks. This was then followed up by subsequent standard synthesis (see below). *Samples were analysed by an earlier LC-MS/MS method from described above.

We followed up and synthesized a standard for both PC 17:0_18:1 and **PC 16:0;Me 18:1** using thionyl chloride, and has been published and is referenced within this manuscript [1]. The below figures have been adapted from that manuscript (**Reviewer Figure 6**).

Reviewer Figure 6 - Synthesis of Glycerophospholipids. Acyl chlorides of selected fatty acid species were prepared by mixing 25 to 250 nmol of either 15-methylhexadecanoic acid (Sigma-Aldrich), 14-methylhexadecanoic acid (Sigma-Aldrich), heptadecanoic acid (Sigma-Aldrich), docosapentaenoic acid (all-cis-7,10,13,16,19, Larodan) or docosapentaenoic acid (all-cis-4,7,10,13,16, Larodan) with 20 μ L of 0.2M thionyl chloride diluted in dichloromethane for 10 minutes at room temperature. Solvent was then removed with N₂ gas at 60°C before the addition of a 10% molar ratio of soy lysophosphatidylcholine (Avanti Polar Lipids) or L- α -glycerophosphorylcholine (Sigma-Aldrich) in 100 μ L dichloromethane and incubating for 10 minutes at room temperature. The solvent was then removed under N₂ gas at 60°C and the residue containing lipid species was reconstituted in butanol:methanol (1:1 v/v) with 10mM ammonium formate. *Samples were analysed by an earlier LC-MS/MS method from described above.

Reviewer comment: As far as I know, all the five species are very rare, and if they exist at all. They would be very minor (very low intensity), and I am sure no way that the feature ions specific for plasmalogen (P-36:1) or 1-O-alkyl PC (PC O-36:2) lipids can be generated by CID on their [M + H]⁺ ions for structural identification or confirmation. PC 35:2 is an odd-chain FA containing phospholipid, which is known to be very rare in mammalian system. Species like PC(37:5)*, PC(37:6)*, PC(39:6) (a)*, PC(39:6) (c)* are also odd chain FA containing PC and are extremely rare.

One of the benefits of reverse phase LC-MS/MS is substantially improved sensitivity. We have extensively validated these chromatographic peaks as described above. Similarly, fractionation by density gradient (for our work in this paper) further increases sensitivity.

For odd and branched species, yes they are low in abundance, but the abundances of PC 35:2 are much higher than the corresponding PC O-36:2 and PC P-36:1 species in plasma (see Figure 1). There is tremendous diversity in these species, we have characterised these extensively, and the diversity extends to many unique isomers.

Branched species are no longer purely known to be derived from ruminant fats, and have been known to exist in mammals since the 1960's [4], has been reported more recently in humans [1, 5-7] and are readily separated by certain C18 and C30 columns even when esterified in

glycerophospholipids or triglycerides [1, 7]. Others have identified their existence other lipid classes including in sphingolipids [8] and have identified genes that show higher specificity to their utilisation. We have confirmed these species elute earlier than their straight-chain isomers.

Reviewer comment: On the other hand, if high resolution mass spectrometry were employed, at the very least, diacyl-PC and alkyl/alkenyl can be differentiated (but not by a triple quadrupole instrument). Thus, unless a solid proof of their existence, they should not be reported.

See above. Other groups have confirmed branched fatty acids by GC among other techniques in mammals [6]. Chromatography bypasses the limitations of using a QQQ and unit resolution, with scheduled MRM, and enables much broader coverage. We have extensively leveraged complementary techniques to validate these measurements (as above and in our publications). Furthermore, we have published substantial genetic associations with these lipids [9], highlighting the enzymatic pathways that influence their concentration in blood. We have run over >30,000 plasma samples with a version of this method in the last few years [10-14].

Reviewer comment: Structural assignments pulled out from computer generated database, simply can't be the final decision for structural assignment.

We fully agree with the reviewer that structural assignments based solely on computer-generated databases are insufficient for confident identification, particularly in complex lipidomic datasets. However, we wish to clarify that our study did not rely on in silico or untargeted database matching for lipid identification. All species reported in this manuscript were identified using targeted approaches, and their identities are not newly proposed but rather previously characterised in-depth through extensive experimental validation in our prior work.

Specifically, the lipid structures highlighted here were confirmed over years of method development and validation (as described in this response) using a combination of retention time alignment across orthogonal LC methods, fragmentation pattern analysis on multiple platforms (QQQ, QTOF, Orbitrap), Lithium adduct fragmentation to localise double bonds and backbone structure, acid hydrolysis to distinguish plasmalogens, and comparison to synthetic standards and biological replicates. These layers of validation, described in detail in our previous publications described throughout this response, underpin the structural certainty of the species shown. In this current manuscript, we focus on the biological findings using these well-validated analytes rather than proposing novel structures. We hope this clarification addresses the reviewer's concern.

Reviewer comment: Again, reporting a big number of lipid species without seal of proof is meaningless, and sometimes can be just laughing matters. My point, again, is the fidelity of the reported lipid repertoire.

We appreciate the reviewer's emphasis on the importance of rigorous structural validation, and we agree that high-confidence lipid identification is essential in lipidomics studies. However, we respectfully disagree with the implication that our reported lipid repertoire lacks proof or rigor. Our targeted lipidomic workflow has been developed, validated, and openly shared with the community. The complete list of transitions, collision energies, and retention times is publicly available via the Baker Metabolomics website (<https://metabolomics.baker.edu.au/method/>) and was published as an open-access application note in collaboration with Agilent Technologies (Application Note 5994-3747EN [15]). These

compounds have been validated across 3+ instrument types (QQQ, Orbitrap, QTOF) and we have provided evidence of compounds measured which we believe constitutes a robust "seal of proof" for the fidelity of the reported lipid species.

We have made the following inclusions (page 19, line 818):

Additional characterisation was carried out on PC, PC(O) and PC(P) isobaric and isomeric species to highlight the separation of the chromatography (**Supplementary Figure 20**).

Supplementary Figure 20. Validation of isobars and isomer separation between PC 35:2, PC O-36:2 and PC P-36:1 with chromatography. A - Transition (772.6/184.1) corresponding to PC 35:2 / PC O-36:2 / PC P-36:1 using pooled human plasma (0.1µl on column injection) run on an Agilent 6495C with chromatographic conditions as previously described (Agilent Infinity II HPLC). B - EIC's corresponding to PC 35:2 and PC O-36:2 / PC P-36:1 using pooled human plasma (0.1µl on column injection) run on an Agilent 6456 QTOF with chromatographic conditions as previously described. Resolution is approximately 60,000 FWHM. The m/z 772.5851 (C43H83NO8P) putatively corresponds to PC 35:2 (Top panel) while 772.6215 (C44H87NO7P) corresponds to PC O-36:2 or PC P-36:1 (Panel). C,D - EIC's corresponding to PC 35:2 and PC O-36:2 / PC P-36:1 using pooled human plasma (0.1µl on column injection) run on an HFx Orbitrap and a Vanquish analytical HPLC with chromatographic conditions as previously described. Analysis was run as full MS1 scan with resolution set to 240,000 FWHM. The EIC 772.6215 (C44H87NO7P) corresponds to PC O-36:2 or PC P-36:1 (top panel) while EIC 772.5851 (C43H83NO8P) putatively corresponds to PC 35:2 (bottom panel). E - Lithium

acetate was added to the running solvent. Species corresponding to PC 35:2 was measured as their lithium adduct $[M+Li]^+$ m/z 778.6, with the loss of the phosphocholine headgroup as the product ion (m/z 595.2). Product ions corresponding to the acyl composition 17:0 (449.2) and 18:2 was observed (439.2) split across two distinct chromatographic peaks. This was then followed up by subsequent synthesis of branched and straight isomer standards¹³⁴. F - Transition (772.6/184.1) corresponding to PC 35:2 / PC O-36:2 / PC P-36:1 using pooled human plasma (0.1 μ l on column injection) run on an Agilent 6490 with chromatographic conditions as previously described. Red trace, HCl vapor treated, Blue trace, untreated control.

Other comments:

The color label for Supplementary Figure 19 is very confusing.

We have updated the figure with labels.

Supplementary Figure 19. Characterization of PS 36:1 in EVs using negative ionization mode. Pooled EV's were run under the same conditions in negative ionization mode, looking for fragments corresponding to the neutral loss of the serine headgroup (red) and all possible fatty acid species, resulting in identification of 18:0 (283.3 m/z) and 18:1 (281.3 m/z).

Same problems with downloading Table 1-41 (as the first version of the manuscript). Nothing can be found in Supplemental material table 1-41, excepting 3 tables that are not useful, All the other Supplementary Tables, such as 40, Supplementary Table 27, 31 and 32. which are important, are not extractable for review.

All Supplementary and Extended Data were provided for direct access. In addition, we provided sharedrive/dropbox links to these data if access was an issue.

Direct access link:

- Supplemental Tables
https://www.dropbox.com/scl/fi/ac1emgijvcnzsqrnwu614/Supplementary-table-2024_06_10.xlsx?rlkey=bihdv2tmcijqghwnz1rb8jk73&st=80o5acbzd&dl=0
- Supplemental Figures

https://www.dropbox.com/scl/fi/3jd1xh0lod3gm9saahkhg/Supplementary-Figures_2025_04_01.pptx?rlkey=7dhey1nlbk6dbha5dt2vv0c53&st=9jt4ozu2&dl=0

Table 2. List of EV and NonEV lipid features, what are those lipids designated with “*” superscript?.

We thank the reviewer for noting this oversight. Lipid features marked with “*” in Table 2 indicate the top 25 enriched lipids in either plasma EVs or NonEVs, based on relative fold change in abundance. These were ranked by fold change to highlight the most differentially abundant lipids between the two fractions. We have now clarified this designation in the main text.

REFERENCES

1. Huynh, K., et al., *High-throughput plasma lipidomics: detailed mapping of the associations with cardiometabolic risk factors*. 2019. **26**(1): p. 71-84. e4.
2. Hsu, F.-F., et al., *Characterization of alkylacyl, alk-1-enylacyl and lyso subclasses of glycerophosphocholine by tandem quadrupole mass spectrometry with electrospray ionization*. *Journal of Mass Spectrometry*, 2003. **38**(7): p. 752-763.
3. Hsu, F.-F. and J. Turk, *Electrospray ionization/tandem quadrupole mass spectrometric studies on phosphatidylcholines: the fragmentation processes*. *Journal of the American Society for Mass Spectrometry*, 2003. **14**(4): p. 352-363.
4. Horning, M.G., et al., *Fatty Acid Synthesis in Adipose Tissue: II. ENZYMATIC SYNTHESIS OF BRANCHED CHAIN AND ODD-NUMBERED FATTY ACIDS*. *Journal of Biological Chemistry*, 1961. **236**(3): p. 669-672.
5. Green, C.R., et al., *Branched-chain amino acid catabolism fuels adipocyte differentiation and lipogenesis*. *Nat Chem Biol*, 2016. **12**(1): p. 15-21.
6. Wallace, M., et al., *Enzyme promiscuity drives branched-chain fatty acid synthesis in adipose tissues*. *Nature chemical biology*, 2018. **14**(11): p. 1021-1031.
7. Green, C.R., et al., *Quantifying acyl-chain diversity in isobaric compound lipids containing monomethyl branched-chain fatty acids*. *Journal of Lipid Research*, 2024. **65**(12).
8. Lone, M.A., et al., *Subunit composition of the mammalian serine-palmitoyltransferase defines the spectrum of straight and methyl-branched long-chain bases*. *Proceedings of the National Academy of Sciences*, 2020. **117**(27): p. 15591-15598.
9. Cadby, G., et al., *Comprehensive genetic analysis of the human lipidome identifies loci associated with lipid homeostasis with links to coronary artery disease*. *Nature Communications*, 2022. **13**(1): p. 1-17.
10. Beyene, H.B., et al., *High-coverage plasma lipidomics reveals novel sex-specific lipidomic fingerprints of age and BMI: Evidence from two large population cohort studies*. 2020. **18**(9): p. e3000870.
11. Wang, T., et al., *A lipidomic based metabolic age score captures cardiometabolic risk independent of chronological age*. *eBioMedicine*, 2024. **105**: p. 105199.
12. Huynh, K., et al., *Concordant peripheral lipidome signatures in two large clinical studies of Alzheimer's disease*. *Nat Commun*, 2020. **11**(1): p. 5698.
13. Yap, C.X., et al., *Interactions between the lipidome and genetic and environmental factors in autism*. *Nature medicine*, 2023. **29**(4): p. 936-949.
14. Wang, T., et al., *APOE ϵ 2 resilience for Alzheimer's disease is mediated by plasma lipid species: Analysis of three independent cohort studies*. *Alzheimer's & Dementia*, 2022: p. 1-16.
15. Huynh, K., et al., *A comprehensive, curated, high-throughput method for the detailed analysis of the plasma lipidome*. 2021: Agilent Application Note (5994-3747EN). p. 45.